# Understanding Train-Validation Split in Meta-Learning with Neural Networks

**Xinzhe Zuo‡, Zixiang Chen◇, Huaxiu Yao†, Yuan Cao⋆, Quqnquan Gu◇**

‡ Department of Mathematics, University of California, Los Angles
`zxz@math.ucla.edu`
◇ Department of Computer Science, University of California, Los Angles
`{chenzx19, qgu}@cs.ucla.edu`
† Department of Computer Science, Stanford University
`huaxiu@cs.stanford.edu`
⋆ Department of Statistics & Actuarial Science, University of Hong Kong
`yuancao@hku.hk`

## Abstract

The goal of meta-learning is to learn a good prior model from a collection of tasks such that the learned prior is able to adapt quickly to new tasks without accessing many data from the new tasks. A common practice in meta-learning is to perform a train-validation split on each task, where the training set is used for adapting the model parameter to that specific task and the validation set is used for learning a prior model that is shared across all tasks. Despite its success and popularity in multitask learning and few-shot learning, the understanding of the train-validation split is still limited, especially when the neural network models are used. In this paper, we study the benefit of train-validation split for classification problems with neural network models trained by gradient descent. For first-order model-agnostic meta-learning (FOMAML), we prove that the train-validation split is necessary to learn a good prior model when the noise in the training sample is large, while the train-train method fails. We validate our theory by conducting experiment on both synthetic and real datasets. To the best of our knowledge, this is the first work towards the theoretical understanding of train-validation split in meta-learning with neural networks.

## 1 Introduction

In recent years, meta-learning has gained increasing popularity and been successfully applied to a wide range of problems including few-shot learning (Ren et al., 2018; Li et al., 2017; Rusu et al., 2018; Snell et al., 2017), reinforcement learning (Gupta et al., 2018b;a), neural machine translation (Gu et al., 2018), and neural architecture search (NAS) (Liu et al., 2018; Real et al., 2019). A popular meta-learning idea is to formulate it as a bi-level optimization problem, where the inner level computes the parameter adaptation to each task, while the outer level tries to minimize the meta-training loss. Such a bi-level optimization formulation is empirically proved effective to learn new tasks quickly using only a few examples with the aid of past experience. Following this idea, meta-learning algorithms such as model agnostic meta-learning (MAML) (Finn et al., 2017) have achieved remarkable success in many applications.

Due to the nature of bi-level optimization, meta-learning algorithms can often take advantage of a train-validation split in the dataset, so that the inner and outer levels of the algorithm use different data points (Finn et al., 2017; Rajeswaran et al., 2019; Bai et al., 2021; Fallah et al., 2020). It is believed that the train-validation split can help the meta-learning algorithm to achieve a better performance. There has been several attempts to understand the importance of train-validation split in meta-learning for linear models (Wang et al., 2021; Bai et al., 2021; Saunshi et al., 2021). Specifically, Wang et al. (2021) showed that when learning linear models, the train-train method performs much worse than the train-validation method if the sample size is small and the noise is large. They also show that the train-train method is able to perform well on linear models when the sample size is large enough. Bai et al. (2021) considered the linear centroid model introduced in Denevi

et al. (2018b) and showed that the train-validation method outperforms the train-train method in an agnostic setting, while in the realizable noiseless setting, the train-train method can asymptotically achieve a strictly smaller mean square error than the train-validation method as the sample size and dimension go to infinity at a fixed ratio. Saunshi et al. (2021) considered a representation learning perspective, and demonstrated that train-validation split encourages the learned representation to be low-rank while the train-train method encourages high-rank representations. However, all these works focus on the linear regression setting, while the advantage of train-validation split remains elusive for meta-learning with neural networks.

Based on the above observation, we raise the following question:

> *How does train-validation split affect the meta learning with neural networks?*

In this paper, we answer the above question via a case study of few-shot binary classification using a two-layer convolutional neural network. We consider a learning problem where the data model consists of large noises, and only a limited number of data are available. Under this setting, we theoretically compare the performance of first-order MAML (FOMAML (Finn et al., 2017)), which is a simplification of MAML that ignores the hessian terms, with a train-validation split (the train-validation method) and without a train-validation split (the train-train method). We summarize our contributions as follows:

1. We show that under our setting, despite the complex bi-level structure of the FOMAML loss and its non-convex landscape, it is guaranteed that the train-validation method and the train-train method can both train a two-layer CNN to a global minimum of the training loss with high probability.

2. We also demonstrate that there is a significant performance gap on new test data. Specifically, we show that the neural network trained by the train-validation method can achieve a test loss that decreases exponentially fast as the number of training tasks increase. On the other hand, we also show that the train-train method can at best achieve a constant level test loss.

3. Our study demonstrates the importance of train-validation split in learning neural networks. To the best of our knowledge, this is the first theoretical work studying train-validation split of meta-learning with neural networks. Notably, the learning problem we consider is linearly realizable, for which Bai et al. (2021) showed that the train-train method can asymptotically achieve better MSE than the train-validation method under a linear model. However, our results give the opposite conclusion for learning CNNs – the train-validation method still outperforms the train-train method even for linearly realizable learning problems. Therefore, our results indicate that train-validation split may have a more significant advantage when learning complicated prediction models.

4. We perform experiments on both synthetic and real datasets with neural networks as backbone model to justify our theoretical results. In particular, even when the data and the neural network structure do not meet our theoretical assumptions, the experiment results still corroborate our theory to a certain extent. This demonstrates the practical value of our analysis.

**Notation.** For an integer $k$, we denote $[k] = \{1, 2, \ldots, k\}$. Given two sequences $\{x_n\}$ and $\{y_n\}$ with $y_n > 0$, we denote $x_n = \mathcal{O}(y_n)$ if $|x_n|/y_n$ is upper bounded by a constant for all $n$. Similarly, we denote $x_n = \Omega(y_n)$ if $|x_n|/y_n$ is lower bounded by a positive constant. We denote $x_n = \Theta(y_n)$ if $x_n = O(y_n)$ and $x_n = \Omega(y_n)$. Finally, we use $\widetilde{\mathcal{O}}(\cdot)$, $\widetilde{\Omega}(\cdot)$, and $\widetilde{\Theta}(\cdot)$ to hide logarithmic factors.

## 2 ADDITIONAL RELATED WORK

**Optimization and generalization guarantees for meta-learning.** A number of recent works studied the optimization guarantees for meta-learning algorithms. Finn & Levine (2017) proved the universality of gradient based meta-learning. Wang et al. (2020) studied the global optimality conditions for MAML with a nonconvex objective. Fallah et al. (2020) studied the convergence guarantee for MAML with nonconvex loss function and proposed Hessian-Free MAML with the same theoretical guarantee of MAML without accessing second order information. Finn et al. (2019); Balcan et al. (2019); Khodak et al. (2019); Denevi et al. (2019; 2018a) studied online meta-learning using online convex optimization. Another series of works have also studied the generalization error and sample complexities of meta-learning methods. Specifically, Amit & Meir (2018) extended the PAC-Bayes argument to the meta-learning setting and established a generalization error bound for

meta-learning. Saunshi et al. (2020) analyzed the sample complexity under the nonconvex setting using Reptile (Nichol et al., 2018). There has also been a stream of work studying generalization in meta-learning from the representation learning perspective (Du et al., 2020; Tripuraneni et al., 2020; Denevi et al., 2018a). These works do not employ a train-validation split, instead they assume some underlying low-rank constraints on the representation. Chen et al. (2022); Huang et al. (2022); Wang et al. (2022) also studied generalization error bounds for meta-learning in the over-parametrized regime.

**Feature learning in over-parameterized neural networks.** There is a series of recent work studying the feature learning dynamics of neural networks. For example, Allen-Zhu & Li (2020a) studied a sparse coding model and showed that adversarial training can help CNN filters to learn "pure" dictionary bases. Allen-Zhu & Li (2020b) studied the impact of ensemble and knowledge distillation on the feature learning process. Zou et al. (2021) demonstrated the generalization gap between Adam and stochastic gradient descent through the lens of feature learning. Cao et al. (2022) studied feature learning and noise memorization in learning two-layer CNNs and showed a phase transition between benign and harmful overfitting.

## 3 PROBLEM SETUP

In this section, we introduce our data model, neural network model, loss function, and the details about the FOMAML algorithm with train-train and train-validation methods.

We first introduce our data model. In meta-learning, the goal is to train a model based on the data from $K$ tasks, so that the trained model can learn a new task efficiently. Theoretical analysis of meta-learning thus requires careful modeling of (i) the relation among different tasks, and (ii) the data distribution for each specific task. To achieve this, we suppose that the data distribution $\mathcal{D}_k$ for the $k$-th task is defined based on a vector $\boldsymbol{\nu}_k$, i.e., $\mathcal{D}_k = \mathcal{D}(\boldsymbol{\nu}_k)$, and that the vectors $\boldsymbol{\nu}_1, \ldots, \boldsymbol{\nu}_K$ are independently drawn from a distribution $\Pi$. Following this framework, the data points of a new task can be generated by first sampling a vector $\tilde{\boldsymbol{\nu}}$ from $\Pi$, then sampling data from the distribution $\mathcal{D}(\tilde{\boldsymbol{\nu}})$. In the following, we present the detailed definitions of $\Pi$ and $\mathcal{D}(\tilde{\boldsymbol{\nu}})$.

**Definition 3.1** (Distribution of tasks). Let $\boldsymbol{\nu}, \mathbf{z}_1, \ldots, \mathbf{z}_M \in \mathbb{R}^d$ be fixed vectors, where $\mathbf{z}_1, \ldots, \mathbf{z}_M$ are orthogonal to $\boldsymbol{\nu}$. A vector $\tilde{\boldsymbol{\nu}}$ is generated from $\Pi$ by (i) randomly pick a vector $\mathbf{z}$ from $\{\mathbf{z}_1, \ldots, \mathbf{z}_M\}$, and (ii) let $\tilde{\boldsymbol{\nu}} = \boldsymbol{\nu} + \mathbf{z}$.

Definition 3.1 is based on a set of fixed vectors $\boldsymbol{\nu}, \mathbf{z}_1, \ldots, \mathbf{z}_M \in \mathbb{R}^d$. Here $\boldsymbol{\nu}$ captures the feature shared by all tasks. And $\mathbf{z}_1, \ldots, \mathbf{z}_M$ give a dictionary of possible unique features of each specific task. It is important to note that we focus on the setting where the number of observed tasks $K \ll M$. Under our setting, with high probability, different tasks will use different unique features.

**Definition 3.2** (Distribution of data). Given a vector $\tilde{\boldsymbol{\nu}} \in \mathbb{R}^d$, each data point $(\mathbf{x}, y)$ with $\mathbf{x} = [\mathbf{x}^{(1)\top}, \mathbf{x}^{(2)\top}]^\top \in \mathbb{R}^{2d}$ and $y \in \{-1, 1\}$ is generated from $\mathcal{D}(\tilde{\boldsymbol{\nu}})$ as follows:

1. The label $y$ is assigned as $+1$ or $-1$ with equal probability.

2. A noise vector $\boldsymbol{\xi}$ is generated from $\mathcal{N}(\mathbf{0}, \sigma_\xi^2 \cdot (\mathbf{I} - \mathbf{P}))$, where $\mathbf{P} \in \mathbb{R}^{d \times d}$ is the projection operator onto $\mathrm{span}(\{\boldsymbol{\nu}, \mathbf{z}_1, \ldots, \mathbf{z}_M\})$.

3. One of $\mathbf{x}^{(1)}, \mathbf{x}^{(2)}$ is randomly selected and assigned as $y \cdot \boldsymbol{\nu}_k$; the other is assigned as $\boldsymbol{\xi}$.

For $k \in [K]$, we denote by $\mathcal{S}_k = \{(\mathbf{x}_{k,i}, y_{k,i})\}_{i=1}^n$ the set of independent samples from the $k$-th observed task. We consider a specific type of data input that consists of two patches, $\mathbf{x}_{k,i} = [\mathbf{x}_{k,i}^{(1)\top}, \mathbf{x}_{k,i}^{(2)\top}]^\top$, to meet our study of convolutional neural networks. Note also that Definition 3.2 requires the noises in the data input to be sampled from the Gaussian distribution $\mathcal{N}(\mathbf{0}, \sigma_\xi^2 \cdot (\mathbf{I} - \mathbf{P}))$ to ensure that the noise vector $\boldsymbol{\xi}$ is orthogonal to the features $\boldsymbol{\nu}, \mathbf{z}_1, \ldots, \mathbf{z}_M$. It is then easy to check that our data model is *linearly realizable*: the linear predictor $\boldsymbol{\theta}^* = \|\boldsymbol{\nu}\|_2^{-2} \cdot [\boldsymbol{\nu}^\top, \boldsymbol{\nu}^\top]^\top \in \mathbb{R}^{2d}$ satisfy $\langle \boldsymbol{\theta}^*, \mathbf{x} \rangle = y$ for all $(\mathbf{x}, y)$ drawn from our data distribution. Our motivation to study this linearly realizable setting is that it has been proved in Bai et al. (2021) that the train-train method is strictly better than the train-validation method when learning linear models in the realizable setting. On the contrary, in this paper we aim to show that for the linearly realizable data model defined in Definitions 3.1 and 3.2, the train-validation method can still outperform the train-train method for FOMAML in learning two-layer CNNs.

We study a two-layer CNN with $m$ hidden layer neurons whose second layer weights are frozen as $\pm 1$'s. We use the Huberized-**ReLU** (Chatterji et al., 2021) activation function, which is defined as

$$
\sigma(z) = \begin{cases} 0, & z < 0 \\ z^2/(2h), & z \in [0, h] \\ z - h/2, & \text{otherwise.} \end{cases}
$$

where we set $h = 1/2$ in our analysis. Let $\mathbf{W}$ represent the collection of all weights of our network. For a data input $\mathbf{x} = [\mathbf{x}^{(1)\top}, \mathbf{x}^{(2)\top}]^\top$, we consider the CNN $f(\mathbf{W}, \mathbf{x}) = F_{+1}(\mathbf{W}_{+1}, \mathbf{x}) - F_{-1}(\mathbf{W}_{-1}, \mathbf{x})$, where

$$
F_j(\mathbf{W}_j, \mathbf{x}) = \sum_{r=1}^{m} \sum_{p=1}^{2} \sigma(\langle \mathbf{w}_{j,r}, \mathbf{x}^{(p)} \rangle), \; j \in \{-1, 1\}.
$$

We use $\mathbf{w}_{j,r}$ to denote the $r$-th convolution filter with second layer weight $j$, and use $\mathbf{W}_j$ to denote the collection of $\mathbf{w}_{j,1}, \ldots, \mathbf{w}_{j,m}$. We consider cross-entropy loss. The loss of a data point $(\mathbf{x}, y)$ is given by $\mathcal{L}(\mathbf{W}, \mathbf{x}, y) = \ell[y \cdot f(\mathbf{W}, \mathbf{x})]$, where $\ell(z) = \log(1 + \exp(-z))$. The loss of a collection of data points $\mathcal{S}$ is defined by

$$
\mathcal{L}(\mathbf{W}, \mathcal{S}) = \frac{1}{|\mathcal{S}|} \sum_{(\mathbf{x}, y) \in \mathcal{S}} \mathcal{L}(\mathbf{W}, \mathbf{x}, y).
$$

Following the data model given in Definitions 3.1 and 3.2, we define the test loss achieved by a CNN with weights $\mathbf{W}$ as $\mathcal{L}_{\text{test}}(\mathbf{W}) := \mathbb{E}_{\tilde{\boldsymbol{\nu}} \sim \Pi, (\mathbf{x}, y) \sim \mathcal{D}(\tilde{\boldsymbol{\nu}})} \ell(y \cdot f(\mathbf{W}, \mathbf{x}))$. We implement the FOMAML algorithm (Finn et al., 2017) to train the neural network. The train-train and train-validation methods are given as follows.

**Train-train**: for each task $k$, we use all of the samples for adapting the parameter in the inner-loop updates. Specifically, the meta objective is to minimize

$$
\widehat{\mathcal{L}}^{\text{tr-tr}}(\mathbf{W}, \{\mathcal{S}_k\}_{k=1}^{K}) = \frac{1}{K} \sum_{k=1}^{K} \mathcal{L}(\widetilde{\mathbf{W}}(\mathbf{W}, \mathcal{S}_k), \mathcal{S}_k),
$$

where $\widetilde{\mathbf{W}}(\mathbf{W}, \mathcal{S}_k)$ represents the weights of the network after $J$ gradient descent steps (w.r.t. loss $\mathcal{L}(\cdot, \mathcal{S}_k)$) starting from $\mathbf{W}$ with step size $\gamma$. The FOMAML algorithm updates the CNN weights using the following gradient update rule with step size $\eta$:

$$
\mathbf{W}^{(t+1)} = \mathbf{W}^{(t)} - \eta \cdot \frac{1}{K} \sum_{k=1}^{K} \nabla_{\mathbf{W}} \mathcal{L}(\mathbf{W}, \mathcal{S}_k)|_{\mathbf{W} = \widetilde{\mathbf{W}}(\mathbf{W}^{(t)}, \mathcal{S}_k)}. \tag{3.1}
$$

**Train-validation**: for each task $k$, we denote by $\mathcal{I}_k^{\text{tr}} = \{1, \ldots, n_1\}$ the training data indices, and $\mathcal{I}_k^{\text{val}} = \{n_1 + 1, \ldots, n\}$ the validation data indices. We then use $\mathcal{S}_k^{\text{tr}} = \{(\mathbf{x}_{k,i}, y_{k,i})\}_{i \in \mathcal{I}_k^{\text{tr}}}$ as the training data set, and $\mathcal{S}_k^{\text{val}} = \{(\mathbf{x}_{k,i}, y_{k,i})\}_{i \in \mathcal{I}_k^{\text{val}}}$ as the validation data set. The meta objective of the train-validation method is to minimize

$$
\widehat{\mathcal{L}}^{\text{tr-val}}(\mathbf{W}, \{\mathcal{S}_k\}_{k=1}^{K}) = \frac{1}{K} \sum_{k=1}^{K} \mathcal{L}(\widetilde{\mathbf{W}}(\mathbf{W}, \mathcal{S}_k^{\text{tr}}), \mathcal{S}_k^{\text{val}}),
$$

where $\widetilde{\mathbf{W}}(\mathbf{W}, \mathcal{S}_k^{\text{tr}})$ represents the weights of the network after $J$ gradient descent steps (w.r.t. loss $\mathcal{L}(\cdot, \mathcal{S}_k^{\text{tr}})$) starting from $\mathbf{W}$ with step size $\gamma$. For the train-validation method, the FOMAML algorithm implements the following outer-loop update rule to train the network:

$$
\mathbf{W}^{(t+1)} = \mathbf{W}^{(t)} - \eta \cdot \frac{1}{K} \sum_{k=1}^{K} \nabla_{\mathbf{W}} \mathcal{L}(\mathbf{W}, \mathcal{S}_k^{\text{val}})|_{\mathbf{W} = \widetilde{\mathbf{W}}(\mathbf{W}^{(t)}, \mathcal{S}_k^{\text{tr}})}. \tag{3.2}
$$

## 4  MAIN RESULTS

In this section, we present our theoretical results for the train-train and train-validation methods. We first introduce the following condition.

**Condition 4.1.** There exists $\sigma_s > 0$ such that $(1/2) \cdot \sigma_s \sqrt{d} \leq \|\mathbf{z}_i\|_2 \leq (3/2) \cdot \sigma_s \sqrt{d}$ for $i \in [M]$, and $\langle \mathbf{z}_i, \mathbf{z}_j \rangle \leq \mathcal{O}(\sigma_s^2 \cdot \sqrt{d \log(d)})$ for all $i \neq j$.

Condition 4.1 specifies the properties of the task-specific features in Definition 3.1. Here we require that the task-specific features to have weak correlations to each other, which is standard in dictionary learning. Moreover, it is easy to see that such $\mathbf{z}_1, \ldots, \mathbf{z}_M$ exist as long as $M = \mathcal{O}(\text{poly}(d))$: with a simple Gaussian concentration argument, one can show that $M$ independent zero-mean Gaussian random vectors with covariance matrix $\sigma_s^2(\mathbf{I} - \boldsymbol{\nu}\boldsymbol{\nu}^\top/\|\boldsymbol{\nu}\|_2^2)$ are orthogonal to $\boldsymbol{\nu}$ and satisfy Condition 4.1 with high probability.

With Condition 4.1 and our data model in Definitions 3.1 and 3.2, it is necessary for a neural network to learn the shared feature $\boldsymbol{\nu}$ to make accurate predictions on a new task. On the other hand, if a neural network only utilizes the task-specific features or the noises in the observed data to fit the labels, then it can only achieve a good training loss, but will have a poor prediction accuracy on new tasks. Following this intuition, we aim to construct a concrete setting under which the train-train and train-validation methods give different test losses while both achieving small training losses. The details of our constructed setting is summarized in the following condition.

**Condition 4.2.** $\|\boldsymbol{\nu}\|_2 = 1$, $\sigma_\xi = d^{-1/2} \cdot \text{polylog}(d)$, $\sigma_s = d^{-1/2}/\text{polylog}(d)$, $n = \Theta(1)$, $K = \text{polylog}(d)$, $m = \text{polylog}(d)$, $\Omega(d^{1/2}) \leq M \leq d/2$.

Condition 4.2 defines an over-parameterized setting where $d \gg Kn$. Moreover, under Condition 4.2, the norms of the shared and task-specific features $\|\boldsymbol{\nu}\|_2, \|\mathbf{z}_1\|_2, \ldots, \|\mathbf{z}_K\|_2$ are all $\mathcal{O}(1)$. In comparison, with high probability, the norms of noise vectors in the data are $\Theta(\text{polylog}(d))$. Therefore, Condition 4.2 defines a setting with relatively large noises.

The FOMAML algorithm involves inner and outer-loop training of the neural network. We specify our detailed hyper-parameter configurations in the following condition.

**Condition 4.3.** In the FOMAML algorithm defined in Eq. (3.1) and Eq. (3.2), we initialize the CNN weights $\mathbf{W}^{(0)}$ by Gaussian random initialization with standard deviation $\sigma_0 = d^{-1/2}$. We set the inner-loop step size $\gamma = \text{polylog}(d)$, and the outer-loop step size $\eta = 1/\text{polylog}(d)$. We run $T = \text{poly}(d)$ outer-loop iterations. Within each outer-loop iteration, we run $J = 5$ inner-loop gradient descent steps.

In Condition 4.3, we use a slightly larger step size for inner-loop compared to that for the outer-loop. This is to ensure that the inner-loop updates can make a difference, and it matches the meta-learning practice. Moreover, we set $J = 5$ to demonstrate that our analysis applies to the case with multiple inner-loop iterations. The exact value of $J$ is not essential – we can easily apply our analysis to other values of $J = \Theta(1)$.

We are now in the position to state our main theoretical results. The theorem below gives the training and test loss guarantees for the train-train method.

**Theorem 4.4.** *Under Conditions 4.1, 4.2 and 4.3, suppose that one uses the train-train method to train the neural network. Then with probability at least $1 - (Kn)^{-10}$,*

*1. the training loss is small:*

$$\min_{t \in [T]} \widehat{\mathcal{L}}^{\text{tr-tr}}(\boldsymbol{W}^{(t)}, \{\mathcal{S}_k\}_{k=1}^K) \leq \mathcal{O}\left(\frac{1}{\text{poly}(d)}\right).$$

*2. the test loss is large:*

$$\min_{t \in [T]} \mathcal{L}_{\text{test}}(\boldsymbol{W}^{(t)}) = \Omega(1).$$

Theorem 4.4 demonstrates that FOMAML with train-train method can successfully train the CNN to minimize the training loss. However, it also shows that the train-train method fails on new tasks and can only achieve a constant level test loss. In comparison, we have the following theorem for the train-validation method.

**Theorem 4.5.** *Under Conditions 4.1, 4.2 and 4.3, suppose that one uses the train-validation method to train the neural network. Then with probability at least $1 - (Kn)^{-10}$,*

1. *the training loss is small:*

$$\min_{t \in [T]} \widehat{\mathcal{L}}^{\mathrm{tr-val}}(\boldsymbol{W}^{(t)}, \{\mathcal{S}_k\}_{k=1}^K) \leq \mathcal{O}\left(\frac{1}{\mathrm{poly}(d)}\right).$$

2. *the test loss is also small: there exists a constant $c > 0$ such that*

$$\mathcal{L}_{\mathrm{test}}(\boldsymbol{W}^{(T)}) = \mathcal{O}(\exp(-K^c)).$$

Although both methods can achieve a $1/\mathrm{poly}(d)$ training loss, the train-validation method is able to achieve a test loss of $\mathcal{O}(\exp(-K^c))$ compared to a constant test loss $\Omega(1)$ by the train-train method. Thus, Theorems 4.4 and 4.5 show that for our data model, it is necessary to perform a train-validation split to achieve good performance on test data.

We also compare FOMAML with the vanilla supervised learning framework where one simply combines all data from the $K$ tasks together and uses gradient descent to minimize the overall cross-entropy loss. In fact, the results of Theorem 4.4 still hold even for $J = 0$, where algorithm Eq. (3.1) reduces to gradient descent for vanilla supervised learning. Therefore, vanilla supervised learning can also only achieve $\Theta(1)$ test loss. Clearly, our results imply that FOMAML with train-validation split can significantly outperform vanilla supervised learning when learning our data model.

**Comparison with Bai et al. (2021).** Recently, Bai et al. (2021) studied the importance of the train-validation split in meta-learning under the linear centroid framework. Specifically, they considered linear ridge regression in the inner-loop and linear ridgeless regression in the outer-loop, and showed that the train-train method strictly outperforms the train-validation method in the realizable setting. As we have discussed in Section 3, our data model is linearly realizable, and thus by Bai et al. (2021), the train-train method should outperform the train-validation method when learning our data model with linear ridge/ridgeless regression. On the contrary, Theorems 4.4 and 4.5 demonstrate that the train-validation method still significantly outperforms the train-train method for FOMAML in learning CNNs. We believe the key reason behind this difference is that the CNN model has a much higher expressive power, and therefore can more easily overfit noises in the training data points when using FOMAML. The train-validation split thus greatly helps feature learning of the neural network by performing a sample splitting between the inner and outer-loops. Therefore, our results indicate that train-validation split may have a more significant advantage when learning complicated prediction models.

## 5 EXPERIMENT

### 5.1 EXPERIMENTAL SETUPS

**Synthetic data.** We generate synthetic data to test our theory. For our data generation we choose: $d = 1000$, $K = 343$, $n = 10$, $\sigma_\xi = 10.42$, $\sigma_s = 0.00066$, $\|\boldsymbol{\nu}\|_2 = 1$. For our neural network we choose: $m = 18$, $\sigma_0 = 0.032$. And finally we choose the following parameters for inner and outer level optimization: $\gamma = 0.001$, $J = 5$, $\eta = 0.0001$.

**Real-world Data.** In our experiments, we further justify our theoretical findings in two real-world datasets: RainbowMNIST, miniImagenet, which are discussed as follows.

- **RainbowMNIST.** Following (Yao et al., 2021), RainbowMNIST is a 10-way meta-learning dataset built upon original MNIST dataset, where each task is constructed by applying one combination of image transformations (e.g., coloring, rotation) on the original data. Here, 40 and 16 combinations are used for meta-training and meta-testing, respectively.

- **miniImagenet.** Following the traditional meta-learning setting (Finn & Levine, 2017; Snell et al., 2017), miniImagenet dataset is split into meta-training, meta-validation and meta-testing classes, where 64/16/20 classes are used for meta-training/validation/testing. We adopt the traditional N-way, K-shot setting to split the training and validation set in our experiment, where N=5 and K=1 in this paper (i.e., 5-way, 1-shot learning).

**Backbones and Hyperparameters.** For all real-world datasets, follow (Finn & Levine, 2017; Snell et al., 2017), we adopt the standard four-block convolutional layers as the base learner, where Huberized-ReLU is used as the activation function. The number of inner-loop steps is set as 5. The inner-loop and outer-loop learning rates are set as: 0.01 and 0.001 (miniImagenet), 0.1 and 0.01

(RainbowMNIST), respectively. We report the average accuracy with 95% confidence interval over all meta-testing tasks. For synthetic data, we follow our theoretical analysis and report the cross-entropy loss with 95% confidence interval over all meta-testing tasks. The inner-loop and out-loop learning rates are 0.01, 0.001, respectively.

## 5.2 RESULTS

**Comparison between Different Activation Functions.** We first conduct experiments to compare the performance between Huberized-ReLU activation function used in our theoretical analysis and the traditional ReLU function. The results are reported in Table 1, indicating that Huberized-ReLU performs similarly to ReLU. Thus it is a reasonable replacement to simplify our analysis.

Table 1: Performance comparison between different activation functions.

| Setting | Activation | Synthetic | RainbowMNIST | miniImagenet |
| --- | --- | --- | --- | --- |
| | | Loss $\downarrow$ | Acc $\uparrow$ | Acc $\uparrow$ |
| Train-Train | ReLU | $8.22 \pm 0.42$ | $68.70 \pm 0.51\%$ | $27.19 \pm 1.23\%$ |
| | Huberized-ReLU | $8.50 \pm 0.43$ | $65.32 \pm 0.54\%$ | $25.93 \pm 1.10\%$ |
| Train-Validation | ReLU | $4.62 \pm 0.24$ | $87.68 \pm 0.21\%$ | $46.73 \pm 1.32\%$ |
| | Huberized-ReLU | $4.66 \pm 0.23$ | $87.52 \pm 0.20\%$ | $46.15 \pm 1.36\%$ |

**Performance w.r.t. the Number of Inner-Loop Steps.** We further compare the performance between the train-train and train-validation methods for different inner-loop optimization steps. The results are reported in Table 2. According to the results, we observe that the train-validation split performs much better than the train-train method with more inner-loop steps. This is what we expected since less number of inner-loop steps corresponds to less overfitting to the training set during the inner-loop optimization. In the case of using less inner-loop steps, using the training set in the outer-loop optimization can still contribute to the optimization.

Table 2: Performance comparison of the number of optimization steps in the inner-loop.

| Setting | # of Inner Steps | Synthetic | RainbowMNIST | miniImagenet |
| --- | --- | --- | --- | --- |
| | | Loss $\downarrow$ | Acc $\uparrow$ | Acc $\uparrow$ |
| 1 step | Train-Train | $7.22 \pm 0.37$ | $79.76 \pm 0.41\%$ | $25.09 \pm 1.11\%$ |
| | Train-Validation | $4.81 \pm 0.24$ | $85.83 \pm 0.25\%$ | $25.17 \pm 1.04\%$ |
| 5 steps | Train-Train | $8.50 \pm 0.43$ | $65.32 \pm 0.54\%$ | $25.93 \pm 1.10\%$ |
| | Train-Validation | $4.66 \pm 0.23$ | $87.52 \pm 0.20\%$ | $46.15 \pm 1.36\%$ |

**Performance w.r.t. the Inner-Loop Learning Rate and the Outer-Loop Learning Rate.** Finally, we analyze the influence of inner-loop and outer-loop learning rates. The results in Table 3 show that train-validation split performs better when the inner-loop learning rate is smaller than the outer-loop learning rate, which corroborates our theoretical results. On synthetic data, the train-train method performs better when the outer-loop learning rate larger than the inner-loop learning rate. This is not surprising since smaller inner-loop learning rate corresponds to less overfitting, similar to the phenomenon when using less inner-loop optimization steps (Table 2).

Table 3: Performance w.r.t. the inner-loop learning rate and the outer-loop learning rate.

| Setting | Learning Rate | Synthetic | RainbowMNIST | miniImagenet |
| --- | --- | --- | --- | --- |
| | | Loss $\downarrow$ | Acc $\uparrow$ | Acc $\uparrow$ |
| outer-lr $>$ inner-lr | Train-Train | $5.76 \pm 0.39$ | $64.82 \pm 0.48\%$ | $20.00 \pm 0.00\%$ |
| | Train-Validation | $5.21 \pm 0.31$ | $66.13 \pm 0.31\%$ | $20.00 \pm 0.00\%$ |
| outer-lr $<$ inner-lr | Train-Train | $8.50 \pm 0.43$ | $65.32 \pm 0.54\%$ | $20.00 \pm 0.10\%$ |
| | Train-Validation | $4.66 \pm 0.23$ | $87.52 \pm 0.20\%$ | $46.15 \pm 1.36\%$ |

## 6 OVERVIEW OF PROOF TECHNIQUES

From here onward, we will use upper script $(t, \tau)$ on $\mathbf{w}_{j,r}$ to denote the $t$-th outer-loop iteration and the $\tau$-th inner-loop iteration. However, according to Eq. (3.1) and Eq. (3.2), the inner-loop updates

for a CNN filter $\mathbf{w}_{j,r}$ is task-specific. Therefore we cannot rigorously use the notation $\mathbf{w}_{j,r}^{(t,\tau)}$ without referring to a specific task $k$. Luckily, all our analyses are based on the study of inner products of the form $\langle \mathbf{w}_{j,r}, \mathbf{x}_{k,i}^{(p)} \rangle$. We can use the notation $\langle \mathbf{w}_{j,r}^{(t,\tau)}, \mathbf{x}_{k,i}^{(p)} \rangle$ as the inner product after $t$ outer-loop updates followed by $\tau$ inner-loop updates, where the inner-loop updates only use samples from task $k$. With these notations, we give the following two key lemmas that highlight the main differences between the train-train and train-validation methods.

**Lemma 6.1.** *Suppose one uses the train-train method to train the neural network. Let $T_{k,i}^{(1)}$ be the first iteration such that $\max_{r \in [m]} \langle \mathbf{w}_{j,r}^{(t,0)}, \boldsymbol{\xi}_{k,i} \rangle \geq \Omega\big(1/\mathrm{polylog}(d)\big)$ for $k \in [K]$, $i \in [n]$, and $j = y_{k,i}$. Then, under the same condition as in Theorem 4.4, for any $t \leq T_{k,i}^{(1)}$,*

*1. for any $k' \in [K]$, $r \in [m]$, and $j' \in \{-1, 1\}$ we have*

$$j'\langle \mathbf{w}_{j',r}^{(t,J)}, \boldsymbol{\nu}_{k'} \rangle \leq |\langle \mathbf{w}_{j',r}^{(t,0)}, \boldsymbol{\nu}_{k'} \rangle| \mathcal{O}(1) \gamma^J .$$

*2. for $r = \arg\max_{r' \in [m]} \langle \mathbf{w}_{j,r'}^{(t,0)}, \boldsymbol{\xi}_{k,i} \rangle$,*

$$\langle \mathbf{w}_{j,r}^{(t,J)}, \boldsymbol{\xi}_{k,i} \rangle \geq \langle \mathbf{w}_{j,r}^{(t,0)}, \boldsymbol{\xi}_{k,i} \rangle \Omega(1) \big(\gamma \Theta(\mathrm{polylog}(d))\big)^J .$$

**Lemma 6.2.** *Suppose one uses the train-validation method to train the neural network. Let $T_j^{(2)}$ be the first iteration such that $\max_{r \in [m]} j \langle \mathbf{w}_{j,r}^{(t,0)}, \boldsymbol{\nu} \rangle \geq \Omega(1/\mathrm{polylog}(d))$ for $j \in \{-1, 1\}$. Then, under the same condition as in Theorem 4.5, for any $t \leq T_j^{(2)}$,*

*1. for any $k \in [K]$ and $r = \arg\max_{r' \in [m]} j \langle \mathbf{w}_{j,r'}^{(t,0)}, \boldsymbol{\nu} \rangle$*

$$j\langle \mathbf{w}_{j,r}^{(t,J)}, \boldsymbol{\nu}_k \rangle = \Omega(1) j \langle \mathbf{w}_{j,r}^{(t,0)}, \boldsymbol{\nu}_k \rangle \Theta(1) \gamma^J .$$

*2. for any $r \in [m]$, $k \in [K]$, $i \in \mathcal{I}_k^{\mathrm{val}}$, and $j' \in \{-1, 1\}$*

$$\langle \mathbf{w}_{j',r}^{(t,J)}, \boldsymbol{\xi}_{k,i} \rangle \leq \mathcal{O}\Big( \max\Big\{ \langle \mathbf{w}_{j',r}^{(t,0)}, \boldsymbol{\xi}_{k,i} \rangle, J\widetilde{\mathcal{O}}(d^{-1/2}) \Big\} \Big) .$$

The above two lemmas give some intuitions for the better generalization power of the train-validation method. According to Lemma 6.1, the $J$ inner-loop gradient steps amplify the noise inner products more than the feature inner product for the train-train method. On the contrary, by Lemma 6.2, the $J$ inner-loop gradient steps amplify the feature inner product and do not have a big impact on the noise inner products for the train-validation method.

In the rest of this section, we mainly sketch the proof of Theorem 4.5 for the train-validation method. Based on Lemma 6.2, we can see that the inner-loop updates in the train-validation method prioritizes feature learning over noise memorization. Our further study of the outer-loop training procedure gives the following lemma.

**Lemma 6.3.** *Under the same condition as in Theorem 4.5, let $T^{(2)} = \max_j T_j^{(2)}$, where $T_j^{(2)}$ is defined in Lemma 6.2. Then $\max_{r \in [m]} \langle \mathbf{w}_{j,r}^{(T^{(2)},0)}, \boldsymbol{\xi}_{k,i} \rangle = \widetilde{\mathcal{O}}(d^{-1/2})$ for any $k \in [K]$, $i \in \mathcal{I}_k^{\mathrm{val}}$ and $j \in \{-1, 1\}$.*

We remind our readers that $T^{(2)}$ (defined in Lemma 6.2) represents the time taken for the feature inner product to grow to $\Omega(1/\mathrm{polylog}(d))$. Therefore Lemma 6.3 is essentially still a comparison between the growth rate of feature and noise inner products: it shows that when the feature inner product grows to $\widetilde{\Theta}(1)$, the noise inner products still remain at their initialization order which is $\widetilde{\mathcal{O}}(d^{-1/2})$. The next lemma provides a key result for the convergence of the training loss.

**Lemma 6.4.** *Let $T = \mathrm{poly}(d)$ be the total number of iterations. Under the same condition as in Theorem 4.5, we have*

$$\min_{t \in [T^{(2)}, T-1]} \sum_{(k,i) \in \Psi} -\ell_{k,i}^{\prime(t,J)} \leq \widetilde{\mathcal{O}}\left(\frac{1}{T\eta}\right) ,$$

*where $\ell_{k,i}^{\prime(t,J)} = \ell'(y_{k,i} \cdot f(\widetilde{\mathbf{W}}(\mathbf{W}^{(t)}, \mathcal{S}_k^{\mathrm{tr}}), \mathbf{x}_{k,i}))$.*

By definition, the feature inner product has grown to $\Omega(1/\text{polylog}(d))$ at time $T^{(2)}$. After $T^{(2)}$, we implement a more careful study of the training process, as some data points may have been well-fitted (with a small loss) and no longer significantly contribute to the training. Our next theorem shows that using the train-validation method, after $T^{(2)}$, the feature inner product will grow even larger and will be at least $\Omega(K^c)$ for some $c > 0$ at the end of the training.

**Lemma 6.5.** *Under the same condition as in Theorem 4.5, suppose one runs the train-validation method for a total of $T = \text{poly}(d)$ iterations. Then $\max_{r \in [m]} j\langle \boldsymbol{w}_{j,r}^{(T,0)}, \boldsymbol{\nu}\rangle \geq \Omega(K^c)$ for all $j \in \{-1, 1\}$ and for some $c > 0$.*

Lemma 6.5 shows that at the end of training, the neural network has sufficiently learned the shared feature $\boldsymbol{\nu}$. When given a fresh task, we should expect a small test loss because the shared feature $\boldsymbol{\nu}$ will also be present in this newly sampled task.

We are now ready to present the proof of Theorem 4.5.

*Proof of Theorem 4.5.* By definition, $\widehat{\mathcal{L}}^{\text{tr-val}}(\mathbf{W}^{(t)}, \{\mathcal{S}_k\}_{k=1}^K) = \sum_{k \in [K]} \sum_{i \in \mathcal{I}_k^{\text{val}}} \ell_{k,i}^{(t,J)}$, where $\ell_{k,i}^{(t,J)} = \ell(y_{k,i} \cdot f(\widetilde{\mathbf{W}}(\mathbf{W}^{(t)}, \mathcal{S}_k^{\text{tr}}), \mathbf{x}_{k,i}))$. Then by Lemma 6.4 and the property of the cross-entropy loss that $-\ell'(x) \geq \exp(-x)/2 \geq \ell(x)/2$ for $x > 0$, we have

$$\min_{t \in [T]} \widehat{\mathcal{L}}^{\text{tr-val}}(\mathbf{W}^{(t)}, \{\mathcal{S}_k\}_{k=1}^K) \leq \min_{t \in [T^{(2)}, T-1]} -2 \sum_{k \in [K]} \sum_{i \in \mathcal{I}_k^{\text{val}}} \ell_{k,i}'^{(t,J)} = \widetilde{\mathcal{O}}\left(\frac{1}{T\eta}\right) = \widetilde{\mathcal{O}}\left(\frac{1}{\text{poly}(d)}\right).$$

This proves the first part of Theorem 4.5. For the second part, consider a new data point $(\mathbf{x}, y)$. By Definition 3.1 and Definition 3.2, the new data input $\mathbf{x}$ consists of two patches, one of which is a noise vector $\boldsymbol{\xi}$ that is generated from $\mathcal{N}(\mathbf{0}, \sigma_\xi^2 \cdot (\mathbf{I} - \mathbf{P}))$. Denote by $\mathcal{E}$ the event that $|\langle \boldsymbol{\xi}_{k,i}, \boldsymbol{\xi}\rangle| \leq d^{-1/4}$ and $|\langle \mathbf{w}_{j,r}^{(0,0)}, \boldsymbol{\xi}\rangle| \leq d^{3/2}$ for all $k \in [K]$, $i \in [n]$, $r \in [m]$ and $j \in \{-1, 1\}$, where $\boldsymbol{\xi}$ is the noise vector from the new data $\mathbf{x}$. Then using Gaussian concentration, we have $\mathbb{P}(\mathcal{E}) \geq 1 - \widetilde{\mathcal{O}}(\exp(-d^{1/4}))$. We divide $\mathcal{L}_{\text{test}}(\mathbf{W}^{(T)})$ into two parts:

$$\mathcal{L}_{\text{test}}(\mathbf{W}^{(T)}) = \mathbb{E}\big[\ell\big(yf(\mathbf{W}^{(T)}, \mathbf{x})\big)\big] = \underbrace{\mathbb{E}[\mathbf{1}(\mathcal{E})\ell\big(yf(\mathbf{W}^{(T)}, \mathbf{x})\big)]}_{I_1} + \underbrace{\mathbb{E}[\mathbf{1}(\mathcal{E}^c)\ell\big(yf(\mathbf{W}^{(T)}, \mathbf{x})\big)]}_{I_2}.$$

From this decomposition, it is clear that bounding $I_1$ is more important, since $\mathbb{P}(\mathcal{E}^c)$ is exponentially small, which makes $I_2$ small. Indeed, one can bound $I_2 = \mathcal{O}(\text{poly}(d)) \exp(-0.5d^{1/4})$. Moreover, under the event $\mathcal{E}$, it holds that $F_{-y}(\mathbf{W}_{-y}^{(T)}, \mathbf{x}) < \log(2)$, and $F_y(\mathbf{W}_y^{(T)}, \mathbf{x}) \geq \sigma\big(\max_{r \in [m]} y \cdot \langle \mathbf{w}_{y,r}^{(T,0)}, \boldsymbol{\nu}\rangle\big)$. Applying Lemma 6.5, we have

$$\ell\big(yf(\mathbf{W}^{(T)}, \mathbf{x})\big) = \ell\big(F_y(\mathbf{W}_y^{(T)}, \mathbf{x}) - F_{-y}(\mathbf{W}_{-y}^{(T)}, \mathbf{x})\big) \leq \ell(K^c - \log(2)) \leq 2\exp(-K^c)$$

where the last inequality is by the definition of $\ell(\cdot)$ and the inequality $\log(1+x) \leq x, \forall x \geq 0$. Therefore we have that $I_1 \leq 2\exp(-K^c)$. Combining the bounds on $I_1$ and $I_2$ yields $\mathcal{L}_{\text{test}}(\mathbf{W}^{(T)}) \leq 2\exp(-K^c) + \mathcal{O}(\text{poly}(d)) \exp(-0.5d^{1/4}) = \mathcal{O}\big(\exp(-K^c)\big)$, which proves the theorem. $\square$

## 7 CONCLUSION AND FUTURE WORK

In this work, we study the FOMAML algorithm applied to a classification problem with a two-layer CNN trained by gradient descent. We proved that although both train-train and train-validation methods can achieve a small training loss, to get good generalization results, it is necessary to perform a train-validation split in the data when the noise is large and the number of samples is limited. It is of interest to extend our result to more other types of data model (for example, natural language data), and deeper neural network structures. We would also like to extend our analysis to other popular meta-learning algorithms such as MAML (with hessian term), iMAML (Rajeswaran et al., 2019), Meta-MiniBatchProx (Zhou et al., 2019), Reptile (Nichol et al., 2018) and closed-form solvers (Bertinetto et al., 2018).

ACKNOWLEDGEMENTS

We thank the anonymous reviewers and area chair for their helpful comments. ZC and QG are supported in part by the National Science Foundation IIS-2008981 and the Sloan Research Fellowship.

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

## A    COMPARISON WITH BAI ET AL. (2021)

In this section, we point out some differences between our experiment setup and that of Bai et al. (2021), which also studies the importance of a train-validation split in meta-learning. The main conclusion of Bai et al. (2021) is that the train-train method could outperform the train-validation method asymptotically under a linear centroid model. They have also run experiments using CNN as backbone to support their theory (see Table 1 of Bai et al. (2021)). This may seem to contradict our experiments at first sight, since we showed in Table 1 that the train-validation method should outperform the train-train method using CNN as backbone. We would like to point out that this is due to the different choice of loss function and optimization algorithms used by us and Bai et al. (2021) as explained as follows.

### A.1    REGULARIZER IN INNER-LOOP

Let us focus on the train-train method. Our loss function is given by

$$\widehat{\mathcal{L}}^{\mathrm{tr-tr}}(\mathbf{W}, \{\mathcal{S}_k\}_{k=1}^K) = \frac{1}{K} \sum_{k=1}^K \mathcal{L}(\widetilde{\mathbf{W}}(\mathbf{W}, \mathcal{S}_k), \mathcal{S}_k),$$

where $\widetilde{\mathbf{W}}(\mathbf{W}, \mathcal{S}_k)$ represents the weights of the network after $J$ gradient descent steps (w.r.t. loss $\mathcal{L}(\cdot, \mathcal{S}_k)$) starting from $\mathbf{W}$. While Bai et al. (2021) has their loss function given by

$$\widehat{\mathcal{L}}^{\mathrm{tr-tr}}(\mathbf{W}, \{\mathcal{S}_k\}_{k=1}^K) = \frac{1}{K} \sum_{k=1}^K \mathcal{L}(\widehat{\mathbf{W}}(\mathbf{W}, \mathcal{S}_k), \mathcal{S}_k), \tag{A.1}$$

where

$$\widehat{\mathbf{W}}(\mathbf{W}, \mathcal{S}_k) = \arg\min_{\mathbf{W}'} \mathcal{L}(\mathbf{W}', \mathcal{S}_k) + \lambda \|\mathbf{W}' - \mathbf{W}\|_F^2, \tag{A.2}$$

where $\lambda$ is a regularization parameter. This regularizer is popularized by Rajeswaran et al. (2019) and Zhou et al. (2019). In practice, however, many meta-learning methods do not use this regularizer in the inner-loop (add citations).

### A.2    EXPERIMENT ALGORITHMS

To compare the performance of train-validation and train-train methods, Bai et al. (2021) used iMAML (Rajeswaran et al., 2019) for the train-validation method and Meta-MiniBatchProx (Zhou et al., 2019) for the train-train method. Both algorithms were developed to target the loss function given by Eq. (A.1). In particular, iMAML used implicit function theorem to calculate the gradient $\nabla_{\mathbf{W}}\widehat{\mathbf{W}}(\mathbf{W}, \mathcal{S}_k)$, and then uses this information to do gradient descent on $\mathbf{W}$ with respect to the loss function given by Eq. (A.1). iMAML is similar to MAML in the sense that both algorithms perform gradient descent on $\mathbf{W}$ with respect to their respective loss functions $\widehat{\mathcal{L}}^{\mathrm{tr-tr}}$. However, the structure of Meta-MiniBatchProx is different from that of iMAML and MAML. It is more closely related to Reptile (Nichol et al., 2018), which is another first order meta-learning algorithm. Instead of performing gradient descent in the outer-loop, Reptile updates $\mathbf{W}$ at each step as a convex combination of $\mathbf{W}$ and $\widetilde{\mathbf{W}}(\mathbf{W}, \mathcal{S}_k)$. This gives

$$\mathbf{W} \leftarrow \mathbf{W} + \varepsilon \frac{1}{K} \sum_{k=1}^K \left( \widetilde{\mathbf{W}}(\mathbf{W}, \mathcal{S}_k) - \mathbf{W} \right), \tag{A.3}$$

for some $\varepsilon \in (0, 1)$. And Meta-MiniBatchProx replaces $\widetilde{\mathbf{W}}(\mathbf{W}, \mathcal{S}_k)$ with $\widehat{\mathbf{W}}(\mathbf{W}, \mathcal{S}_k)$ in the above. In our experiments, we used FOMAML for both train-validation and train-train. We believe that by using the same algorithm for train-validation with train-train, the comparison is fairer.

## B    ADDITIONAL EXPERIMENTS

### B.1    COMPARISON WITH REPTILE (NICHOL ET AL., 2018)

We compare FOMAML (using the train-validation method) with Reptile, which is another first-order algorithm used in meta-learning. Different from FOMAML, Reptile does not require a train-validation split in the data set. In addition, the outer-loop of Reptile does not perform gradient

descent, but use a convex combination between the unadapted weights and the task-adapted weights (see Eq. (A.3)). The results are summarized in Table 4. We see that on both RainbowMNIST and miniImagenet, FOMAML with train-validation outperforms Reptile by a small margin.

Table 4: Performance comparison between FOMAML and Reptile.

| Algorithm | RainbowMNIST | miniImagenet |
|---|---|---|
| FOMAML | $87.52 \pm 0.20\%$ | $46.15 \pm 1.36\%$ |
| Reptile | $84.97 \pm 0.26\%$ | $45.19 \pm 1.31\%$ |

### B.2    TIME EVOLUTION OF TEST LOSS OF FOMAML

We plot the time evolution of test loss using trained by FOMAML with train-validation and with train-train on our synthetic data set. The results are illustrated in Figure 1a and 1b for train-validation and train-train, respectively.

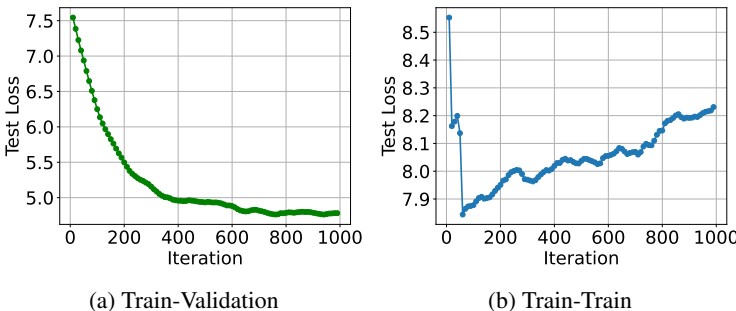

(a) Train-Validation          (b) Train-Train

Figure 1: Comparison of test loss over time of FOMAML trained with train-validation and with train-train. The test loss for the train-validation method decreases almost monotonically, whereas the test loss for the train-train method first decreases and then increases due to overfitting.

### B.3    FOMAML WITH DIFFERENT NEURAL NETWORKS

In this section, we show that FOMAML with train-validaiton still outperforms FOMAML with train-train on CNN with more layers and ResNet. The experiments are run on miniImagenet. The results are recorded in Table 5. We see that FOMAML with train-validation outperforms FOMAML with train-train by a large margin under all 3 neural network structures.

Table 5: Performance w.r.t. different neural networks.

| Setting | CNN 4 layers + normalization | CNN 6 layers + normalization | ResNet18 |
|---|---|---|---|
| Train-train | $25.93 \pm 1.10\%$ | $26.89 \pm 1.18\%$ | $35.63 \pm 1.26\%$ |
| Train-validation | $46.15 \pm 1.36\%$ | $50.89 \pm 1.39\%$ | $55.32 \pm 1.43\%$ |

### B.4    EFFECT OF THE NUMBER OF SAMPLES PER TASK

We then discuss the performance with respect to the number of examples. In Table 6, we report the performance on miniImagenent when the number of examples per class is 1, 3, 5, which corresponds to 5-way 1-shot, 5-way 3-shot and 5-way 5-shot settings. We observe that as the number of samples per class increases, both the train-train and train-validation methods receive a performance increase.

## C    PRELIMINARY CALCULATIONS

In this section, we present some calculations that are useful for our derivations later. Recall that $\ell(z) = \log(1 + \exp(-z))$, which gives $\ell'(z) = -(1 + \exp(z))^{-1}$. Then, we can compute the partial derivative of the loss function using samples from task $k$ as

$$\frac{\partial}{\partial \mathbf{w}_{j,r}} \mathcal{L}(\mathbf{W}, \mathcal{S}_k) = \frac{1}{n} \sum_{i=1}^{n} \frac{\partial}{\partial \mathbf{w}_{j,r}} \ell(y_{k,i} \cdot f(\mathbf{W}, \mathbf{x}_{k,i}))$$

Table 6: Performance comparison w.r.t. the number of samples per class on miniImagenet.

| Number of samples per class | 1 | 3 | 5 |
|---|---|---|---|
| Train-train | $25.93 \pm 1.10\%$ | $36.23 \pm 0.82\%$ | $38.76 \pm 0.66\%$ |
| Train-validation | $46.15 \pm 1.36\%$ | $59.97 \pm 1.02\%$ | $63.18 \pm 0.90\%$ |

$$
= \frac{1}{n} \sum_{i=1}^{n} \ell'(y_{k,i} \cdot f(\mathbf{W}, \mathbf{x}_{k,i})) \frac{\partial}{\partial \mathbf{w}_{j,r}} F_j(\mathbf{W}_j, \mathbf{x}_{k,i}) j
$$

$$
= \frac{1}{n} \sum_{i=1}^{n} \ell'(y_{k,i} \cdot f(\mathbf{W}, \mathbf{x}_{k,i})) y_{k,i} j \left( \sum_{p=1}^{2} \sigma'(\langle \mathbf{w}_{j,r}, \mathbf{x}_{k,i}^{(p)} \rangle) \mathbf{x}_{k,i}^{(p)} \right) .
$$

Similarly, if we only evaluate the loss on the training set of $\mathcal{S}_k$ we have

$$
\frac{\partial}{\partial \mathbf{w}_{j,r}} \mathcal{L}(\mathbf{W}, \mathcal{S}_k^{\mathrm{tr}}) = \frac{1}{n_1} \sum_{i=1}^{n_1} \ell'(y_{k,i} \cdot f(\mathbf{W}, \mathbf{x}_{k,i})) y_{k,i} j \left( \sum_{p=1}^{2} \sigma'(\langle \mathbf{w}_{j,r}, \mathbf{x}_{k,i}^{(p)} \rangle) \mathbf{x}_{k,i}^{(p)} \right) .
$$

Let $t \geq 0$ and $\tau \in [0, J]$ be the indices for the outer-loop and inner-loop respectively. We remind the readers our notation. Denote $\ell_{k,i}'^{(t,J)} = \ell'(y_{k,i} \cdot f(\widetilde{\mathbf{W}}(\mathbf{W}^{(t)}, \mathcal{S}_k), \mathbf{x}_{k,i}))$ when referring to the train-train method and $\ell_{k,i}'^{(t,J)} = \ell'(y_{k,i} \cdot f(\widetilde{\mathbf{W}}(\mathbf{W}^{(t)}, \mathcal{S}_k^{\mathrm{tr}}), \mathbf{x}_{k,i}))$ when referring to the train-validation method.

At outer step $t$, using the samples from task $k$, we have the inner-loop updates as

$$
j\langle \mathbf{w}_{j,r}^{(t,\tau+1)}, \boldsymbol{\nu}_k \rangle = j\langle \mathbf{w}_{j,r}^{(t,\tau)}, \boldsymbol{\nu}_k \rangle - \frac{\gamma}{n} \sum_{i=1}^{n} \ell_{k,i}'^{(t,\tau)} \sigma'(\langle \mathbf{w}_{j,r}^{(t,\tau)}, \boldsymbol{\nu}_k \rangle y_{k,i}) \|\boldsymbol{\nu}_k\|_2^2 . \tag{C.1}
$$

For noise $\boldsymbol{\xi}_{k,i}$, we have the inner-loop updates as (suppose we are using all the samples from task $k$ for the inner-loop optimization)

$$
\langle \mathbf{w}_{j,r}^{(t,\tau+1)}, \boldsymbol{\xi}_{k,i} \rangle = \langle \mathbf{w}_{j,r}^{(t,\tau)}, \boldsymbol{\xi}_{k,i} \rangle - \frac{\gamma}{n} \sum_{i' \neq i} \ell_{k,i'}'^{(t,\tau)} y_{k,i'} j \sigma'(\langle \mathbf{w}_{j,r}^{(t,\tau)}, \boldsymbol{\xi}_{k,i'} \rangle) \langle \boldsymbol{\xi}_{k,i'}, \boldsymbol{\xi}_{k,i} \rangle
$$
$$
- \frac{\gamma}{n} \ell_{k,i}'^{(t,\tau)} y_{k,i} j \sigma'(\langle \mathbf{w}_{j,r}^{(t,\tau)}, \boldsymbol{\xi}_{k,i} \rangle) \|\boldsymbol{\xi}_{k,i}\|_2^2 , \tag{C.2}
$$

where the first sum goes from $i' = 1$ to $i' = n$, omitting the term when $i' = i$. If we consider a noise $\boldsymbol{\xi}_{k,i}$ from the validation set of some task, i.e. $k \in [K]$ and $i \in \mathcal{I}_k^{\mathrm{val}}$, and we only use the training set of task $k$ for inner-loop updates, then

$$
\langle \mathbf{w}_{j,r}^{(t,\tau+1)}, \boldsymbol{\xi}_{k,i} \rangle = \langle \mathbf{w}_{j,r}^{(t,\tau)}, \boldsymbol{\xi}_{k,i} \rangle - \frac{\gamma}{n_1} \sum_{i'=1}^{n_1} \ell_{k,i'}'^{(t,\tau)} y_{k,i'} j \sigma'(\langle \mathbf{w}_{j,r}^{(t,\tau)}, \boldsymbol{\xi}_{k,i'} \rangle) \langle \boldsymbol{\xi}_{k,i'}, \boldsymbol{\xi}_{k,i} \rangle . \tag{C.3}
$$

If we consider a noise $\boldsymbol{\xi}_{k,i}$ from the training set of a task, i.e. $k \in [K]$ and $i \in \mathcal{I}_k^{\mathrm{tr}}$, and we only use the training set of task $k$ for inner-loop updates, then

$$
\langle \mathbf{w}_{j,r}^{(t,\tau+1)}, \boldsymbol{\xi}_{k,i} \rangle = \langle \mathbf{w}_{j,r}^{(t,\tau)}, \boldsymbol{\xi}_{k,i} \rangle - \frac{\gamma}{n_1} \sum_{i' \neq i}^{n_1} \ell_{k,i'}'^{(t,\tau)} y_{k,i'} j \sigma'(\langle \mathbf{w}_{j,r}^{(t,\tau)}, \boldsymbol{\xi}_{k,i'} \rangle) \langle \boldsymbol{\xi}_{k,i'}, \boldsymbol{\xi}_{k,i} \rangle
$$
$$
- \frac{\gamma}{n_1} \ell_{k,i}'^{(t,\tau)} y_{k,i} j \sigma'(\langle \mathbf{w}_{j,r}^{(t,\tau)}, \boldsymbol{\xi}_{k,i} \rangle) \|\boldsymbol{\xi}_{k,i}\|_2^2 , \tag{C.4}
$$

where the sum goes from $i' = 1$ to $i' = n_1$ except when $i' = i$. Using only the training set of task $k$, the feature inner product update is

$$
j\langle \mathbf{w}_{j,r}^{(t,\tau+1)}, \boldsymbol{\nu}_k \rangle = j\langle \mathbf{w}_{j,r}^{(t,\tau)}, \boldsymbol{\nu}_k \rangle - \frac{\gamma}{n_1} \sum_{i=1}^{n_1} \ell_{k,i}'^{(t,\tau)} \sigma'(\langle \mathbf{w}_{j,r}^{(t,\tau)}, \boldsymbol{\nu}_k \rangle y_{k,i}) \|\boldsymbol{\nu}_k\|_2^2 . \tag{C.5}
$$

Next we look at the outer-loop updates. For the train-train method, the noise update is

$$
\langle \mathbf{w}_{j,r}^{(t+1,0)}, \boldsymbol{\xi}_{k,i} \rangle = \langle \mathbf{w}_{j,r}^{(t,0)}, \boldsymbol{\xi}_{k,i} \rangle - \frac{\eta}{Kn} \sum_{(k',i') \neq (k,i)} \ell_{k',i'}'^{(t,J)} y_{k',i'} j \sigma'(\langle \mathbf{w}_{j,r}^{(t,J)}, \boldsymbol{\xi}_{k',i'} \rangle) \langle \boldsymbol{\xi}_{k',i'}, \boldsymbol{\xi}_{k,i} \rangle
$$

$$- \frac{\eta}{Kn} \ell'^{(t,J)}_{k,i} y_{k,i} j \sigma'(\langle \mathbf{w}^{(t,J)}_{j,r}, \boldsymbol{\xi}_{k,i} \rangle) \|\boldsymbol{\xi}_{k,i}\|_2^2 , \tag{C.6}$$

where the sum is over all $k' \in [K]$ and $i' \in [n]$ except when $(k',i') = (k,i)$. Hence, the number of terms in the sum is $Kn - 1$. And

$$\ell'^{(t,\tau)}_{k,i} = \ell'\Big( y_{k,i} \sum_{j=\pm 1} j \sum_{r=1}^m \sigma(\langle \mathbf{w}^{(t,\tau)}_{j,r} \boldsymbol{\xi}_{k,i} \rangle) + \sigma(\langle \mathbf{w}^{(t,\tau)}_{j,r} \boldsymbol{\nu}_k \rangle y_{k,i}) \Big) . \tag{C.7}$$

The arguments in $\sigma'(\cdot)$ in Eq. (C.6) are given by Eq. (C.2). The outer-loop feature update is

$$j\langle \mathbf{w}^{(t+1,0)}_{j,r}, \boldsymbol{\nu} \rangle = j\langle \mathbf{w}^{(t,0)}_{j,r}, \boldsymbol{\nu} \rangle - \frac{\eta}{Kn} \sum_{k,i} \ell'^{(t,J)}_{k,i} \sigma'(\langle \mathbf{w}^{(t,J)}_{j,r}, \boldsymbol{\nu}_k \rangle y_{k,i}) \|\boldsymbol{\nu}\|_2^2 , \tag{C.8}$$

where the arguments in $\sigma'(\cdot)$ are given by Eq. (C.1). The arguments in $\sigma(\cdot)$ in Eq. (C.7)) are given by Eq. (C.2) and Eq. (C.1). For the train-validation method, consider a noise $\boldsymbol{\xi}_{k,i}$ from the validation set of the $k$-th task, i.e. $i \in \mathcal{I}^{\mathrm{val}}_k$. We have

$$\langle \mathbf{w}^{(t+1,0)}_{j,r}, \boldsymbol{\xi}_{k,i} \rangle = \langle \mathbf{w}^{(t,0)}_{j,r}, \boldsymbol{\xi}_{k,i} \rangle - \frac{\eta}{Kn_2} \sum_{i'>n_1, (k',i')\neq(k,i)} \ell'^{(t,J)}_{k',i'} y_{k',i'} j \sigma'(\langle \mathbf{w}^{(t,J)}_{j,r}, \boldsymbol{\xi}_{k',i'} \rangle) \langle \boldsymbol{\xi}_{k',i'}, \boldsymbol{\xi}_{k,i} \rangle$$
$$- \frac{\eta}{Kn_2} \ell'^{(t,J)}_{k,i} y_{k,i} j \sigma'(\langle \mathbf{w}^{(t,J)}_{j,r}, \boldsymbol{\xi}_{k,i} \rangle) \|\boldsymbol{\xi}_{k,i}\|_2^2 , \tag{C.9}$$

where the sum is over all $k' \in [K]$ and $i' \in \mathcal{I}^{\mathrm{val}}_k$ except when $(k',i') = (k,i)$. The arguments in $\sigma'(\cdot)$ are given by Eq. (C.3). If we consider a noise $\boldsymbol{\xi}_{k,i}$ from the training set, i.e. $k \in [K]$ and $i \in \mathcal{I}^{\mathrm{tr}}_k$, then the outer-loop update is given by

$$\langle \mathbf{w}^{(t+1,0)}_{j,r}, \boldsymbol{\xi}_{k,i} \rangle = \langle \mathbf{w}^{(t,0)}_{j,r}, \boldsymbol{\xi}_{k,i} \rangle - \frac{\eta}{Kn_2} \sum_{k,i'>n_1} \ell'^{(t,J)}_{k',i'} y_{k',i'} j \sigma'(\langle \mathbf{w}^{(t,J)}_{j,r}, \boldsymbol{\xi}_{k',i'} \rangle) \langle \boldsymbol{\xi}_{k',i'}, \boldsymbol{\xi}_{k,i} \rangle , \tag{C.10}$$

where the sum is over all $k \in [K]$ and $i \in \mathcal{I}^{\mathrm{val}}_k$. And the arguments in $\sigma'(\cdot)$ are given by Eq. (C.4). The outer-loop update of the feature using the train-validation method is similar to using the train-train method:

$$j\langle \mathbf{w}^{(t+1,0)}_{j,r}, \boldsymbol{\nu} \rangle = j\langle \mathbf{w}^{(t,0)}_{j,r}, \boldsymbol{\nu} \rangle - \frac{\eta}{Kn_2} \sum_{k,i>n_1} \ell'^{(t,J)}_{k,i} \sigma'(\langle \mathbf{w}^{(t,J)}_{j,r}, \boldsymbol{\nu}_k \rangle y_{k,i}) \|\boldsymbol{\nu}\|_2^2 , \tag{C.11}$$

where the arguments in $\sigma'(\cdot)$ in Eq. (C.11) are given by Eq. (C.5). Note that the $\ell'$ term in Eq. (C.9), Eq. (C.10) and Eq. (C.11) are given by Eq. (C.7), but the arguments in $\sigma(\cdot)$ in Eq. (C.7)) are now given by Eq. (C.3) and Eq. (C.5).

## D  PRELIMINARY LEMMAS

We will work under the following parameters for the rest of the proof.

**Condition D.1.** $\|\boldsymbol{\nu}\|_2 = 1$, $\sigma_0 = \frac{1}{\sqrt{d}}$, $\sigma_s = \frac{1}{\sqrt{d} \log(d)^{0.4}}$, $\sigma_\xi = \frac{\log(d)^{0.5}}{\sqrt{d}}$, $n = \Theta(1)$, $K = \log(d)^{0.5}$, $m = \log(d)^{0.2}$, $\eta = \frac{1}{\log(d)^{-5}}$, $\gamma = \log(d)^{0.4}$, $J = 5$, $h = \frac{1}{2}$, $M = d/2$. We will also assume that our data split is symmetric: for each $k \in [K]$, and $j \in \{-1,1\}$

$$\big|\{i : i \in \mathcal{I}^{\mathrm{tr}}_k, y_{k,i} = j\}\big| > 0 , \quad \big|\{i : i \in \mathcal{I}^{\mathrm{val}}_k, y_{k,i} = j\}\big| > 0 .$$

**Lemma D.2.** *Under Condition D.1 and Condition 4.1, the following estimates hold with probability at least $1 - (Kn)^{-10}$.*

1. $\|\boldsymbol{\xi}_{k,i}\|_2^2 = \Theta(\sigma_\xi^2 d)$ *for any $k \in [K]$ and $i \in [n]$.*

2. $\max\big\{|\langle \boldsymbol{\xi}_{k,i}, \boldsymbol{\xi}_{k',i'} \rangle| : k,k' \in [K], i,i' \in [n], (k,i) \neq (k',i')\big\} \leq \mathcal{O}(\sigma_\xi^2 \sqrt{d} \sqrt{\log(Kn)})$.

3. $\max\big\{|\langle \boldsymbol{w}^{(0,0)}_{j,r}, \boldsymbol{\nu} \rangle| : r \in [m]\big\} = \Theta(\sigma_0 \sqrt{\log(mKn)})$ *for any $j \in \{-1,1\}$.*

4. $\max\big\{|\langle \boldsymbol{w}^{(0,0)}_{j,r}, \boldsymbol{\xi}_{k,i} \rangle| : r \in [m]\big\} = \Theta(\sigma_0 \sigma_\xi \sqrt{d} \sqrt{\log(mKn)})$ *for any $j \in \{-1,1\}, k \in [K], i \in [n]$.*

5. $\Omega(\sigma_0) \le \max_{r \in [m]} j \langle \mathbf{w}_{j,r}^{(0,0)}, \boldsymbol{\nu} \rangle \le \mathcal{O}(\sigma_0 \sqrt{\log(mKn)})$ *for any* $j \in \{-1, 1\}$.

6. $\Omega(\sigma_0 \sigma_\xi \sqrt{d}) \le \max_{r \in [m]} \langle \mathbf{w}_{j,r}^{(0,0)}, \boldsymbol{\xi}_{k,i} \rangle \le \mathcal{O}(\sigma_0 \sigma_\xi \sqrt{d} \sqrt{\log(mKn)})$ *for any* $j \in \{-1, 1\}$, $k \in [K]$, $i \in [n]$.

7. $\max\left\{ |\langle \mathbf{w}_{j,r}^{(0,0)}, \mathbf{z}_k \rangle| : r \in [m] \right\} = \Theta(\sigma_0 \sigma_s \sqrt{d} \sqrt{\log(mKn)})$ *for any* $j \in \{-1, 1\}, k \in [K]$.

8. $\Omega(\sigma_0 \sigma_s \sqrt{d}) \le \max_{r \in [m]} j \langle \mathbf{w}_{j,r}^{(0,0)}, \mathbf{z}_k \rangle \le \mathcal{O}(\sigma_0 \sigma_s \sqrt{d} \sqrt{\log(mKn)})$ *for any* $j \in \{-1, 1\}, k \in [M]$.

*Proof.* By Lemma G.1, we have with probability at least $1 - \delta/3$,

$$\frac{1}{2}\sigma_\xi^2 d \le \|\boldsymbol{\xi}_{k,i}\|_2^2 \le \frac{3}{2}\sigma_\xi^2 d,$$

$$|\langle \boldsymbol{\xi}_{k,i}, \boldsymbol{\xi}_{k',i'} \rangle| \le 2\sigma_\xi^2 \sqrt{d} \sqrt{\log\left(12(Kn)^2/\delta\right)},$$

for all $k, k' \in [K]$ and $i, i' \in [n]$ with $(k, i) \ne (k', i')$. By Lemma G.2, we have with probability at least $1 - \delta/3$,

$$|\langle \mathbf{w}_{j,r}^{(0,0)}, \boldsymbol{\nu} \rangle| \le \sqrt{2\log(24m/\delta)}\sigma_0,$$

$$|\langle \mathbf{w}_{j,r}^{(0,0)}, \boldsymbol{\xi}_{k,i} \rangle| \le \sqrt{2\log(24mKn/\delta)}\sigma_0 \sigma_\xi \sqrt{d},$$

for all $r \in [m]$, $j \in \{-1, 1\}$, $k \in [K]$ and $i \in [n]$. Moreover,

$$\frac{\sigma_0}{2} \le \max_{r \in [m]} j \langle \mathbf{w}_{j,r}^{(0,0)}, \boldsymbol{\nu} \rangle \le \sqrt{2\log(24m/\delta)}\sigma_0,$$

$$\frac{\sigma_0 \sigma_\xi \sqrt{d}}{4} \le \max_{r \in [m]} \langle \mathbf{w}_{j,r}^{(0,0)}, \boldsymbol{\xi}_{k,i} \rangle \le \sqrt{2\log(24mKn/\delta)}\sigma_0 \sigma_\xi \sqrt{d},$$

for all $j \in \{-1, 1\}$, $k \in [K]$ and $i \in [n]$. Using similar ideas, we get that with probability at least $1 - \delta/3$,

$$|\langle \mathbf{w}_{j,r}^{(0,0)}, \mathbf{z}_k \rangle| = |\langle \mathbf{w}_{j,r}^{(0,0)}, \mathbf{z}_k / \|\mathbf{z}_k\|_2 \rangle| \|\mathbf{z}_k\|_2 \le \frac{3}{2}\sigma_s \sqrt{d} |\langle \mathbf{w}_{j,r}^{(0,0)}, \mathbf{x} \rangle| \le \frac{3}{2}\sqrt{2\log(12m/\delta)}\sigma_0 \sigma_s \sqrt{d},$$

for any $k \in [K]$, $r \in [m]$, and $j \in \{-1, 1\}$. The second inequality is by Condition 4.1. The last inequality is again by Lemma G.2. Moreover,

$$\frac{\sigma_0 \sigma_s \sqrt{d}}{4} \le \max_{r \in [m]} j \langle \mathbf{w}_{j,r}^{(0,0)}, \mathbf{z}_k \rangle \le \frac{3}{2}\sqrt{2\log(12m/\delta)}\sigma_0 \sigma_s \sqrt{d}.$$

Combining the above results, we get that all of the above events hold simultaneously with probability at least $1 - \delta$. Taking $\delta = (Kn)^{-10}$ gives the desired result.

$\square$

*Remark* D.3. Since we will use the estimates in Lemma D.2 in the rest of our proofs repeatedly, we will not mention the high probability bound in our theorems. It should be understood that our theorems hold with the high probability bound given in Lemma D.2.

*Remark* D.4. Recall that under Condition D.1, the term $\sqrt{\log(mKn)} = \text{polyloglog}(d)$. Its presence/absence will not affect any of our proofs.

**Lemma D.5.** *Let* $\{x_t, y_t\}$ *be two positive sequences updated as*

$$x_{t+1} \ge x_t(1 + A),$$
$$y_{t+1} \le y_t(1 + B),$$

*where* $x_0 = o(1)$, $y_0 = o(1)$, $A = o(1)$, *and* $B = o(1)$. *For any* $D = \mathcal{O}(1)$, *let* $T$ *be the first iteration that* $x_t \ge D$. *We have that* $y_T \le \mathcal{O}(Gy_0)$ *if* $x_0 \ge \mathcal{O}(G^{-\frac{A}{B}})$.

*Proof.* Let us consider the above two sequences $x_t$ and $y_t$ with the inequalities replaced with equality. Note that this will not affect our conclusion. Then we have

$$x_t = x_0(1 + A)^t,$$
$$y_t = y_0(1 + B)^t.$$

And for simplicity, let us also assume that $x_T = D$. Then we have $\frac{D}{x_0} = (1 + A)^T$, which entails

$$T = \frac{\log(D) + \log(\frac{1}{x_0})}{\log(1 + A)}$$
$$= \mathcal{O}\left(\frac{\log(\frac{1}{x_0})}{A}\right). \tag{D.1}$$

Let $T'$ be the first iteration such that $y_{T'} \geq Gy_0$. Again, for simplicity, let us suppose that $y_{T'} = Gy_0$. Using the same procedure, we obtain that

$$T' = \Omega\left(\frac{\log(G)}{B}\right). \tag{D.2}$$

Applying our assumption that $x_0 \geq \mathcal{O}(G^{-\frac{A}{B}})$ to Eq. (D.1) and Eq. (D.2), we conclude that $T < T'$. □

## E  TRAIN-TRAIN METHOD

### E.1  PHASE I

In this section, we will show that under Condition D.1, our neural network will memorize the noise and will not learn the feature. Let $\Xi_{k,i}^{(t)} = \max\left\{\langle \mathbf{w}_{j,r}^{(t,0)}, \boldsymbol{\xi}_{k,i}\rangle : j = y_{k,i}, r \in [m]\right\}$ and $\Lambda_j^{(t)} = \max\left\{j\langle \mathbf{w}_{j,r}^{(t,0)}, \boldsymbol{\nu}\rangle : r \in [m]\right\}$. Let $T_{k,i}^{(1)}$ be the first iteration such that $\Xi_{k,i}^{(1)} \geq m^{-1/2}\left(1 + \frac{\gamma\sigma_\xi^2 d}{n}\right)^{-J} = \widetilde{\Theta}(1)$ and let $T^{(1)} = \max_{k,i} T_{k,i}^{(1)}$ and $(\hat{k}, \hat{i}) = \arg\max_{k,i} T_{k,i}^{(1)}$. So $T^{(1)}$ is the time for the slowest learned noise to have an inner product of size $\widetilde{\Theta}(1)$. And this is witnessed by the noise vector $\boldsymbol{\xi}_{\hat{k},\hat{i}}$.

**Lemma E.1** (Restatement of Lemma 6.1). *Under Condition D.1 and Condition 4.1, if one uses the train-train method, then for any $t \leq T^{(1)}$*

*1. For any $k \in [K]$, $r \in [m]$, and $j \in \{-1, 1\}$ we have*

$$j\langle \mathbf{w}_{j,r}^{(t,J)}, \boldsymbol{\nu}_k\rangle \leq |\langle \mathbf{w}_{j,r}^{(t,0)}, \boldsymbol{\nu}_k\rangle|\,(1 + \gamma\mathcal{O}(1))^J.$$

*2. For $j = y_{\hat{k},\hat{i}}$ and $r = \arg\max_{r' \in [m]}\langle \mathbf{w}_{j,r}^{(t,0)}, \boldsymbol{\xi}_{\hat{k},\hat{i}}\rangle$,*

$$\langle \mathbf{w}_{j,r}^{(t,J)}, \boldsymbol{\xi}_{\hat{k},\hat{i}}\rangle \geq \langle \mathbf{w}_{j,r}^{(t,0)}, \boldsymbol{\xi}_{\hat{k},\hat{i}}\rangle \left(1 + \frac{\gamma}{n}\Omega(1)\Theta(\sigma_\xi^2 d)\right)^J.$$

*Proof.* The proof for both parts relies on the following hypothesis which we will verify inductively later:

$$\max_r j\langle \mathbf{w}_{j,r}^{(t,0)}, \mathbf{z}_k\rangle = o(1)\Lambda_j^{(t)}, \text{for all } k \in [K], j \in \{-1, 1\}. \tag{E.1}$$

$$-\ell_{\hat{k},\hat{i}}'^{(t,\tau)} = \Theta(1), \text{for all } \tau \in [J]. \tag{E.2}$$

$$j\langle \mathbf{w}_{j,r}^{(t,\tau)}, \boldsymbol{\nu}_k\rangle = o(1), \text{for all } \tau \in [J], k \in [K], r \in [m], j \in \{-1, 1\}. \tag{E.3}$$

$$\sigma'(\langle \mathbf{w}_{j,r}^{(t,\tau)}, \boldsymbol{\xi}_{\hat{k},\hat{i}}\rangle) = \Theta(1)\langle \mathbf{w}_{j,r}^{(t,\tau)}, \boldsymbol{\xi}_{\hat{k},\hat{i}}\rangle, \text{for all } \tau \in [J] \text{ if } j = y_{\hat{k},\hat{i}} \text{ and } \langle \mathbf{w}_{j,r}^{(t,\tau)}, \boldsymbol{\xi}_{\hat{k},\hat{i}}\rangle > 0. \tag{E.4}$$

Let us first suppose that the above hypothesis hold. By Eq. (C.1), we know $j\langle \mathbf{w}_{j,r}^{(t,\tau)}, \boldsymbol{\nu}_k\rangle$ is an increasing sequence in $\tau$. Without loss of generality, suppose that $j\langle \mathbf{w}_{j,r}^{(t,0)}, \boldsymbol{\nu}_k\rangle > 0$. Then Hypothesis E.3 implies

$$\sigma'(\langle \mathbf{w}_{j,r}^{(t,\tau)}, \boldsymbol{\nu}_k\rangle j) = 2j\langle \mathbf{w}_{j,r}^{(t,\tau)}, \boldsymbol{\nu}_k\rangle.$$

Then using the fact that $-\ell'(\cdot) \le 1$, we can apply Eq. (C.1) repeatedly to get

$$j\langle \mathbf{w}_{j,r}^{(t,J)}, \boldsymbol{\nu}_k \rangle \le j\langle \mathbf{w}_{j,r}^{(t,0)}, \boldsymbol{\nu}_k \rangle \left(1 + \gamma \mathcal{O}(1)\right)^J, \tag{E.5}$$

where we have also used $\|\boldsymbol{\nu}_k\|_2^2 = \|\boldsymbol{\nu}\|_2^2 + \|\mathbf{z}_k\|_2^2 = 1 + o(1)$. This proves part (1) of the lemma. Let $j = y_{\hat{k},\hat{i}}$. Define $r(t) = \arg\max_{r' \in [m]} \langle \mathbf{w}_{j,r'}^{(t,0)}, \boldsymbol{\xi}_{\hat{k},\hat{i}} \rangle$. By Eq. (C.2), we have

$$
\begin{aligned}
\langle \mathbf{w}_{j,r(t)}^{(t,\tau+1)}, \boldsymbol{\xi}_{\hat{k},\hat{i}} \rangle &\ge \langle \mathbf{w}_{j,r(t)}^{(t,\tau)}, \boldsymbol{\xi}_{\hat{k},\hat{i}} \rangle - \frac{\gamma}{n} \sum_{i \ne \hat{i}} \mathcal{O}(1) |\langle \boldsymbol{\xi}_{\hat{k},i}, \boldsymbol{\xi}_{\hat{k},\hat{i}} \rangle| - \frac{\gamma}{n} \ell'^{(t,\tau)}_{\hat{k},\hat{i}} \sigma'(\langle \mathbf{w}_{j,r(t)}^{(t,\tau)}, \boldsymbol{\xi}_{\hat{k},\hat{i}} \rangle) \left\| \boldsymbol{\xi}_{\hat{k},\hat{i}} \right\|_2^2 \\
&\ge \langle \mathbf{w}_{j,r(t)}^{(t,\tau)}, \boldsymbol{\xi}_{\hat{k},\hat{i}} \rangle - \gamma \mathcal{O}(1) \Theta(\sigma_\xi^2 \sqrt{d} \sqrt{\log(Kn)}) \\
&\quad + \frac{\gamma}{n} \Theta(1) \sigma'(\langle \mathbf{w}_{j,r(t)}^{(t,\tau)}, \boldsymbol{\xi}_{\hat{k},\hat{i}} \rangle) \Theta(\sigma_\xi^2 d),
\end{aligned}
\tag{E.6}
$$

where the first inequality is because $-\ell'(\cdot) \le 1$ and $\sigma'(\cdot) \le 1$. And we have used Lemma D.2 and Hypothesis E.2 to get the second inequality. Let us compare the size of the second and third term on the right hand side of Eq. (E.6). At $(t, \tau) = (0, 0)$, if we leave out $\gamma$ in both terms and consider, the third term is of size $\Theta(\sigma_0 \sigma_\xi^3 d^{3/2}/n) = \Theta(\log(d)^{1.5}/\sqrt{d})$, whereas the second term always has size $\mathcal{O}(\sigma_\xi^2 \sqrt{d} \sqrt{\log(Kn)}) = \mathcal{O}(\log(d)/\sqrt{d}) \mathcal{O}(\text{polyloglog}(d))$. Hence the third term will dominate the second term at $(t, \tau) = (0, 0)$. Note that the size of the second term does not change as $t$ and $\tau$ increase, meaning that if at time $(t \ge 0, \tau \ge 0)$ we have

$$\langle \mathbf{w}_{j,r(t)}^{(t,\tau)}, \boldsymbol{\xi}_{\hat{k},\hat{i}} \rangle = \Omega(1) \boldsymbol{\Xi}_{\hat{k},\hat{i}}^{(0)},$$

then it always holds that the third term dominates the second term on the right hand side of Eq. (E.6), which implies

$$\langle \mathbf{w}_{j,r(t)}^{(t,\tau+1)}, \boldsymbol{\xi}_{\hat{k},\hat{i}} \rangle \ge \langle \mathbf{w}_{j,r(t)}^{(t,\tau)}, \boldsymbol{\xi}_{\hat{k},\hat{i}} \rangle + \frac{\gamma}{n} \Omega(1) \sigma'(\langle \mathbf{w}_{j,r(t)}^{(t,\tau)}, \boldsymbol{\xi}_{\hat{k},\hat{i}} \rangle) \Theta(\sigma_\xi^2 d). \tag{E.7}$$

Applying Eq. (E.7) $J$ times at $t = 0$ and using Hypothesis E.4, we obtain

$$\langle \mathbf{w}_{j,r(0)}^{(0,J)}, \boldsymbol{\xi}_{\hat{k},\hat{i}} \rangle \ge \langle \mathbf{w}_{j,r(0)}^{(0,0)}, \boldsymbol{\xi}_{\hat{k},\hat{i}} \rangle \left(1 + \frac{\gamma}{n} \Omega(1) \Theta(\sigma_\xi^2 d)\right)^J. \tag{E.8}$$

By Eq. (C.6), we have

$$
\begin{aligned}
\langle \mathbf{w}_{j,r(t)}^{(t+1,0)}, \boldsymbol{\xi}_{\hat{k},\hat{i}} \rangle &\ge \langle \mathbf{w}_{j,r(t)}^{(t,0)}, \boldsymbol{\xi}_{\hat{k},\hat{i}} \rangle - \frac{\eta}{Kn} \sum_{(k,i) \ne (\hat{k},\hat{i})} \mathcal{O}(1) |\langle \boldsymbol{\xi}_{k,i}, \boldsymbol{\xi}_{\hat{k},\hat{i}} \rangle| \\
&\quad + \frac{\eta}{Kn} \Theta(1) \sigma'(\langle \mathbf{w}_{j,r(t)}^{(t,J)}, \boldsymbol{\xi}_{\hat{k},\hat{i}} \rangle) \left\| \boldsymbol{\xi}_{\hat{k},\hat{i}} \right\|_2^2 \\
&\ge \langle \mathbf{w}_{j,r(t)}^{(t,0)}, \boldsymbol{\xi}_{\hat{k},\hat{i}} \rangle - \eta \mathcal{O}(1) \Theta(\sigma_\xi^2 \sqrt{d} \sqrt{\log(Kn)}) \\
&\quad + \frac{\eta}{Kn} \Theta(1) \sigma'(\langle \mathbf{w}_{j,r(t)}^{(t,J)}, \boldsymbol{\xi}_{\hat{k},\hat{i}} \rangle) \Theta(\sigma_\xi^2 d),
\end{aligned}
\tag{E.9}
$$

where we have used Hypothesis E.2 and the fact that $-\ell'(\cdot) \le 1$ and $\sigma'(\cdot) \le 1$ to get first inequality. Let us compare the second and third term on the right hand side of Eq. (E.9). At $t = 0$, using Eq. (E.8) we have that

$$\sigma'(\langle \mathbf{w}_{j,r(0)}^{(0,J)}, \boldsymbol{\xi}_{\hat{k},\hat{i}} \rangle) \ge \Theta(1) \langle \mathbf{w}_{j,r(0)}^{(0,0)}, \boldsymbol{\xi}_{\hat{k},\hat{i}} \rangle \left(1 + \frac{\gamma}{n} \Omega(1) \Theta(\sigma_\xi^2 d)\right)^J. \tag{E.10}$$

If we drop $\eta$ in both terms, under Condition D.1 and Condition 4.1, one can calculate that the third term has size $\Theta(\log(d)^8/\sqrt{d})$ whereas the second term has size $\mathcal{O}(\log(d)/\sqrt{d}) \mathcal{O}(\text{polyloglog}(d))$. Thus the third term dominates the second term in Eq. (E.9) at $t = 0$. And we get

$$
\begin{aligned}
\langle \mathbf{w}_{j,r(1)}^{(1,0)}, \boldsymbol{\xi}_{\hat{k},\hat{i}} \rangle &\ge \langle \mathbf{w}_{j,r(0)}^{(1,0)}, \boldsymbol{\xi}_{\hat{k},\hat{i}} \rangle \ge \langle \mathbf{w}_{j,r(0)}^{(0,0)}, \boldsymbol{\xi}_{\hat{k},\hat{i}} \rangle \left(1 + \frac{\eta}{Kn} \left(1 + \frac{\gamma}{n} \Omega(1) \Theta(\sigma_\xi^2 d)\right)^J \Theta(\sigma_\xi^2 d)\right) \\
&= \langle \mathbf{w}_{j,r(0)}^{(0,0)}, \boldsymbol{\xi}_{\hat{k},\hat{i}} \rangle \left(1 + \Omega(1) \frac{\eta \sigma_\xi^2 d}{Kn} \left(\frac{\gamma \sigma_\xi^2 d}{n}\right)^J\right),
\end{aligned}
$$

where the first inequality in the first line is by definition of $r(t)$. We note that the size of the second term on the right hand side of Eq. (E.9) does not change as $t$ and $\tau$ increase. Thus, if $\langle \mathbf{w}_{j,r}^{(t,0)}, \boldsymbol{\xi}_{\hat{k},\hat{i}} \rangle = \Omega(1) \Xi_{\hat{k},\hat{i}}^{(0)}$ for some $t > 0$, Eq. (E.10) implies that $\sigma'(\langle \mathbf{w}_{j,r}^{(t,J)}, \boldsymbol{\xi}_{\hat{k},\hat{i}} \rangle) = \Omega(1)\sigma'(\langle \mathbf{w}_{j,r}^{(0,J)}, \boldsymbol{\xi}_{\hat{k},\hat{i}} \rangle)$. Then we have that the third term will dominate the second term on the right hand side of Eq. (E.9) at $t > 0$, which yields $\langle \mathbf{w}_{j,r(t)}^{(t+1,0)}, \boldsymbol{\xi}_{\hat{k},\hat{i}} \rangle > \langle \mathbf{w}_{j,r(t)}^{(t,0)}, \boldsymbol{\xi}_{\hat{k},\hat{i}} \rangle$ since the third term on the right hand side of Eq. (E.9) is positive. We have shown that

$$\langle \mathbf{w}_{j,r(t+1)}^{(t+1,0)}, \boldsymbol{\xi}_{\hat{k},\hat{i}} \rangle \geq \langle \mathbf{w}_{j,r(t)}^{(t+1,0)}, \boldsymbol{\xi}_{\hat{k},\hat{i}} \rangle > \langle \mathbf{w}_{j,r(t)}^{(t,0)}, \boldsymbol{\xi}_{\hat{k},\hat{i}} \rangle .$$

Then using the same derivation of Eq. (E.8), we obtain

$$\langle \mathbf{w}_{j,r(t)}^{(t,J)}, \boldsymbol{\xi}_{\hat{k},\hat{i}} \rangle \geq \langle \mathbf{w}_{j,r(t)}^{(t,0)}, \boldsymbol{\xi}_{\hat{k},\hat{i}} \rangle \left( 1 + \frac{\gamma}{n} \Omega(1)\Theta(\sigma_\xi^2 d) \right)^J .$$

for any $t \leq T^{(1)}$. This proves the second part of the lemma. $\qquad\square$

We have the following theorem characterizing the size of $\boldsymbol{\Lambda}_j^{T^{(1)}}$.

**Theorem E.2.** *Under Condition D.1 and Condition 4.1, if one uses the train-train method, then for any $t \leq T^{(1)}$*

*1. $\boldsymbol{\Lambda}_j^t = \Theta(1)\boldsymbol{\Lambda}_j^0 = \widetilde{\mathcal{O}}(d^{-1/2})$, for $j \in \{-1, 1\}$.*

*2. For any $k \in [K]$, $i \in [n]$, $j = -y_{k,i}$ and $r \in [m]$, we have*

$$\langle \boldsymbol{w}_{j,r}^{(t,0)}, \boldsymbol{\xi}_{k,i} \rangle \leq \widetilde{\mathcal{O}}(d^{-1/2}) .$$

*Proof.* Recall that the outer-loop for the feature is given by

$$j\langle \mathbf{w}_{j,r}^{(t+1,0)}, \boldsymbol{\nu}_k \rangle = j\langle \mathbf{w}_{j,r}^{(t,0)}, \boldsymbol{\nu}_k \rangle - \frac{\eta}{Kn} \sum_{k',i} \ell_{k',i}'^{(t,J)} \sigma'(\langle \mathbf{w}_{j,r}^{(t,J)}, \boldsymbol{\nu}_{k'} \rangle y_{k',i}) \langle \boldsymbol{\nu}_k, \boldsymbol{\nu}_{k'} \rangle , \qquad (E.11)$$

where the summation is over all $k' \in [K]$ and $i \in [n]$. Since $\langle \boldsymbol{\nu}_k, \boldsymbol{\nu}_{k'} \rangle = \|\boldsymbol{\nu}\|_2 + \langle \mathbf{z}_k, \mathbf{z}_{k'} \rangle \geq 1 - o(1) > 0$, we get that $j\langle \mathbf{w}_{j,r}^{(t,0)}, \boldsymbol{\nu}_k \rangle$ is an increasing sequence in $t$ for any $k \in [K]$ and $r \in [m]$. And Eq. (C.8) shows that $j\langle \mathbf{w}_{j,r}^{(t,0)}, \boldsymbol{\nu} \rangle$ is an increasing sequence in $t$. We have

$$\sigma'(\langle \mathbf{w}_{j,r}^{(t,J)} \boldsymbol{\nu}_k \rangle y_{k,i}) \leq 2j\langle \mathbf{w}_{j,r}^{(t,J)} \boldsymbol{\nu}_k \rangle \leq 2j\langle \mathbf{w}_{j,r}^{(t,0)}, \boldsymbol{\nu}_k \rangle (1 + \gamma\mathcal{O}(1))^J$$
$$= \mathcal{O}(1)j\langle \mathbf{w}_{j,r}^{(t,0)}, \boldsymbol{\nu} \rangle (1 + \gamma\mathcal{O}(1))^J ,$$

where is first inequality is by Hypothesis E.3 and our assumption that $j\langle \mathbf{w}_{j,r}^{(t,0)}, \boldsymbol{\nu}_k \rangle > 0$. The second inequality is by Lemma E.1. And the last equality is due to our Hypothesis E.1. Without loss of generality, consider $j = 1$. Let $r(t) = \arg\max_{r' \in [m]} \langle \mathbf{w}_{1,r'}^{(t,0)}, \boldsymbol{\nu} \rangle$. We can upper bound the growth of the feature in the outer-loop by

$$\boldsymbol{\Lambda}_1^{(t+1)} = \langle \mathbf{w}_{1,r(t+1)}^{(t+1,0)}, \boldsymbol{\nu} \rangle \leq \langle \mathbf{w}_{1,r(t+1)}^{(t,0)}, \boldsymbol{\nu} \rangle \left( 1 + \eta\mathcal{O}(1)\left( 1 + \gamma\Theta(1) \right)^J \right)$$
$$= \langle \mathbf{w}_{1,r(t+1)}^{(t,0)}, \boldsymbol{\nu} \rangle \left( 1 + \mathcal{O}(1)\eta\gamma^J \right)$$
$$\leq \langle \mathbf{w}_{1,r(t)}^{(t,0)}, \boldsymbol{\nu} \rangle \left( 1 + \mathcal{O}(1)\eta\gamma^J \right)$$
$$= \boldsymbol{\Lambda}_1^{(t)} \left( 1 + \mathcal{O}(1)\eta\gamma^J \right) , \qquad (E.12)$$

where we have also used the fact that $-\ell'(\cdot) < 1$ by definition. Note that Eq. (E.12) also holds for $j = -1$, i.e., $\boldsymbol{\Lambda}_{-1}^{(t+1)} \leq \boldsymbol{\Lambda}_{-1}^{(t)} \left( 1 + \mathcal{O}(1)\eta\gamma^J \right)$ using the same derivation. Next, we want to find a lower bound on the learning speed of the noise. After that, we wish to show that the lower bound for the noise actually grows much faster than the upper bound of the memorization of the feature, which implies that noise will be learnt sufficiently well before our neural network can pick up any learning

on the feature. Consider $j = y_{\hat{k},\hat{\imath}}$. Denote $r'(t) = \arg\max_{r \in [m]} \langle \mathbf{w}_{j,r}^{(t,0)}, \boldsymbol{\xi}_{\hat{k},\hat{\imath}} \rangle$. Plugging Lemma E.1 and Hypothesis E.4 into Eq. (E.9) we have the following upper bound hold for all $t \leq T^{(1)}$

$$\langle \mathbf{w}_{j,r'(t+1)}^{(t+1,0)}, \boldsymbol{\xi}_{\hat{k},\hat{\imath}} \rangle \geq \langle \mathbf{w}_{j,r'(t)}^{(t+1,0)}, \boldsymbol{\xi}_{\hat{k},\hat{\imath}} \rangle \geq \langle \mathbf{w}_{j,r'(t)}^{(t,0)}, \boldsymbol{\xi}_{\hat{k},\hat{\imath}} \rangle \Big( 1 + \Omega(1) \frac{\eta \sigma_\xi^2 d}{Kn} \Big( \frac{\gamma \sigma_\xi^2 d}{n} \Big)^J \Big), \qquad \text{(E.13)}$$

where the first inequality is by definition of $r'(t)$. Now we can compare the learning speed of the feature and noise:

$$\boldsymbol{\Lambda}_j^{(t+1)} \leq \boldsymbol{\Lambda}_j^{(t)} \left( 1 + \mathcal{O}(1) \eta \gamma^J \right), \qquad \text{(E.14)}$$

$$\boldsymbol{\Xi}_{\hat{k},\hat{\imath}}^{(t+1)} \geq \boldsymbol{\Xi}_{\hat{k},\hat{\imath}}^{(t)} \Big( 1 + \Omega(1) \frac{\eta \gamma^J}{K} \Big( \frac{\sigma_\xi^2 d}{n} \Big)^{J+1} \Big). \qquad \text{(E.15)}$$

We can apply Lemma D.5 once we check all of its conditions are satisfied. We have $\boldsymbol{\Lambda}_j^{(0)} = \widetilde{\Theta}(d^{-1/2}) = o(1)$, $\boldsymbol{\Xi}_{\hat{k},\hat{\imath}}^{(0)} = \widetilde{\Theta}(d^{-1/2}) = o(1)$. Let $G = 2$, $A = \mathcal{O}(1)\eta\gamma^J$ and $B = \Omega(1) \frac{\eta\gamma^J}{K} \big( \frac{\sigma_\xi^2 d}{n} \big)^{J+1}$, we have $\boldsymbol{\Xi}_{\hat{k},\hat{\imath}}^{(0)} = \widetilde{\Theta}(d^{-1/2}) > G^{-A/B} = 2^{-\log(d)^{5.5}}$. Therefore, by Lemma D.5, for any $t \leq T^{(1)}$, we have $\boldsymbol{\Lambda}_j^{(t)} \leq 2\boldsymbol{\Lambda}_j^{(0)}$. This proves part (1) of our theorem.

We now prove part (2) of our theorem. Let $k \in [K]$, $i \in [n]$, $r \in [m]$ and $j = -y_{k,i}$. By Eq. (C.6)

$$\begin{aligned}
\langle \mathbf{w}_{j,r}^{(t+1,0)}, \boldsymbol{\xi}_{k,i} \rangle &\leq \langle \mathbf{w}_{j,r}^{(t,0)}, \boldsymbol{\xi}_{k,i} \rangle - \frac{\eta}{Kn} \sum_{(k',i') \neq (k,i)} \ell_{k',i'}^{\prime(t,J)} y_{k',i'} j \sigma'(\langle \mathbf{w}_{j,r}^{(t,J)}, \boldsymbol{\xi}_{k',i'} \rangle) \langle \boldsymbol{\xi}_{k',i'}, \boldsymbol{\xi}_{k,i} \rangle \\
&\leq \langle \mathbf{w}_{j,r}^{(t,0)}, \boldsymbol{\xi}_{k,i} \rangle + \frac{\eta}{Kn} \sum_{(k',i') \neq (k,i)} \mathcal{O}(1) |\langle \boldsymbol{\xi}_{k',i'}, \boldsymbol{\xi}_{k,i} \rangle| \\
&\leq \langle \mathbf{w}_{j,r}^{(t,0)}, \boldsymbol{\xi}_{k,i} \rangle + \eta \mathcal{O}(1) \Theta(\sigma_\xi^2 \sqrt{d} \sqrt{\log(Kn)}), \qquad \text{(E.16)}
\end{aligned}$$

where the second inequality is because both $-\ell'(\cdot)$ and $\sigma'(\cdot)$ are bounded by 1. Using Eq. (E.15) and the definition of $T^{(1)}$, we can calculate that $T^{(1)} = \widetilde{\mathcal{O}}(1)$ under Condition D.1. Therefore, for any $t \leq T^{(1)} - 1$ we have

$$\begin{aligned}
\max_{r \in [m]} \langle \mathbf{w}_{j,r}^{(t+1,0)}, \boldsymbol{\xi}_{k,i} \rangle &\leq \max_{r \in [m]} \langle \mathbf{w}_{j,r}^{(0,0)}, \boldsymbol{\xi}_{k,i} \rangle + t\eta \mathcal{O}(1) \Theta(\sigma_\xi^2 \sqrt{d} \sqrt{\log(Kn)}) \\
&\leq \mathcal{O}(\sigma_0 \sigma_\xi \sqrt{d} \sqrt{\log(mKn)}) + T^{(1)} \eta \mathcal{O}(1) \Theta(\sigma_\xi^2 \sqrt{d} \sqrt{\log(Kn)}) \\
&= \widetilde{\mathcal{O}}(d^{-1/2}),
\end{aligned}$$

where we have used Lemma D.2 to get the second inequality. $\qquad \square$

It remains to verify Hypothesis E.1-E.4. Let us suppose that Hypothesis E.1-E.4 hold for all $t < T^{(1)}$. Then we have $\boldsymbol{\Lambda}_j^{(t+1)} \leq 2\boldsymbol{\Lambda}_j^{(0)}$, and $\langle \mathbf{w}_{j,r}^{(t+1,0)}, \boldsymbol{\xi}_{k,i} \rangle \leq \widetilde{\mathcal{O}}(d^{-1/2})$ for any $k \in [K]$, $i \in [n]$, $r \in [m]$ and $j = -y_{k,i}$ by Theorem E.2.

*Proof of Hypothesis E.1.* For each task-specific feature $\mathbf{z}_k$, we have at $t = 0$ using Lemma D.2

$$\Omega(\sigma_0 \sigma_s \sqrt{d}) \leq \max_r j \langle \mathbf{w}_{j,r}^{(0,0)}, \mathbf{z}_k \rangle \leq \mathcal{O}(\sigma_0 \sigma_s \sqrt{d} \sqrt{\log(mKn)}),$$

$$\max_r |\langle \mathbf{w}_{j,r}^{(0,0)}, \boldsymbol{\nu} \rangle| = \Theta(\sigma_0 \sqrt{\log(mKn)}).$$

Since $\sigma_s \sqrt{d} = o(1)$, we see that Hypothesis E.1 holds at $t = 0$. We want to prove that it holds at $t + 1$. For any $k \in [K]$, $r \in [m]$ we have

$$\begin{aligned}
j \langle \mathbf{w}_{j,r}^{(t+1,0)}, \mathbf{z}_k \rangle = {}& j \langle \mathbf{w}_{j,r}^{(t,0)}, \mathbf{z}_k \rangle - \frac{\eta}{Kn} \sum_{k' \neq k} \sum_{i=1}^n \ell_{k',i}^{\prime(t,J)} \sigma'(\langle \mathbf{w}_{j,r}^{(t,J)}, \boldsymbol{\nu}_{k'} \rangle y_{k',i}) \langle \mathbf{z}_{k'}, \mathbf{z}_k \rangle \\
& - \frac{\eta}{Kn} \sum_{i=1}^n \ell_{k,i}^{\prime(t,J)} \sigma'(\langle \mathbf{w}_{j,r}^{(t,J)}, \boldsymbol{\nu}_k \rangle y_{k,i}) \|\mathbf{z}_k\|_2^2
\end{aligned}$$

$$\leq j\langle \mathbf{w}_{j,r}^{(t,0)}, \mathbf{z}_k\rangle - \frac{\eta}{Kn}\sum_{k'\neq k}\sum_{i=1}^{n}\ell_{k',i}'^{(t,J)}\sigma'(\langle\mathbf{w}_{j,r}^{(t,J)},\boldsymbol{\nu}_{k'}\rangle y_{k',i})\Theta(\sigma_s^2\sqrt{d}\sqrt{\log(d)})$$

$$- \frac{\eta}{Kn}\sum_{i=1}^{n}\ell_{k,i}'^{(t,J)}\sigma'(\langle\mathbf{w}_{j,r}^{(t,J)},\boldsymbol{\nu}_{k}\rangle y_{k,i})\Theta(\sigma_s^2 d)$$

$$\leq j\langle \mathbf{w}_{j,r}^{(t,0)}, \mathbf{z}_k\rangle - \frac{\eta}{Kn}\sum_{k'=1}^{K}\sum_{i=1}^{n}\ell_{k',i}'^{(t,J)}\sigma'(\langle\mathbf{w}_{j,r}^{(t,J)},\boldsymbol{\nu}_{k'}\rangle y_{k',i})\Theta(\sigma_s^2 d)\,, \qquad \text{(E.17)}$$

where we used Condition 4.1 to get the first inequality. The last inequality is because $\log(d) \ll d$. Suppose that at time $t+1$, there exists some $\tilde{r}\in[m]$ and $\tilde{k}\in[K]$ such that

$$j\langle\mathbf{w}_{j,\tilde{r}}^{(t+1,0)},\mathbf{z}_{\tilde{k}}\rangle \geq 2\max_r j\langle\mathbf{w}_{j,r}^{(0,0)},\mathbf{z}_{\tilde{k}}\rangle \geq \Omega(\sigma_s\sigma_0\sqrt{d})\,.$$

Then, by Eq. (E.17) we get

$$-\frac{\eta}{Kn}\sum_{t'=0}^{t}\sum_{k,i}\ell_{k,i}'^{(t',J)}\sigma'(\langle\mathbf{w}_{j,\tilde{r}}^{(t',J)},\boldsymbol{\nu}_k\rangle y_{k,i})\Theta(\sigma_s^2 d) \geq 2\max_r j\langle\mathbf{w}_{j,r}^{(0,0)},\mathbf{z}_{\tilde{k}}\rangle - j\langle\mathbf{w}_{j,\tilde{r}}^{(0,0)},\mathbf{z}_{\tilde{k}}\rangle$$

$$\geq \Theta(\sigma_s\sigma_0\sqrt{d})\,,$$

$$-\frac{\eta}{Kn}\sum_{t'=0}^{t}\sum_{k,i}\ell_{k,i}'^{(t,J)}\sigma'(\langle\mathbf{w}_{j,\tilde{r}}^{(t,J)},\boldsymbol{\nu}_k\rangle y_{k,i}) \geq \Theta\left(\frac{\sigma_0}{\sigma_s\sqrt{d}}\right)$$

$$= \Theta\left(\frac{\log(d)^{0.2}}{\sqrt{d}}\right)\,.$$

Here the second inequality is by Lemma D.2. But then, Eq. (C.8) implies

$$j\langle\mathbf{w}_{j,\tilde{r}}^{(t+1,0)},\boldsymbol{\nu}\rangle = j\langle\mathbf{w}_{j,\tilde{r}}^{(0,0)},\boldsymbol{\nu}\rangle - \frac{\eta}{Kn}\sum_{t'=0}^{t}\sum_{k,i}\ell_{k,i}'^{(t,J)}\sigma'(\langle\mathbf{w}_{j,\tilde{r}}^{(t,J)},\boldsymbol{\nu}_k\rangle y_{k,i})$$

$$\geq \min_{r\in[m]}j\langle\mathbf{w}_{j,r}^{(0,0)},\boldsymbol{\nu}\rangle + \Theta\left(\frac{\log(d)^{0.2}}{\sqrt{d}}\right)$$

$$\geq -\Theta(\sigma_0\sqrt{\log(mKn)}) + \Theta\left(\frac{\log(d)^{0.2}}{\sqrt{d}}\right)$$

$$= \Theta(\log(d)^{0.1})\boldsymbol{\Lambda}_j^{(0)}\,.$$

We used Lemma D.2 to get the third line and last line. This is a contradiction, since we should have

$$j\langle\mathbf{w}_{j,\tilde{r}}^{(t+1,0)},\boldsymbol{\nu}\rangle \leq \boldsymbol{\Lambda}_j^{(t+1)} \leq 2\boldsymbol{\Lambda}_j^{(0)}\,.$$

Therefore for all $k\in[K]$ we conclude

$$\max_r j\langle\mathbf{w}_{j,r}^{(t+1,0)},\mathbf{z}_k\rangle \leq 2\max_r j\langle\mathbf{w}_{j,r}^{(0,0)},\mathbf{z}_k\rangle = o(1)\max_r j\langle\mathbf{w}_{j,r}^{(0,0)},\boldsymbol{\nu}\rangle = o(1)\max_r j\langle\mathbf{w}_{j,r}^{(t+1,0)},\boldsymbol{\nu}\rangle\,.$$

This proves that Hypothesis E.1 holds at time $t+1$. $\qquad\square$

*Proof of Hypothesis E.2.* Without loss of generality, let us assume that $y_{\hat{k},\hat{i}} = 1$. By our estimates in Lemma D.2, Hypothesis E.2 clearly holds at initialization. Suppose it holds at time $t$. Recall Eq. (C.7) and that

$$\sum_{j=\pm 1}\sum_{r=1}^{m}j\big(\sigma(\langle\mathbf{w}_{j,r}^{(t+1,\tau)},\boldsymbol{\xi}_{\hat{k},\hat{i}}\rangle)+\sigma(\langle\mathbf{w}_{j,r}^{(t+1,\tau)},\boldsymbol{\nu}_{\hat{k}}\rangle)\big)$$

$$\leq \sum_{r=1}^{m}\sigma(\langle\mathbf{w}_{1,r}^{(t+1,\tau)},\boldsymbol{\xi}_{\hat{k},\hat{i}}\rangle) + \sigma(\langle\mathbf{w}_{1,r}^{(t+1,\tau)},\boldsymbol{\nu}_{\hat{k}}\rangle)\,, \qquad \text{(E.18)}$$

by the non-negativity of $\sigma(\cdot)$. For any $r \in [m]$, we have

$$j\langle \mathbf{w}_{j,r}^{(t+1,0)}, \boldsymbol{\nu}_{\hat{k}} \rangle = j\langle \mathbf{w}_{j,r}^{(t+1,0)}, \mathbf{z}_{\hat{k}} \rangle + j\langle \mathbf{w}_{j,r}^{(t+1,0)}, \boldsymbol{\nu} \rangle$$

$$\leq \underset{r' \in [m]}{\arg\max}\, j\langle \mathbf{w}_{j,r'}^{(t+1,0)}, \mathbf{z}_{\hat{k}} \rangle + \underset{r' \in [m]}{\arg\max}\, j\langle \mathbf{w}_{j,r'}^{(t+1,0)}, \boldsymbol{\nu} \rangle$$

$$= (1 + o(1)) \underset{r' \in [m]}{\arg\max}\, j\langle \mathbf{w}_{j,r'}^{(t+1,0)}, \boldsymbol{\nu} \rangle$$

$$= \mathcal{O}(1)\boldsymbol{\Lambda}_j^{(t+1)}$$

$$\leq 2\boldsymbol{\Lambda}_j^{(0)},$$

where we used Hypothesis E.1 to get the third line. Then, by Eq. (E.5) we have that for any $\tau \in [J]$

$$\langle \mathbf{w}_{1,r}^{(t+1,\tau)}, \boldsymbol{\nu}_{\hat{k}} \rangle \leq \langle \mathbf{w}_{1,r}^{(t+1,0)}, \boldsymbol{\nu}_{\hat{k}} \rangle (1 + \gamma\mathcal{O}(1))^{\tau}$$

$$= \mathcal{O}(1)\boldsymbol{\Lambda}_1^{(t+1)} (1 + \gamma\mathcal{O}(1))^{\tau}$$

$$= \widetilde{\mathcal{O}}(1)\boldsymbol{\Lambda}_1^{(0)}. \tag{E.19}$$

This implies

$$\sum_{r=1}^{m} \sigma(\langle \mathbf{w}_{1,r}^{(t+1,\tau)}, \boldsymbol{\nu}_{\hat{k}} \rangle) \leq \widetilde{\mathcal{O}}(1)m\boldsymbol{\Lambda}_1^{(0)} = o(1).$$

We also have that

$$\sum_{r=1}^{m} \sigma(\langle \mathbf{w}_{1,r}^{(t+1,0)}, \boldsymbol{\xi}_{\hat{k},\hat{i}} \rangle) \leq m\sigma(\boldsymbol{\Xi}_{\hat{k},\hat{i}}^{(t+1)}) = o(1).$$

where the inequality is due to $t + 1 < T^{(1)}$ which implies $\boldsymbol{\Xi}_{\hat{k},\hat{i}}^{(t+1)} < m^{-1/2}\left(1 + \frac{\gamma\sigma_\xi^2 d}{n}\right)^{-J}$. Hence, by Eq. (E.18) we get that $-\ell_{\hat{k},\hat{i}}^{\prime(t+1,0)} = -\ell'(o(1)) = \Omega(1)$. Now, suppose that $-\ell_{\hat{k},\hat{i}}^{\prime(t+1,\tau)} = \Omega(1)$ for some $0 \leq \tau \leq J - 1$. We wish to show that $-\ell_{\hat{k},\hat{i}}^{\prime(t+1,\tau+1)} = \Omega(1)$. By our previous argument, it suffices to show $\sum_{r=1}^{m} \sigma(\langle \mathbf{w}_{1,r}^{(t+1,\tau+1)}, \boldsymbol{\xi}_{\hat{k},\hat{i}} \rangle) = \mathcal{O}(1)$. Because that would imply $-\ell_{\hat{k},\hat{i}}^{\prime(t+1,\tau+1)} = -\ell'(\mathcal{O}(1)) = \Omega(1)$. We have,

$$\sum_{r=1}^{m} \sigma(\langle \mathbf{w}_{1,r}^{(t+1,\tau+1)}, \boldsymbol{\xi}_{\hat{k},\hat{i}} \rangle) \leq m\sigma\left(\boldsymbol{\Xi}_{\hat{k},\hat{i}}^{(t+1)}\left(1 + \frac{\gamma}{n}\mathcal{O}(1)\Theta(\sigma_\xi^2 d)\right)^{\tau+1}\right)$$

$$\leq m\sigma\left(m^{-1/2}\left(1 + \frac{\gamma\sigma_\xi^2 d}{n}\right)^{-J}\left(1 + \frac{\gamma}{n}\mathcal{O}(1)\Theta(\sigma_\xi^2 d)\right)^{J}\right)$$

$$\leq \mathcal{O}(1),$$

where the second inequality is because $\tau + 1 \leq J$. This implies $-\ell_{\hat{k},\hat{i}}^{\prime(t+1,\tau+1)} = -\ell'(\mathcal{O}(1)) = \Omega(1)$ and we are done. $\qquad\square$

*Proof of Hypothesis E.3.* By our estimates in Lemma D.2, Hypothesis E.3 clearly holds at initialization. Suppose it holds at time $t$. Using Eq. (E.19), we immediately have that for any $\tau \in [J]$, $r \in [m]$ and $k \in [K]$,

$$j\langle \mathbf{w}_{j,r}^{(t+1,\tau)}, \boldsymbol{\nu}_k \rangle \leq \widetilde{\mathcal{O}}(1)\boldsymbol{\Lambda}_j^{(0)} = o(1).$$

$\qquad\square$

*Proof of Hypothesis E.4.* By our estimates in Lemma D.2, Hypothesis E.4 clearly holds at initialization. Suppose it holds at time $t$. Suppose that $\langle \mathbf{w}_{j,r}^{(t+1,0)}, \boldsymbol{\xi}_{\hat{k},\hat{i}} \rangle > 0$ for $j = y_{\hat{k},\hat{i}}$ and some $r \in [m]$. Recall that by the assumption of $t + 1 \leq T^{(1)}$, we have

$$\langle \mathbf{w}_{j,r}^{(t+1,0)}, \boldsymbol{\xi}_{\hat{k},\hat{i}} \rangle \leq \boldsymbol{\Xi}_{\hat{k},\hat{i}}^{(t+1)} \leq m^{-1/2}\left(1 + \frac{\gamma\sigma_\xi^2 d}{n}\right)^{-J} = o(1).$$

Hence, by definition of $\sigma'(\cdot)$ we have $\sigma'(\langle \mathbf{w}_{j,r}^{(t+1,0)}, \boldsymbol{\xi}_{\hat{k},\hat{\imath}} \rangle) = 2\langle \mathbf{w}_{j,r}^{(t+1,0)}, \boldsymbol{\xi}_{\hat{k},\hat{\imath}} \rangle$. While $\langle \mathbf{w}_{j,r}^{(t+1,\tau)}, \boldsymbol{\xi}_{\hat{k},\hat{\imath}} \rangle = o(1)$, we have $\sigma'(\langle \mathbf{w}_{j,r}^{(t+1,\tau)}, \boldsymbol{\xi}_{\hat{k},\hat{\imath}} \rangle) = 2\langle \mathbf{w}_{j,r}^{(t+1,\tau)}, \boldsymbol{\xi}_{\hat{k},\hat{\imath}} \rangle$. By Eq. (C.2) we have

$$
\begin{aligned}
\langle \mathbf{w}_{j,r}^{(t+1,\tau+1)}, \boldsymbol{\xi}_{\hat{k},\hat{\imath}} \rangle &\leq \langle \mathbf{w}_{j,r}^{(t+1,\tau)}, \boldsymbol{\xi}_{\hat{k},\hat{\imath}} \rangle + \frac{\gamma}{n} \sum_{i \neq \hat{\imath}} \mathcal{O}(1) |\langle \boldsymbol{\xi}_{\hat{k},i}, \boldsymbol{\xi}_{\hat{k},\hat{\imath}} \rangle| - \frac{\gamma}{n} \ell'^{(t+1,\tau)}_{\hat{k},\hat{\imath}} \sigma'(\langle \mathbf{w}_{j,r}^{(t+1,\tau)}, \boldsymbol{\xi}_{\hat{k},\hat{\imath}} \rangle) \left\| \boldsymbol{\xi}_{\hat{k},\hat{\imath}} \right\|_2^2 \\
&= \langle \mathbf{w}_{j,r}^{(t+1,\tau)}, \boldsymbol{\xi}_{\hat{k},\hat{\imath}} \rangle + \gamma \mathcal{O}(1) \mathcal{O}(\sigma_\xi^2 \sqrt{d} \sqrt{\log(Kn)}) + \frac{\gamma}{n} \mathcal{O}(1) \sigma'(\langle \mathbf{w}_{j,r}^{(t+1,\tau)}, \boldsymbol{\xi}_{\hat{k},\hat{\imath}} \rangle) \Theta(\sigma_\xi^2 d) \\
&= \langle \mathbf{w}_{j,r}^{(t+1,\tau)}, \boldsymbol{\xi}_{\hat{k},\hat{\imath}} \rangle \left( 1 + \frac{\gamma}{n} \mathcal{O}(1) \Theta(\sigma_\xi^2 d) \right) + \widetilde{\mathcal{O}}(d^{-1/2}) \\
&\leq \langle \mathbf{w}_{j,r}^{(t+1,0)}, \boldsymbol{\xi}_{\hat{k},\hat{\imath}} \rangle \left( 1 + \frac{\gamma}{n} \mathcal{O}(1) \Theta(\sigma_\xi^2 d) \right)^{\tau+1} + \widetilde{\mathcal{O}}(d^{-1/2}) \tau \left( 1 + \frac{\gamma}{n} \mathcal{O}(1) \Theta(\sigma_\xi^2 d) \right)^{\tau} \\
&= \langle \mathbf{w}_{j,r}^{(t+1,0)}, \boldsymbol{\xi}_{\hat{k},\hat{\imath}} \rangle \left( 1 + \frac{\gamma}{n} \mathcal{O}(1) \Theta(\sigma_\xi^2 d) \right)^{\tau+1} + \widetilde{\mathcal{O}}(d^{-1/2}) \\
&\leq \Xi_{\hat{k},\hat{\imath}}^{(t+1)} \left( 1 + \frac{\gamma}{n} \mathcal{O}(1) \Theta(\sigma_\xi^2 d) \right)^{\tau+1} + \widetilde{\mathcal{O}}(d^{-1/2}) .
\end{aligned}
\tag{E.20}
$$

We used $-\ell'(\cdot) \leq 1$ and $\sigma'(\cdot) \leq 1$ to get the first line. We used Lemma D.2 for the second line. It is straightforward to check recursively that $\langle \mathbf{w}_{j,r}^{(t+1,\tau)}, \boldsymbol{\xi}_{\hat{k},\hat{\imath}} \rangle \leq o(1)$ for $\tau \in [J-1]$ and hence Eq. (E.20) holds for $\tau \in [J-1]$. In the meantime, this also implies $\sigma'(\langle \mathbf{w}_{j,r}^{(t+1,\tau)}, \boldsymbol{\xi}_{\hat{k},\hat{\imath}} \rangle) = 2\langle \mathbf{w}_{j,r}^{(t+1,\tau)}, \boldsymbol{\xi}_{\hat{k},\hat{\imath}} \rangle$ for $\tau \in [J-1]$. Then Eq. (E.20) implies

$$
\begin{aligned}
\langle \mathbf{w}_{j,r}^{(t+1,J)}, \boldsymbol{\xi}_{\hat{k},\hat{\imath}} \rangle &\leq \Xi_{\hat{k},\hat{\imath}}^{(t+1)} \left( 1 + \frac{\gamma}{n} \mathcal{O}(1) \Theta(\sigma_\xi^2 d) \right)^J + \widetilde{\mathcal{O}}(d^{-1/2}) \\
&\leq \mathcal{O}(m^{-1/2}) = o(1) .
\end{aligned}
$$

Therefore, $\sigma'(\langle \mathbf{w}_{j,r}^{(t+1,J)}, \boldsymbol{\xi}_{\hat{k},\hat{\imath}} \rangle) = 2\langle \mathbf{w}_{j,r}^{(t+1,J)}, \boldsymbol{\xi}_{\hat{k},\hat{\imath}} \rangle$. □

We note that by our definition, $T^{(1)}$ is the time taken for the slowest-learnt noise to be memorized by our neural network. Other noises are learned faster. Our next theorem guarantees that other noises has almost the same inner product as the slowest-learnt noise at time $T^{(1)}$.

**Theorem E.3.** *Under Condition D.1 and Condition 4.1, suppose one uses the train-train method. For any $k \in [K]$, $i \in [n]$ with $(k,i) \neq (\hat{k}, \hat{\imath})$, let $t \in [T_{k,i}^{(1)}, T^{(1)}]$, then we have $\Xi_{k,i}^{(t)} = \widetilde{\Theta}(1)$. More precisely, it holds that*

$$
\Xi_{k,i}^{(t)} \geq \Omega \left( m^{-1/2} \left( 1 + \frac{\gamma \sigma_\xi^2 d}{n} \right)^{-J} \right) .
$$

*And*

$$
\Xi_{k,i}^{(t)} \leq \mathcal{O} \left( m^{-1/2} \left( 1 + \frac{\gamma \sigma_\xi^2 d}{n} \right)^{-J+0.5} \right) .
$$

*Proof.* We first show that $\Xi_{k,i}^{(t)} \geq \Omega \left( m^{-1/2} \left( 1 + \frac{\gamma \sigma_\xi^2 d}{n} \right)^{-J} \right) = \widetilde{\Omega}(1)$. By definition of $T_{k,i}^{(1)}$, we know that $\Xi_{k,i}^{(t)} \geq m^{-1/2} \left( 1 + \frac{\gamma \sigma_\xi^2 d}{n} \right)^{-J}$ when $t = T_{k,i}^{(1)}$. Suppose there exists $t \in [T_{k,i}^{(1)}, T^{(1)}]$ such that it is the first iteration such that $\Xi_{k,i}^{(t)} < m^{-1/2} \left( 1 + \frac{\gamma \sigma_\xi^2 d}{n} \right)^{-J}$. If such $t$ does not exist, then we automatically have $\Xi_{k,i}^{(t)} \geq m^{-1/2} \left( 1 + \frac{\gamma \sigma_\xi^2 d}{n} \right)^{-J}$ for all $t \in [T_{k,i}^{(1)}, T^{(1)}]$. By definition of $t$ we have $\Xi_{k,i}^{(t-1)} \geq m^{-1/2} \left( 1 + \frac{\gamma \sigma_\xi^2 d}{n} \right)^{-J}$. By Eq. (E.9), we have the following lower bound on $\Xi_{k,i}^{(t)}$

$$
\begin{aligned}
\Xi_{k,i}^{(t)} &\geq \Xi_{k,i}^{(t-1)} - \eta \mathcal{O}(1) \Theta(\sigma_\xi^2 \sqrt{d} \sqrt{\log(Kn)}) \\
&\geq m^{-1/2} \left( 1 + \frac{\gamma \sigma_\xi^2 d}{n} \right)^{-J} - \widetilde{\mathcal{O}}(d^{-1/2}) \\
&= \Omega \left( m^{-1/2} \left( 1 + \frac{\gamma \sigma_\xi^2 d}{n} \right)^{-J} \right) ,
\end{aligned}
$$

where in the first inequality we used $-\ell'(\cdot) \leq 1$ and $\sigma'(\cdot) \leq 1$ and Lemma D.2. The second and the third line are direct calculations under Condition D.1. Using the same proof as Hypothesis E.2, we have that $-\ell_{k,i}^{\prime(t,\tau)} = \Omega(1)$ for any $\tau \in [J]$. Using a similar derivation of Eq. (E.13), we found that

$$\Xi_{k,i}^{(t+1)} \geq \Xi_{k,i}^{(t)}\Big(1 + \Omega(1)\frac{\eta\sigma_\xi^2 d}{Kn}\Big(\frac{\gamma\sigma_\xi^2 d}{n}\Big)^J\Big)$$

$$\geq \big(\Xi_{k,i}^{(t-1)} - \widetilde{\mathcal{O}}(d^{-1/2})\big)\Big(1 + \Omega(1)\frac{\eta\sigma_\xi^2 d}{Kn}\Big(\frac{\gamma\sigma_\xi^2 d}{n}\Big)^J\Big)$$

$$> \Xi_{k,i}^{(t-1)},$$

where the last inequality is because $\Xi_{k,i}^{(t-1)} \geq m^{-1/2}\big(1 + \frac{\gamma\sigma_\xi^2 d}{n}\big)^{-J} = \widetilde{\Omega}(1) \gg \widetilde{\mathcal{O}}(d^{-1/2})$, and $\big(1 + \Omega(1)\frac{\eta\sigma_\xi^2 d}{Kn}\big(\frac{\gamma\sigma_\xi^2 d}{n}\big)^J\big) = \Omega(\mathrm{polylog}(d)) \gg \mathcal{O}(1)$. This shows that even if $\Xi_{k,i}^{(t)} < \Xi_{k,i}^{(t-1)}$ at some $t$, we still have $\Xi_{k,i}^{(t)} = \Omega\big(m^{-1/2}\big(1 + \frac{\gamma\sigma_\xi^2 d}{n}\big)^{-J}\big) = \widetilde{\Omega}(1)$ and $\Xi_{k,i}^{(t+1)} > \Xi_{k,i}^{(t-1)}$. Hence, $\Xi_{k,i}^{(t)} \geq \widetilde{\Omega}(1)$ for all $t \in [T_{k,i}^{(1)}, T^{(1)}]$. Next, we need to show that $\Xi_{k,i}^{(t)} \leq \mathcal{O}\big(m^{-1/2}\big(1 + \frac{\gamma\sigma_\xi^2 d}{n}\big)^{-J+0.5}\big)$.

Let

$$t' = \inf\Big\{t \in (T_{k,i}^{(1)}, T^{(1)}] : \Xi_{k,i}^{(\tilde{t})} \geq m^{-1/2}\big(1 + \frac{\gamma\sigma_\xi^2 d}{n}\big)^{-J+0.5}, \forall \tilde{t} \in [t, T^{(1)}]\Big\},$$

where the inf of an empty set is defined as $\infty$. If $t' = \infty$ then we are done, because that implies $\Xi_{k,i}^{(T^{(1)})} < m^{-1/2}\big(1 + \frac{\gamma\sigma_\xi^2 d}{n}\big)^{-J+0.5}$. Assume $t'$ is finite. We first need to show that $\Xi_{k,i}^{(t')} = \mathcal{O}\big(m^{-1/2}\big(1 + \frac{\gamma\sigma_\xi^2 d}{n}\big)^{-J+0.5}\big)$. By definition of $t'$, we know $\Xi_{k,i}^{(t'-1)} < m^{-1/2}\big(1 + \frac{\gamma\sigma_\xi^2 d}{n}\big)^{-J+0.5}$. By Eq. (C.6), we have

$$\Xi_{k,i}^{(t')} \leq \Xi_{k,i}^{(t'-1)} + \widetilde{\mathcal{O}}(d^{-1/2}) + \frac{\eta}{Kn}\Theta(\sigma_\xi^2 d)$$

$$< m^{-1/2}\Big(1 + \frac{\gamma\sigma_\xi^2 d}{n}\Big)^{-J+0.5} + \widetilde{\mathcal{O}}(d^{-1/2}) + \Theta\Big(\frac{\eta\sigma_\xi^2 d}{Kn}\Big)$$

$$\leq 1.5\Big(m^{-1/2}\Big(1 + \frac{\gamma\sigma_\xi^2 d}{n}\Big)^{-J+0.5}\Big),$$

where we used $\ell'(\cdot) \leq 1$, $\sigma'(\cdot) \leq 1$ and Lemma D.2 in the first inequality. The last inequality is because $\widetilde{\mathcal{O}}(d^{-1/2}) + \Theta\big(\frac{\eta\sigma_\xi^2 d}{Kn}\big) = o(1)m^{-1/2}\big(1 + \frac{\gamma\sigma_\xi^2 d}{n}\big)^{-J+0.5}$ by a direct calculation. We now show that $-\ell_{k,i}^{\prime(t,J)} = \mathcal{O}(\exp(-\log(d)^{0.6}))$ for all $t \in [t', T^{(1)}]$. Without loss of generality, assume $y_{k,i} = 1$. Let $r(t) = \arg\max_{r \in [m]}\langle \mathbf{w}_{1,r}^{(t,0)}, \boldsymbol{\xi}_{k,i}\rangle$. Using similar proof of Hypothesis E.2, we have that $-\ell_{k,i}^{\prime(t,\tau)} = \Theta(1)$ for $\tau \in [J-1]$. Since $\langle \mathbf{w}_{1,r(t)}^{(t,0)}, \boldsymbol{\xi}_{k,i}\rangle \geq m^{-1/2}\big(1 + \frac{\gamma\sigma_\xi^2 d}{n}\big)^{-J+0.5}$, we apply Eq. (E.7) $J$ times to get $\langle \mathbf{w}_{1,r(t)}^{(t,J)}, \boldsymbol{\xi}_{k,i}\rangle \geq \Theta\big(m^{-1/2}\big(1 + \frac{\gamma\sigma_\xi^2 d}{n}\big)^{0.5}\big)$. Then we have

$$-\ell_{k,i}^{\prime(t,J)} = -\ell'\Bigg(\sum_{j=\pm 1}\sum_{r'=1}^m j\Big(\sigma(\langle \mathbf{w}_{j,r'}^{(t,J)}, \boldsymbol{\xi}_{k,i}\rangle) + \sigma(\langle \mathbf{w}_{j,r'}^{(t,J)}, \boldsymbol{\nu}_k\rangle)\Big)\Bigg)$$

$$\leq -\ell'\Bigg(\sigma(\langle \mathbf{w}_{1,r(t)}^{(t,J)}, \boldsymbol{\xi}_{k,i}\rangle) - \sum_{r'=1}^m \Big(\sigma(\langle \mathbf{w}_{-1,r'}^{(t,J)}, \boldsymbol{\xi}_{k,i}\rangle) + \sigma(\langle \mathbf{w}_{-1,r'}^{(t,J)}, \boldsymbol{\nu}_k\rangle)\Big)\Bigg)$$

$$\leq -\ell'\Big(\Theta\Big(m^{-1/2}\Big(1 + \frac{\gamma\sigma_\xi^2 d}{n}\Big)^{0.5}\Big) - o(1) - o(1)\Big)$$

$$\leq \mathcal{O}\big(\exp\big(-\Theta(1)\log(d)^{0.6}\big)\big). \tag{E.21}$$

The second line is because $-\ell'(\cdot)$ is non-increasing. To justify the third line, we first note that the second part of Theorem E.2 implies $\langle \mathbf{w}_{-1,r'}^{(t,0)}, \boldsymbol{\xi}_{k,i}\rangle = \widetilde{\mathcal{O}}(d^{-1/2})$ for all $r' \in [m]$. And

$$\langle \mathbf{w}_{-1,r'}^{(t,\tau+1)}, \boldsymbol{\xi}_{k,i}\rangle = \langle \mathbf{w}_{-1,r'}^{(t,\tau)}, \boldsymbol{\xi}_{k,i}\rangle + \frac{\gamma}{n}\sum_{i'\neq i}\ell_{k,i'}^{\prime(t,\tau)}y_{k,i'}\sigma'(\langle \mathbf{w}_{-1,r'}^{(t,\tau)}, \boldsymbol{\xi}_{k,i'}\rangle)\langle \boldsymbol{\xi}_{k,i'}, \boldsymbol{\xi}_{k,i}\rangle$$

$$+ \frac{\gamma}{n} \ell_{k,i}^{\prime(t,\tau)} \sigma^{\prime}(\langle \mathbf{w}_{-1,r'}^{(t,\tau)}, \boldsymbol{\xi}_{k,i} \rangle) \| \boldsymbol{\xi}_{k,i} \|_2^2$$

$$\le \langle \mathbf{w}_{-1,r'}^{(t,\tau)}, \boldsymbol{\xi}_{k,i} \rangle + \frac{\gamma}{n} \sum_{i' \neq i} \mathcal{O}(1) |\langle \boldsymbol{\xi}_{k,i'}, \boldsymbol{\xi}_{k,i} \rangle|$$

$$\le \langle \mathbf{w}_{-1,r'}^{(t,\tau)}, \boldsymbol{\xi}_{k,i} \rangle + \widetilde{\mathcal{O}}(d^{-1/2}),$$

where we used $-\ell'(\cdot) \le 1$, $\sigma'(\cdot) \le 1$ and Lemma D.2. This implies

$$\langle \mathbf{w}_{-1,r'}^{(t,J)}, \boldsymbol{\xi}_{k,i} \rangle \le \langle \mathbf{w}_{-1,r'}^{(t,0)}, \boldsymbol{\xi}_{k,i} \rangle + J \widetilde{\mathcal{O}}(d^{-1/2}) = \widetilde{\mathcal{O}}(d^{-1/2}),$$

since $J = \mathcal{O}(1)$ and $\langle \mathbf{w}_{-1,r'}^{(t,0)}, \boldsymbol{\xi}_{k,i} \rangle = \widetilde{\mathcal{O}}(d^{-1/2})$. Hence,

$$\sum_{r'=1}^{m} \sigma(\langle \mathbf{w}_{-1,r'}^{(t,J)}, \boldsymbol{\xi}_{k,i} \rangle) \le m \widetilde{\mathcal{O}}(d^{-1/2}) = o(1).$$

By Eq. (C.1) and Eq. (E.11) we get that

$$\langle \mathbf{w}_{-1,r}^{(t,J)}, \boldsymbol{\nu}_k \rangle \le \langle \mathbf{w}_{-1,r}^{(t,0)}, \boldsymbol{\nu}_k \rangle \le \langle \mathbf{w}_{-1,r}^{(0,0)}, \boldsymbol{\nu}_k \rangle = \mathcal{O}(d^{-1/2}\sqrt{\log(m)}),$$

where the last equality is due to Lemma D.2. Thus,

$$\sum_{r=1}^{m} \sigma(\langle \mathbf{w}_{-1,r}^{(t,J)}, \boldsymbol{\nu}_k \rangle) \le m \widetilde{\mathcal{O}}(d^{-1/2}\sqrt{\log(m)}) = o(1).$$

This justifies the third line of Eq. (E.21). Using Eq. (E.15), one can directly calculate that $T^{(1)} = \mathcal{O}(\mathrm{polylog}(d))$. Then, using Eq. (C.6) we have

$$\boldsymbol{\Xi}_{k,i}^{(t+1)} \le \boldsymbol{\Xi}_{k,i}^{(t)} + \widetilde{\mathcal{O}}(d^{-1/2}) - \frac{\eta}{Kn} \ell_{k,i}^{\prime(t,J)} \Theta(\sigma_\xi^2 d)$$

$$\le \boldsymbol{\Xi}_{k,i}^{(t)} + \widetilde{\mathcal{O}}(d^{-1/2}) + \frac{\eta}{Kn} \Theta(\sigma_\xi^2 d) \mathcal{O}\big( \exp\big( -\log(d)^{0.6} \big) \big),$$

$$\boldsymbol{\Xi}_{k,i}^{(T^{(1)})} \le \boldsymbol{\Xi}_{k,i}^{(t')} + (T^{(1)} - t')\big(\widetilde{\mathcal{O}}(d^{-1/2}) + \frac{\eta}{Kn} \Theta(\sigma_\xi^2 d) \mathcal{O}\big( \exp\big( -\log(d)^{0.6} \big)\big)\big)$$

$$\le \boldsymbol{\Xi}_{k,i}^{(t')} + T^{(1)}\big(\widetilde{\mathcal{O}}(d^{-1/2}) + \frac{\eta}{Kn} \Theta(\sigma_\xi^2 d) \mathcal{O}\big( \exp\big( -\log(d)^{0.6} \big)\big)\big)$$

$$= \boldsymbol{\Xi}_{k,i}^{(t')} + \widetilde{\mathcal{O}}(d^{-1/2}) + \widetilde{\Theta}(1) \mathcal{O}\big( \exp\big( -\log(d)^{0.6} \big)\big)$$

$$\le 2\boldsymbol{\Xi}_{k,i}^{(t')} \le 3\Big(m^{-1/2}\Big(1 + \frac{\gamma \sigma_\xi^2 d}{n}\Big)^{-J+0.5}\Big).$$

In the second last line, we can absorb $T^{(1)}$ into $\widetilde{\mathcal{O}}(d^{-1/2})$. And $T^{(1)} \frac{\eta}{Kn} \Theta(\sigma_\xi^2 d) = \widetilde{\Theta}(1)$. To get the last line, recall that $\boldsymbol{\Xi}_{k,i}^{(t')} \ge \big(m^{-1/2}\big(1 + \frac{\gamma \sigma_\xi^2 d}{n}\big)^{-J+0.5}\big) = \Omega(1/\mathrm{polylog}(d))$. And $\widetilde{\Theta}(1)\mathcal{O}\big( \exp\big( -\log(d)^{0.6} \big)\big) = o(1/\mathrm{polylog}(d))$. This concludes our proof. $\square$

In short, we have shown that in Phase I, the feature has not been learnt well by any neurons, which is characterized by the inequality $\boldsymbol{\Lambda}_j^{(T^{(1)})} \le \mathcal{O}(d^{-1/2})$ for $j \in \{-1, 1\}$. And for each noise vector $\boldsymbol{\xi}_{k,i}$, there exists weights $\mathbf{w}_{j,r}^{(T^{(1)},0)}$ with $j = y_{k,i}$ such that their inner product is large: $\langle \mathbf{w}_{j,r}^{(T^{(1)},0)}, \boldsymbol{\xi}_{k,i} \rangle = \widetilde{\Theta}(1)$.

### E.2 PHASE II

We have already proven that at the end of Phase I, all of the noises are memorized by the network with an inner product of size $\widetilde{\Theta}(1)$ whereas the feature is barely memorized, with an inner product of size $\widetilde{\mathcal{O}}(d^{-1/2})$. After Phase I, there is no guarantee that $-\ell_{k,i}^{\prime(t,J)} = \Omega(1)$. For this reason, the learning of both the feature and noises will slow down. The difficulty in analysis at this stage is that we no longer have $-\ell_{k,i}^{\prime(t,J)} = \Theta(1)$. The value of $\ell_{k,i}^{\prime(t,J)}$ is getting smaller as the inner products

$\langle \mathbf{w}_{j,r}^{(t,J)}, \boldsymbol{\xi}_{k,i} \rangle$ gets larger. Moreover, if we examine Eq. (C.6), we observe that the second term on the right hand side could have a non-trivial effect on the growth of the noise inner product, if the third term on the right hand side becomes exponentially smaller than the second term because of $\ell_{k,i}^{\prime(t,J)}$. But the sign of each of the summand in the second term is undetermined, which makes the analysis even harder. We first make some claims that will help prove our theorems. And we will verify these claims at the end of this section. Let $T = \mathrm{poly}(d)$ be the total run time. We have the following claims under Condition D.1 and Condition 4.1.

**Claim E.4.** *For any $k \in [K]$, $i \in [n]$, $r \in [m]$, and $t \in [T^{(1)}, T]$, let $j = -y_{k,i}$, then*

$$\langle \boldsymbol{w}_{j,r}^{(t,0)}, \boldsymbol{\xi}_{k,i} \rangle = \widetilde{\mathcal{O}}(d^{-1/2}).$$

*And*

$$\langle \boldsymbol{w}_{j,r}^{(t,J)}, \boldsymbol{\xi}_{k,i} \rangle = \widetilde{\mathcal{O}}(d^{-1/2}).$$

**Claim E.5.** *For any $j \in \{-1, 1\}$ and $t \in [T^{(1)}, T]$ we have*

$$\boldsymbol{\Lambda}_j^{(t)} \le \mathcal{O}(d^{-1/4}).$$

**Claim E.6.** *For any $j \in \{-1, 1\}$, $k \in [K]$, $r \in [m]$ and $t \in [T^{(1)}, T]$ we have*

$$\max_r j \langle \boldsymbol{w}_{j,r}^{(t,0)}, \mathbf{z}_k \rangle = o(1) \boldsymbol{\Lambda}_j^{(t)}.$$

**Lemma E.7.** *Let $t \in [T^{(1)}, T]$, $k \in [K]$, $i \in [n]$, $j = y_{k,i}$ and $r = \arg\max_{r' \in [m]} \langle \boldsymbol{w}_{j,r'}^{(t,0)}, \boldsymbol{\xi}_{k,i} \rangle$. Suppose Claim E.4 E.5E.6 hold at $t$. Then*

$$\langle \boldsymbol{w}_{j,r}^{(t,0)}, \boldsymbol{\xi}_{k,i} \rangle \ge \Omega\Big(m^{-1/2}\Big(1 + \frac{\gamma \sigma_\xi^2 d}{n}\Big)^{-J}\Big).$$

*And*

$$\langle \boldsymbol{w}_{j,r}^{(t,J)}, \boldsymbol{\xi}_{k,i} \rangle \ge \Omega(m^{-1/2}).$$

*Proof.* The proof of the first inequality is very similar to the first part of the proof of Theorem E.3. Hence we will omit it and focus on proving the second inequality. We first recall from Eq. (C.2) that

$$\langle \mathbf{w}_{j,r}^{(t,\tau+1)}, \boldsymbol{\xi}_{k,i} \rangle = \langle \mathbf{w}_{j,r}^{(t,\tau)}, \boldsymbol{\xi}_{k,i} \rangle - \frac{\gamma}{n} \sum_{i' \ne i} \ell_{k,i'}^{\prime(t,\tau)} y_{k,i'} j \sigma'(\langle \mathbf{w}_{j,r}^{(t,\tau)}, \boldsymbol{\xi}_{k,i'} \rangle) \langle \boldsymbol{\xi}_{k,i'}, \boldsymbol{\xi}_{k,i} \rangle$$

$$- \frac{\gamma}{n} \ell_{k,i}^{\prime(t,\tau)} \sigma'(\langle \mathbf{w}_{j,r}^{(t,\tau)}, \boldsymbol{\xi}_{k,i} \rangle) \|\boldsymbol{\xi}_{k,i}\|_2^2$$

$$\ge \langle \mathbf{w}_{j,r}^{(t,\tau)}, \boldsymbol{\xi}_{k,i} \rangle - \widetilde{\mathcal{O}}(d^{-1/2}) - \frac{\gamma}{n} \ell_{k,i}^{\prime(t,\tau)} \sigma'(\langle \mathbf{w}_{j,r}^{(t,\tau)}, \boldsymbol{\xi}_{k,i} \rangle) \|\boldsymbol{\xi}_{k,i}\|_2^2, \qquad \text{(E.22)}$$

for all $\tau \in [J-1]$. Here we have used $-\ell'(\cdot) \le 1$, $\sigma'(\cdot) \le 1$ and Lemma D.2 to get the inequality. If $\langle \mathbf{w}_{j,r}^{(t,0)}, \boldsymbol{\xi}_{k,i} \rangle > m^{-1/2}$, then we have

$$\langle \mathbf{w}_{j,r}^{(t,J)}, \boldsymbol{\xi}_{k,i} \rangle \ge \langle \mathbf{w}_{j,r}^{(t,0)}, \boldsymbol{\xi}_{k,i} \rangle - J\widetilde{\mathcal{O}}(d^{-1/2}) = \Omega(m^{-1/2}).$$

The last equality is because $m^{-1/2} \gg J\widetilde{\mathcal{O}}(d^{-1/2})$. By the first part of this theorem, we have $\langle \mathbf{w}_{j,r}^{(t,0)}, \boldsymbol{\xi}_{k,i} \rangle \ge \Omega(m^{-1/2}(1 + \frac{\gamma \sigma_\xi^2 d}{n})^{-J})$. Let us consider $\Omega(m^{-1/2}(1 + \frac{\gamma \sigma_\xi^2 d}{n})^{-(\tau+1)}) \le \langle \mathbf{w}_{j,r}^{(t,0)}, \boldsymbol{\xi}_{k,i} \rangle \le \mathcal{O}(m^{-1/2}(1 + \frac{\gamma \sigma_\xi^2 d}{n})^{-\tau})$ for some $\tau \in [J-1]$. Using similar proof as the proof of Hypothesis E.2, it can be shown that $-\ell_{k,i}^{\prime(t,\tau')} = \Omega(1)$ for all $\tau' \le \tau$ if $\langle \mathbf{w}_{j,r}^{(t,0)}, \boldsymbol{\xi}_{k,i} \rangle \le \Omega(m^{-1/2}(1 + \frac{\gamma \sigma_\xi^2 d}{n})^{-\tau})$. Applying Eq. (E.22) $\tau + 1$ times, we get $\langle \mathbf{w}_{j,r}^{(t,\tau+1)}, \boldsymbol{\xi}_{k,i} \rangle \ge \langle \mathbf{w}_{j,r}^{(t,0)}, \boldsymbol{\xi}_{k,i} \rangle \Omega((1 + \frac{\gamma \sigma_\xi^2 d}{n})^{\tau+1})$, which implies $\langle \mathbf{w}_{j,r}^{(t,\tau+1)}, \boldsymbol{\xi}_{k,i} \rangle \ge \Omega(m^{-1/2})$ if $\langle \mathbf{w}_{j,r}^{(t,0)}, \boldsymbol{\xi}_{k,i} \rangle \ge \Omega(m^{-1/2}(1 + \frac{\gamma \sigma_\xi^2 d}{n})^{-(\tau+1)})$. Lastly, Eq. (E.22) also implies

$$\langle \mathbf{w}_{j,r}^{(t,J)}, \boldsymbol{\xi}_{k,i} \rangle \ge \langle \mathbf{w}_{j,r}^{(t,\tau+1)}, \boldsymbol{\xi}_{k,i} \rangle - (J - (\tau+1))\widetilde{\mathcal{O}}(d^{-1/2}) = \Omega(m^{-1/2}).$$

This proves the lemma. $\qquad \square$

**Lemma E.8.** *Suppose Claim E.4 E.5E.6 hold at t. Let $k \in [K]$, $i \in [n]$. If*

$$\Xi_{k,i}^{(t)} = \Theta\Big(m^{-1/2}\Big(1 + \frac{\gamma\sigma_\xi^2 d}{n}\Big)^{-J+1}\Big),$$

*then $-\ell_{k,i}'^{(t,J)} = \mathcal{O}(d^{-\log(d)^{0.3}})$.*

*Proof.* Without loss of generality, assume $y_{k,i} = 1$. Let $r'(t) = \arg\max_{r\in[m]}\langle\mathbf{w}_{1,r}^{(t,0)}, \boldsymbol{\xi}_{k,i}\rangle$. We have

$$-\ell_{k,i}'^{(t,J)} = -\ell'\bigg(\sum_{j=\pm 1}\sum_{r=1}^m j\Big(\sigma(\langle\mathbf{w}_{j,r}^{(t,J)}, \boldsymbol{\xi}_{k,i}\rangle) + \sigma(\langle\mathbf{w}_{j,r}^{(t,J)}, \boldsymbol{\nu}_k\rangle)\Big)\bigg)$$

$$\leq -\ell'\bigg(\sigma(\langle\mathbf{w}_{1,r'(t)}^{(t,J)}, \boldsymbol{\xi}_{k,i}\rangle) - \sum_{r=1}^m\Big(\sigma(\langle\mathbf{w}_{-1,r}^{(t,J)}, \boldsymbol{\xi}_{k,i}\rangle) + \sigma(\langle\mathbf{w}_{-1,r}^{(t,J)}, \boldsymbol{\nu}_k\rangle)\Big)\bigg).$$

Using similar proof as the proof of Hypothesis E.2, it can be shown that $-\ell_{k,i}'^{(t,\tau)} = \Omega(1)$ for all $\tau \in [J-1]$ under Claim E.5E.6. We then apply Eq. (E.22) $J-1$ times to get

$$\langle\mathbf{w}_{1,r'(t)}^{(t,J)}, \boldsymbol{\xi}_{k,i}\rangle \geq \Big(\langle\mathbf{w}_{1,r'(t)}^{(t,0)}, \boldsymbol{\xi}_{k,i}\rangle - J\widetilde{\mathcal{O}}(d^{-1/2})\Big)\Big(1 - \Theta(1)\frac{\gamma\sigma_\xi^2 d}{n}\Big)^J$$

$$= \Big(\Xi_{k,i}^{(t)} - J\widetilde{\mathcal{O}}(d^{-1/2})\Big)\Big(1 - \Theta(1)\frac{\gamma\sigma_\xi^2 d}{n}\Big)^J$$

$$= \Omega(1)\Xi_{k,i}^{(t)}\Big(1 - \Theta(1)\frac{\gamma\sigma_\xi^2 d}{n}\Big)^J$$

$$= \Omega(1)m^{-1/2}\frac{\gamma\sigma_\xi^2 d}{n}.$$

The third line is because by assumption we have $\Xi_{k,i}^{(t)} = \Theta\big(m^{-1/2}\big(1 + \frac{\gamma\sigma_\xi^2 d}{n}\big)^{-J+1}\big) = \widetilde{\Theta}(1) \gg J\widetilde{\mathcal{O}}(d^{-1/2})$. On the other hand, Claim E.4, tells us that

$$\langle\mathbf{w}_{-1,r}^{(t,J)}, \boldsymbol{\xi}_{k,i}\rangle = \widetilde{\mathcal{O}}(d^{-1/2}),$$

for all $r \in [m]$. Hence,

$$\sum_{r=1}^m\sigma(\langle\mathbf{w}_{-1,r}^{(t,J)}, \boldsymbol{\xi}_{k,i}\rangle) = m\widetilde{\mathcal{O}}(d^{-1/2}) = o(1).$$

By Eq. (C.1) and Eq. (E.11) we get that

$$\langle\mathbf{w}_{-1,r}^{(t,J)}, \boldsymbol{\nu}_k\rangle \leq \langle\mathbf{w}_{-1,r}^{(t,0)}, \boldsymbol{\nu}_k\rangle \leq \langle\mathbf{w}_{-1,r}^{(0,0)}, \boldsymbol{\nu}_k\rangle = \mathcal{O}(d^{-1/2}).$$

Thus,

$$\sum_{r=1}^m\sigma(\langle\mathbf{w}_{-1,r}^{(t,J)}, \boldsymbol{\nu}_k\rangle) = m\widetilde{\mathcal{O}}(d^{-1/2}) = o(1).$$

We now have

$$-\ell_{k,i}'^{(t,J)} \leq -\ell'\bigg(\sigma(\langle\mathbf{w}_{1,r'}^{(t,J)}, \boldsymbol{\xi}_{k,i}\rangle) - \sum_{r=1}^m\Big(\sigma(\langle\mathbf{w}_{-1,r}^{(t,J)}, \boldsymbol{\xi}_{k,i}\rangle) + \sigma(\langle\mathbf{w}_{-1,r}^{(t,J)}, \boldsymbol{\nu}_k\rangle)\Big)\bigg)$$

$$\leq -\ell'\Big(\Omega(1)m^{-1/2}\frac{\gamma\sigma_\xi^2 d}{n} - o(1) - o(1)\Big)$$

$$= \mathcal{O}(1)\exp\Big(-\Omega(1)m^{-1/2}\frac{\gamma\sigma_\xi^2 d}{n}\Big)$$

$$= \mathcal{O}(d^{-\log(d)^{0.3}}).$$

This proves the lemma. □

**Theorem E.9.** *If Claim E.4E.5E.6 hold for all $T^{(1)} \leq t \leq T - 1$, then*

$$\Xi_{k,i}^{(T)} \leq \mathcal{O}\Big(m^{-1/2}\Big(1 + \frac{\gamma\sigma_\xi^2 d}{n}\Big)^{-J+1}\Big),$$

*for all $k \in [K]$, $i \in [n]$.*

*Proof.* Set $A(q) = m^{-1/2}\left(1 + \frac{\gamma\sigma_\xi^2 d}{n}\right)^{-q}$ for $q \in [J]$. We proved in Theorem E.3 that $\Omega(A(J)) \leq \Xi_{k,i}^{(T^{(1)})} \leq \mathcal{O}(A(J - 0.5))$ for all $k \in [K]$, $i \in [n]$. Define

$$t(k,i) = \inf\left\{t < T : \Xi_{k,i}^{(t')} > A(J - 1), \forall t' > t\right\},$$

where the infimum over an empty set is defined as $\infty$. In words, $t(k,i)$ is the last iteration with $\Xi_{k,i}^{(t)} \leq A(J - 1)$. We have shown in the proof of Theorem E.3 that $\Xi_{k,i}^{(t(k,i)+1)} < 1.5A(J-1)$. Similarly define

$$\tilde{t}(k,i) = \inf\left\{t(k,i) \leq t \leq T : \Xi_{k,i}^{(t')} > 4A(J - 1), \forall t' > t\right\}.$$

Again the inf over an empty set is defined as $\infty$. So $\tilde{t}(k,i)$ is the last iteration with $\Xi_{k,i}^{(t)} \leq 4A(J - 1)$. Our goal is to show that $\tilde{t}(k,i) = \infty$ for any $k \in [K]$ and $i \in [n]$. We will prove this by contradiction. Let us suppose that $\tilde{t}(k,i) < \infty$. Then, for any $t \in [t(k,i) + 1, \tilde{t}(k,i)]$ we have that $-\ell_{k,i}^{\prime(t,J)} = \mathcal{O}(d^{-\log(d)^{0.3}})$ by Lemma E.8. Recall

$$\begin{aligned}
\Xi_{k,i}^{(t+1)} &\geq \Xi_{k,i}^{(t)} + \frac{\eta}{Kn} \sum_{(k',i') \neq (k,i)} \ell_{k',i'}^{\prime(t,J)} \sigma'(\langle \mathbf{w}_{j,r}^{(t,J)}, \boldsymbol{\xi}_{k',i'}\rangle)\Theta(\sigma_\xi^2\sqrt{d}\sqrt{\log(Kn)}) \\
&\quad - \frac{\eta}{Kn}\ell_{k,i}^{\prime(t,J)}\sigma'(\langle \mathbf{w}_{j,r}^{(t,J)}, \boldsymbol{\xi}_{k,i}\rangle)\Theta(\sigma_\xi^2 d) \\
&\geq \Xi_{k,i}^{(t)} + \frac{\eta}{Kn} \sum_{(k',i') \neq (k,i)} \ell_{k',i'}^{\prime(t,J)}\Theta(\sigma_\xi^2\sqrt{d}\sqrt{\log(Kn)}) \\
&\quad - \frac{\eta}{Kn}\ell_{k,i}^{\prime(t,J)}\sigma'(\langle \mathbf{w}_{j,r}^{(t,J)}, \boldsymbol{\xi}_{k,i}\rangle)\Theta(\sigma_\xi^2 d),
\end{aligned}$$
(E.23)

where the second inequality is because $\sigma'(\cdot) \leq 1$. And similarly we have

$$\begin{aligned}
\Xi_{k,i}^{(t+1)} &\leq \Xi_{k,i}^{(t)} - \frac{\eta}{Kn} \sum_{(k',i') \neq (k,i)} \ell_{k',i'}^{\prime(t,J)} \sigma'(\langle \mathbf{w}_{j,r}^{(t,J)}, \boldsymbol{\xi}_{k',i'}\rangle)\Theta(\sigma_\xi^2\sqrt{d}\sqrt{\log(Kn)}) \\
&\quad - \frac{\eta}{Kn}\ell_{k,i}^{\prime(t,J)}\sigma'(\langle \mathbf{w}_{j,r}^{(t,J)}, \boldsymbol{\xi}_{k,i}\rangle)\Theta(\sigma_\xi^2 d) \\
&\leq \Xi_{k,i}^{(t)} - \frac{\eta}{Kn} \sum_{(k',i') \neq (k,i)} \ell_{k',i'}^{\prime(t,J)}\Theta(\sigma_\xi^2\sqrt{d}\sqrt{\log(Kn)}) - \frac{\eta}{Kn}\ell_{k,i}^{\prime(t,J)}\Theta(\sigma_\xi^2 d),
\end{aligned}$$
(E.24)

where we have again used $\sigma'(\cdot) \leq 1$ to the second inequality. We have the following bound due to Lemma E.8

$$\begin{aligned}
-\sum_{t=t(k,i)+1}^{\tilde{t}(k,i)} \frac{\eta}{Kn}\ell_{k,i}^{\prime(t,J)}\Theta(\sigma_\xi^2 d) &= \sum_{t=t(k,i)+1}^{\tilde{t}(k,i)} \frac{\eta}{Kn}\Theta(\sigma_\xi^2 d)\mathcal{O}(d^{-\log(d)^{0.3}}) \\
&\leq \frac{\eta\sigma_\xi^2 d}{Kn}T\mathcal{O}(d^{-\log(d)^{0.3}}) \\
&= o(d^{-1}),
\end{aligned}$$
(E.25)

where the last equality is because $T = \mathcal{O}(\text{poly}(d))$. Let

$$(k',i') = \underset{(\hat{k},\hat{i}) \neq (k,i)}{\arg\max} \sum_{t=t(k,i)+1}^{\tilde{t}(k,i)} -\ell_{\hat{k},\hat{i}}^{\prime(t,J)}.$$

Suppose it holds that

$$\sum_{t=t(k,i)+1}^{\tilde{t}(k,i)} -\ell_{k',i'}^{\prime(t,J)} < d^{1/4} \, . \tag{E.26}$$

Then using Eq. (E.24) we get

$$\Xi_{k,i}^{(\tilde{t}(k,i)+1)} \leq \Xi_{k,i}^{(t(k,i)+1)} - \frac{\eta}{Kn} \sum_{t=t(k,i)+1}^{\tilde{t}(k,i)} \sum_{(\hat{k},\hat{i})\neq(k,i)} \ell_{k',i'}^{\prime(t,J)} \Theta(\sigma_\xi^2 \sqrt{d}\sqrt{\log(Kn)})$$

$$- \frac{\eta}{Kn} \sum_{t=t(k,i)+1}^{\tilde{t}(k,i)} \ell_{k,i}^{\prime(t,J)} \Theta(\sigma_\xi^2 d)$$

$$\leq \Xi_{k,i}^{(t(k,i)+1)} + \eta\Theta(\sigma_\xi^2 \sqrt{d}\sqrt{\log(Kn)})d^{1/4} + o(d^{-1})$$

$$\leq 2\Xi_{k,i}^{(t(k,i)+1)}$$

$$\leq 3A(J-1) \, .$$

The second inequality is due to Eq. (E.25), Eq. (E.26) and the definition of $(k',i')$. The third inequality is because $\eta\Theta(\sigma_\xi^2 \sqrt{d}\sqrt{\log(Kn)})d^{1/4} + o(d^{-1}) = \widetilde{\mathcal{O}}(d^{-1/4})$ and $\Xi_{k,i}^{(t(k,i)+1)} \geq A(J-1) = \widetilde{\Theta}(1)$. Thus Eq. (E.26) contradicts $\Xi_{k,i}^{(\tilde{t}(k,i)+1)} \geq 4A(J-1)$ and we must have $\tilde{t}(k,i) = \infty$. Next, we will show that if Eq. (E.26) does not hold, then we also have $\tilde{t}(k,i) = \infty$. Let us suppose

$$\sum_{t=t(k,i)+1}^{\tilde{t}(k,i)} -\ell_{k',i'}^{\prime(t,J)} \geq d^{1/4} \, .$$

Since $-\ell_{k',i'}^{\prime(t,J)} > 0$ for any $t$, we get that $\sum_{t=t(k,i)+1}^{t_0} -\ell_{k',i'}^{\prime(t,J)}$ is increasing in $t_0$. Let $t(k,i)+1 < t_0 < \tilde{t}(k,i)$ be the first iteration such that

$$\sum_{t=t(k,i)+1}^{t_0} -\ell_{k',i'}^{\prime(t,J)} \geq d^{-1/8} \sum_{t=t(k,i)+1}^{\tilde{t}(k,i)} -\ell_{k',i'}^{\prime(t,J)} \geq d^{-1/8+1/4} = d^{1/8} \, .$$

Since each $-\ell_{k',i'}^{\prime(t,J)} < 1$, we get that

$$\sum_{t=t(k,i)+1}^{t_0} -\ell_{k',i'}^{\prime(t,J)} = \sum_{t=t(k,i)+1}^{t_0-1} -\ell_{k',i'}^{\prime(t,J)} - \ell_{k',i'}^{\prime(t_0,J)}$$

$$\leq 1 + d^{-1/8} \sum_{t=t(k,i)+1}^{\tilde{t}(k,i)} -\ell_{k',i'}^{\prime(t,J)}$$

$$\leq 2d^{-1/8} \sum_{t=t(k,i)+1}^{\tilde{t}(k,i)} -\ell_{k',i'}^{\prime(t,J)} \, , \tag{E.27}$$

where the second and third line are by definition of $t_0$. We can sum Eq. (E.23) over time to obtain

$$\sum_{t=t(k,i)+1}^{t_0} \Xi_{k',i'}^{(t+1)} \geq \sum_{t=t(k,i)+1}^{t_0} \Xi_{k',i'}^{(t)} + \frac{\eta}{Kn} \sum_{t=t(k,i)+1}^{t_0} \sum_{(\hat{k},\hat{i})\neq(k',i')} \ell_{\hat{k},\hat{i}}^{\prime(t,J)} \Theta(\sigma_\xi^2 \sqrt{d}\sqrt{\log(Kn)})$$

$$- \frac{\eta}{Kn} \sum_{t=t(k,i)+1}^{t_0} \ell_{k',i'}^{\prime(t,J)} \sigma'(\langle \mathbf{w}_{j,r}^{(t,J)}, \boldsymbol{\xi}_{k',i'} \rangle) \Theta(\sigma_\xi^2 d)$$

$$\geq \sum_{t=t(k,i)+1}^{t_0} \Xi_{k',i'}^{(t)} + \frac{\eta}{Kn} \sum_{t=t(k,i)+1}^{\tilde{t}(k,i)} \sum_{(\hat{k},\hat{i})\neq(k',i')} \ell_{\hat{k},\hat{i}}^{\prime(t,J)} \Theta(\sigma_\xi^2 \sqrt{d}\sqrt{\log(Kn)})$$

$$- \frac{\eta}{Kn} \sum_{t=t(k,i)+1}^{t_0} \ell'^{(t,J)}_{k',i'} \sigma'(\langle \mathbf{w}_{j,r}^{(t,J)}, \boldsymbol{\xi}_{k',i'} \rangle) \Theta(\sigma_\xi^2 d)$$

$$\geq \sum_{t=t(k,i)+1}^{t_0} \boldsymbol{\Xi}_{k',i'}^{(t)} + \frac{\eta}{Kn} \sum_{t=t(k,i)+1}^{\tilde{t}(k,i)} \sum_{(\hat{k},\hat{\imath})\neq(k',i')} \ell'^{(t,J)}_{k',i'} \Theta(\sigma_\xi^2 \sqrt{d}\sqrt{\log(Kn)})$$

$$- \frac{\eta}{Kn} \sum_{t=t(k,i)+1}^{t_0} \ell'^{(t,J)}_{k',i'} \sigma'(\langle \mathbf{w}_{j,r}^{(t,J)}, \boldsymbol{\xi}_{k',i'} \rangle) \Theta(\sigma_\xi^2 d) ,$$

where the second inequality is because $t_0 < \tilde{t}(k,i)$. We have used the definition of $(k', i')$ to get the last inequality. Then, subtracting $\sum_{t=t(k,i)+1}^{t_0} \boldsymbol{\Xi}_{k',i'}^{(t)}$ from both sides yields

$$\boldsymbol{\Xi}_{k',i'}^{(t_0+1)} - \boldsymbol{\Xi}_{k',i'}^{(t(k,i)+1)} \geq \Theta\Big(\frac{\eta\sigma_\xi^2\sqrt{d}}{Kn}\sqrt{\log(Kn)}\Big) Kn \sum_{t=t(k,i)+1}^{\tilde{t}(k,i)} \ell'^{(t,J)}_{k',i'}$$

$$- \Theta\Big(\frac{\eta\sigma_\xi^2 d}{Kn\sqrt{m}}\Big) \sum_{t=t(k,i)+1}^{t_0} \ell'^{(t,J)}_{k',i'}$$

$$= \widetilde{\Theta}(d^{-1/2}) \sum_{t=t(k,i)+1}^{\tilde{t}(k,i)} \ell'^{(t,J)}_{k',i'} - \widetilde{\Theta}(1)d^{-1/8} \sum_{t=t(k,i)+1}^{\tilde{t}(k,i)} \ell'^{(t,J)}_{k',i'}$$

$$= -\widetilde{\Theta}(1)d^{-1/8} \sum_{t=t(k,i)+1}^{\tilde{t}(k,i)} \ell'^{(t,J)}_{k',i'}$$

$$\geq \widetilde{\Theta}(d^{1/8}) , \tag{E.28}$$

where we used Lemma E.7 to get the first inequality. The first equality is due to Eq. (E.27). The last inequality is by our assumption $\sum_{t=t(k,i)+1}^{\tilde{t}(k,i)} -\ell'^{(t,J)}_{k',i'} \geq d^{1/4}$. It is then straightforward to check that

$$\boldsymbol{\Xi}_{k',i'}^{(t)} \geq \widetilde{\Theta}(d^{1/8}) ,$$

for all $t_0 + 1 \leq t \leq \tilde{t}(k,i)$. To see this, let us sum Eq. (E.23) over time to obtain

$$\sum_{\hat{t}=t_0+1}^{t-1} \boldsymbol{\Xi}_{k',i'}^{(\hat{t}+1)} \geq \sum_{\hat{t}=t_0+1}^{t-1} \boldsymbol{\Xi}_{k',i'}^{(\hat{t})} + \frac{\eta}{Kn} \sum_{\hat{t}=t_0+1}^{t-1} \sum_{(\tilde{k},\tilde{\imath})\neq(k',i')} \ell'^{(\hat{t},J)}_{\tilde{k},\tilde{\imath}} \Theta(\sigma_\xi^2 \sqrt{d}\sqrt{\log(Kn)})$$

$$- \sum_{\hat{t}=t_0+1}^{t-1} \frac{\eta}{Kn} \ell'^{(\hat{t},J)}_{k',i'} \sigma'(\langle \mathbf{w}_{j,r}^{(\hat{t},J)}, \boldsymbol{\xi}_{k',i'} \rangle) \Theta(\sigma_\xi^2 d)$$

$$\geq \sum_{\hat{t}=t_0+1}^{t-1} \boldsymbol{\Xi}_{k',i'}^{(\hat{t})} + \frac{\eta}{Kn} \sum_{\hat{t}=t_0+1}^{t-1} \sum_{(\tilde{k},\tilde{\imath})\neq(k',i')} \ell'^{(\hat{t},J)}_{\tilde{k},\tilde{\imath}} \Theta(\sigma_\xi^2 \sqrt{d}\sqrt{\log(Kn)})$$

$$\geq \sum_{\hat{t}=t_0+1}^{t-1} \boldsymbol{\Xi}_{k',i'}^{(\hat{t})} + \frac{\eta}{Kn} \sum_{\hat{t}=t(k,i)+1}^{\tilde{t}(k,i)} \sum_{(\tilde{k},\tilde{\imath})\neq(k',i')} \ell'^{(\hat{t},J)}_{\tilde{k},\tilde{\imath}} \Theta(\sigma_\xi^2 \sqrt{d}\sqrt{\log(Kn)})$$

$$\geq \sum_{\hat{t}=t_0+1}^{t-1} \boldsymbol{\Xi}_{k',i'}^{(\hat{t})} + \frac{\eta}{Kn} \sum_{\hat{t}=t(k,i)+1}^{\tilde{t}(k,i)} Kn\ell'^{(\hat{t},J)}_{k',i'} \Theta(\sigma_\xi^2 \sqrt{d}\sqrt{\log(Kn)})$$

$$\boldsymbol{\Xi}_{k',i'}^{(t)} \geq \boldsymbol{\Xi}_{k',i'}^{(t_0+1)} + \Theta\Big(\frac{\eta\sigma_\xi^2\sqrt{d}\sqrt{\log(Kn)}}{Kn}\Big) Kn \sum_{t=t(k,i)+1}^{\tilde{t}(k,i)} \ell'^{(t,J)}_{k',i'}$$

$$\geq \widetilde{\Theta}(1)d^{-1/8} \sum_{t=t(k,i)+1}^{\tilde{t}(k,i)} -\ell'^{(t,J)}_{k',i'} + \widetilde{\Theta}(d^{-1/2}) \sum_{t=t(k,i)+1}^{\tilde{t}(k,i)} \ell'^{(t,J)}_{k',i'}$$

$$= \widetilde{\Theta}(1) d^{-1/8} \sum_{t=t(k,i)+1}^{\tilde{t}(k,i)} -\ell'^{(t,J)}_{k',i'}$$

$$\geq \widetilde{\Theta}(d^{1/8}),$$

where the second inequality is because we are removing a positive term from the sum. The third inequality is because $t-1 < \tilde{t}(k,i)$ and $t_0 > t(k,i)$. The fourth inequality is by definition of $(k', i')$. The second last inequality is by the second last line of Eq. (E.28). And our last inequality follows Eq. (E.28). Then, using similar proof of Lemma E.8, we have $-\ell'^{(t,J)}_{k',i'} = \mathcal{O}(\exp(-\widetilde{\Theta}(d^{1/8})))$ for all $t_0 + 1 \leq t \leq \tilde{t}(k,i)$. This yields

$$\sum_{t=t(k,i)+1}^{\tilde{t}(k,i)} -\ell'^{(t,J)}_{k',i'} = \sum_{t=t(k,i)+1}^{t_0} -\ell'^{(t,J)}_{k',i'} + \sum_{t=t_0+1}^{\tilde{t}(k,i)} -\ell'^{(t,J)}_{k',i'}$$

$$\leq 2d^{-1/8} \sum_{t=t(k,i)+1}^{\tilde{t}(k,i)} -\ell'^{(t,J)}_{k',i'} + \sum_{t=t_0+1}^{\tilde{t}(k,i)} \mathcal{O}(\exp(-\widetilde{\Theta}(d^{1/8})))$$

$$\leq 2d^{-1/8} \sum_{t=t(k,i)+1}^{\tilde{t}(k,i)} -\ell'^{(t,J)}_{k',i'} + T \cdot \mathcal{O}(\exp(-\widetilde{\Theta}(d^{1/8})))$$

$$\leq 2d^{-1/8} \sum_{t=t(k,i)+1}^{\tilde{t}(k,i)} -\ell'^{(t,J)}_{k',i'} + o(d^{-1})$$

$$\leq (2d^{-1/8} + o(d^{-1})) \sum_{t=t(k,i)+1}^{\tilde{t}(k,i)} -\ell'^{(t,J)}_{k',i'}, \tag{E.29}$$

where we used Eq. (E.27) to get the second line. And we used $T = \mathcal{O}(\text{poly}(d))$ to get the fourth line. To obtain the last inequality, we recall the assumption $\sum_{t=t(k,i)+1}^{\tilde{t}(k,i)} -\ell'^{(t,J)}_{k',i'} > d^{1/4}$, which implies $o(d^{-1}) < o(d^{-1}) \sum_{t=t(k,i)+1}^{\tilde{t}(k,i)} -\ell'^{(t,J)}_{k',i'}$. Eq. (E.29) is a contradiction in the form of $x \leq Ax$ for some $x \geq 0$ and $A < 1$. This implies that under the assumption $\sum_{t=t(k,i)+1}^{\tilde{t}(k,i)} -\ell'^{(t,J)}_{k',i'} > d^{1/4}$, we should still have $\tilde{t}(k,i) = \infty$. Combining with our previous result which says $\tilde{t}(k,i) = \infty$ when $\sum_{t=t(k,i)+1}^{\tilde{t}(k,i)} -\ell'^{(t,J)}_{k',i'} \leq d^{1/4}$, we conclude that $\tilde{t}(k,i) = \infty$ and we are done. $\qquad\square$

Now that we know the noise inner products are upper bounded by $\mathcal{O}\big(m^{-1/2}\big(1 + \frac{\gamma\sigma_\xi^2 d}{n}\big)^{-J+1}\big)$, we can provide a convergence guarantee based on this upper bound.

**Theorem E.10** (Restatement of Theorem 4.4 (1))**.** *Under Condition D.1 and Condition 4.1, let* $T = \text{poly}(d)$ *be the total number of iterations. Then*

$$\min_{t \in [T]} \widehat{\mathcal{L}}^{tr\text{-}tr}(\boldsymbol{W}^{(t)}) \leq \widetilde{\Theta}(T^{-1}\eta^{-1}).$$

*Proof.* By Theorem E.9, we know that

$$\boldsymbol{\Xi}^{(T)}_{k,i} \leq \mathcal{O}\Big(m^{-1/2}\Big(1 + \frac{\gamma\sigma_\xi^2 d}{n}\Big)^{-J+1}\Big) = \widetilde{\mathcal{O}}(1),$$

for all $k \in [K]$ and $i \in [n]$. Denote $\boldsymbol{\Xi}^{(t)} = \sum_{k,i} \boldsymbol{\Xi}^{(t)}_{k,i}$. It then follows that $\boldsymbol{\Xi}^{(T)} \leq Kn\widetilde{\mathcal{O}}(1) = \widetilde{\mathcal{O}}(1)$. Moreover, if we sum Eq. (E.24) over all $k, i$, we get

$$\boldsymbol{\Xi}^{(t+1)} \geq \boldsymbol{\Xi}^{(t)} + \eta(Kn-1) \sum_{(k,i)} \ell'^{(t,J)}_{k,i} \Theta(\sigma_\xi^2\sqrt{d}\sqrt{\log(Kn)}) - \frac{\eta}{Kn} \sum_{k,i} \ell'^{(t,J)}_{k,i} \Omega\Big(\frac{1}{\sqrt{m}}\Big)\Theta(\sigma_\xi^2 d)$$

$$\geq \boldsymbol{\Xi}^{(t)} - \Omega(1)\frac{\eta}{Kn} \sum_{k,i} \ell'^{(t,J)}_{k,i} \Omega\Big(\frac{1}{\sqrt{m}}\Big)\Theta(\sigma_\xi^2 d),$$

where the $\Omega\left(\frac{1}{\sqrt{m}}\right)$ term is by Lemma E.7. We can then sum over $t$ to obtain

$$\sum_{t=T^{(1)}}^{T-1} \Xi^{(t+1)} \geq \sum_{t=T^{(1)}}^{T-1} \Xi^{(t)} - \Omega(1)\frac{\eta}{Kn}\sum_{t=T^{(1)}}^{T-1}\sum_{k,i}\ell_{k,i}^{\prime(t,J)}\frac{2}{\sqrt{m}}\Theta(\sigma_\xi^2 d)\,,$$

$$\Xi^{(T)} - \Xi^{(T^{(1)})} \geq -\Omega\left(\frac{\eta\sigma_\xi^2 d}{Kn\sqrt{m}}\right)\sum_{t=T^{(1)}}^{T-1}\sum_{k,i}\ell_{k,i}^{\prime(t,J)}\,,$$

$$\eta\sum_{t=T^{(1)}}^{T-1}\sum_{k,i}-\ell_{k,i}^{\prime(t,J)} \leq \mathcal{O}\left(\frac{K\sqrt{m}}{\sigma_\xi^2 d}\right)\widetilde{\mathcal{O}}(1) = \widetilde{\mathcal{O}}(1)\,. \tag{E.30}$$

The last inequality is because $\Xi^{(T)} - \Xi^{(T^{(1)})} \leq \widetilde{\mathcal{O}}(1)$. And we can bound the minimum over $t$ by

$$\min_{t\in[T^{(1)},T-1]}\sum_{k,i}-\ell_{k,i}^{\prime(t,J)} \leq \widetilde{\mathcal{O}}\left(\frac{1}{(T-T^{(1)})\eta}\right) = \widetilde{\mathcal{O}}\left(\frac{1}{T\eta}\right)\,,$$

where the second equality is because $T^{(1)} = \text{polylog}(d) = o(T)$. Denote $-\ell_{k,i}^{\prime(t,J)} = -\ell'(\alpha(t,J,k,i))$, i.e. the argument of $-\ell_{k,i}^{\prime(t,J)}$ is $\alpha(t,J,k,i)$. Then the loss is related to $-\ell_{k,i}^{\prime(t,J)}$ by

$$\widehat{\mathcal{L}}^{\text{tr-tr}}(\mathbf{W}^{(t)}) = \sum_{k,i}\ell\left(\alpha(t,J,k,i)\right)\,.$$

Let $t' = \arg\min_t\sum_{k,i}-\ell_{k,i}^{\prime(t,J)}$. By definition of $\ell'(\cdot)$ and $\ell(\cdot)$ we can bound $-\ell'(x) \geq \exp(-x)/2 \geq \ell(x)/2$ for $x > 0$. This can be used to bound our loss by

$$\min_{t\in[T]}\widehat{\mathcal{L}}^{\text{tr-tr}}(\mathbf{W}^{(t)}) \leq \widehat{\mathcal{L}}^{\text{tr-tr}}(\mathbf{W}^{(t')})$$

$$= \sum_{k,i}\ell\left(\alpha(t',J,k,i)\right)$$

$$\leq -2\sum_{k,i}\ell'\left(\alpha(t,J,k,i)\right)$$

$$\leq \widetilde{\mathcal{O}}\left(\frac{1}{T\eta}\right)\,.$$

This proves our theorem. $\qquad\square$

It remains to prove Claims E.4 E.5 E.6.

*Proof of Claim E.4.* Without loss of generality, assume $y_{k,i} = 1$. By Eq. (E.16), we have the following upper bound

$$\langle\mathbf{w}_{-1,r}^{(t+1,0)},\boldsymbol{\xi}_{k,i}\rangle \leq \langle\mathbf{w}_{-1,r}^{(t,0)},\boldsymbol{\xi}_{k,i}\rangle - \frac{\eta}{Kn}\sum_{(k',i')\neq(k,i)}\ell_{k',i'}^{\prime(t,J)}\sigma'(\langle\mathbf{w}_{-1,r}^{(t,J)},\boldsymbol{\xi}_{k',i'}\rangle)\Theta(\sigma_\xi^2\sqrt{d}\sqrt{\log(Kn)})$$

$$\leq \langle\mathbf{w}_{-1,r}^{(t,0)},\boldsymbol{\xi}_{k,i}\rangle - \frac{\eta}{Kn}\sum_{(k',i')}\ell_{k',i'}^{\prime(t,J)}\sigma'(\langle\mathbf{w}_{-1,r}^{(t,J)},\boldsymbol{\xi}_{k',i'}\rangle)\Theta(\sigma_\xi^2\sqrt{d}\sqrt{\log(Kn)})$$

$$\leq \langle\mathbf{w}_{-1,r}^{(t,0)},\boldsymbol{\xi}_{k,i}\rangle - \frac{\eta}{Kn}\Theta(\sigma_\xi^2\sqrt{d}\sqrt{\log(Kn)})\sum_{(k',i')}\ell_{k',i'}^{\prime(t,J)}\,,$$

where the second inequality is because we are adding a non-negative term to the sum. The last inequality is because $\sigma'(\cdot) \leq 1$. Suppose all of our claims hold for $t \leq T'$, where $T' \leq T - 1$. Using a similar derivation of Eq. (E.30) we have

$$-\frac{\eta}{Kn}\Theta(\sigma_\xi^2\sqrt{d}\sqrt{\log(Kn)})\sum_{t=T^{(1)}}^{T'}\sum_{(k',i')}\ell_{k',i'}^{\prime(t,J)} \leq \frac{K\Theta(\sigma_\xi^2\sqrt{d}\sqrt{\log(Kn)})}{(\sigma_\xi^2 d)^J\gamma^{J-1}n^{J-2}}\,.$$

Then we get

$$\begin{aligned}
\langle \mathbf{w}_{-1,r}^{(T'+1,0)}, \boldsymbol{\xi}_{k,i} \rangle &\leq \langle \mathbf{w}_{-1,r}^{(T^{(1)},0)}, \boldsymbol{\xi}_{k,i} \rangle - \frac{\eta}{Kn} \Theta(\sigma_\xi^2 \sqrt{d}\sqrt{\log(Kn)}) \sum_{t=T^{(1)}}^{T'} \sum_{(k',i')} \ell_{k',i'}^{\prime(t,J)} \\
&\leq \langle \mathbf{w}_{-1,r}^{(T^{(1)},0)}, \boldsymbol{\xi}_{k,i} \rangle + \frac{K\Theta(\sigma_\xi^2 \sqrt{d}\sqrt{\log(Kn)})}{(\sigma_\xi^2 d)^J \gamma^{J-1} n^{J-2}} \\
&\leq \widetilde{\mathcal{O}}(d^{-1/2}) + \widetilde{\mathcal{O}}(d^{-1/2}) \\
&= \widetilde{\mathcal{O}}(d^{-1/2}),
\end{aligned}$$

where the third line is due to part (2) of Theorem E.2. This proves our claim. $\qquad\square$

*Proof of Claim E.5.* We will show that for any $T' \leq T = \mathcal{O}(\mathrm{poly}(d))$, we have that $\boldsymbol{\Lambda}_j^{(T')} = \mathcal{O}(d^{-1/4})$ for $j \in \{-1, 1\}$. We have the following upper bound

$$\begin{aligned}
\boldsymbol{\Lambda}_j^{(t+1)} &\leq \boldsymbol{\Lambda}_j^{(t)} - \frac{\eta}{Kn} \sum_{k,i} \ell_{k,i}^{\prime(t,J)} \boldsymbol{\Lambda}_j^{(t)} \gamma^J \mathcal{O}(1) \\
&\leq \boldsymbol{\Lambda}_j^{(t)} \left( 1 - \mathcal{O}(1) \frac{\eta\gamma^J}{Kn} \sum_{k,i} \ell_{k,i}^{\prime(t,J)} \right)
\end{aligned}$$

Suppose that there exists some $\tilde{t} \leq T'$ such that $\boldsymbol{\Lambda}_j^{(\tilde{t})} \geq d^{-1/4}$ for the first time. Then

$$d^{-1/4} \leq \boldsymbol{\Lambda}_j^{(\tilde{t})} \leq \boldsymbol{\Lambda}_j^{(T^{(1)})} \prod_{t=T^{(1)}}^{\tilde{t}-1} \left( 1 - \mathcal{O}(1) \frac{\eta\gamma^J}{Kn} \sum_{k,i} \ell_{k,i}^{\prime(t,J)} \right),$$

$$\frac{d^{-1/4}}{\boldsymbol{\Lambda}_j^{(T^{(1)})}} \leq \prod_{t=T^{(1)}}^{\tilde{t}-1} \left( 1 - \mathcal{O}(1) \frac{\eta\gamma^J}{Kn} \sum_{k,i} \ell_{k,i}^{\prime(t,J)} \right),$$

$$\prod_{t=T^{(1)}}^{\tilde{t}-1} \left( 1 - \mathcal{O}(1) \frac{\eta\gamma^J}{Kn} \sum_{k,i} \ell_{k,i}^{\prime(t,J)} \right) \geq \widetilde{\Omega}(d^{1/4}),$$

$$\sum_{t=T^{(1)}}^{\tilde{t}-1} \log \left( 1 - \mathcal{O}(1) \frac{\eta\gamma^J}{Kn} \sum_{k,i} \ell_{k,i}^{\prime(t,J)} \right) \geq \Omega(\log(d)),$$

$$\sum_{t=T^{(1)}}^{\tilde{t}-1} -\mathcal{O}(1) \frac{\eta\gamma^J}{Kn} \sum_{k,i} \ell_{k,i}^{\prime(t,J)} \geq \Omega(\log(d)),$$

$$\sum_{t=T^{(1)}}^{\tilde{t}-1} \sum_{k,i} -\ell_{k,i}^{\prime(t,J)} \geq \Omega\left( \frac{\log(d)Kn}{\eta\gamma^J} \right).$$

The third inequality is because by the first part of Theorem E.2, we have $\boldsymbol{\Lambda}_j^{(T^{(1)})} = \widetilde{\mathcal{O}}(d^{-1/2})$. And hence $d^{-1/4}/\boldsymbol{\Lambda}_j^{(T^{(1)})} = \widetilde{\Omega}(d^{1/4})$. Note that in the proof of Theorem E.10, we have that

$$\sum_{t=T^{(1)}}^{T} \sum_{k,i} -\ell_{k,i}^{\prime(t,J)} \leq \frac{K^2}{\gamma^{J-1}\eta(\sigma_\xi^2 d)^J} = o\left( \frac{\log(d)Kn}{\eta\gamma^J} \right),$$

which creates a contradiction. Hence, for all $\tilde{t} \leq T'$, we have $\boldsymbol{\Lambda}_j^{(\tilde{t})} \leq d^{-1/4}$. This proves the claim. $\qquad\square$

*Proof of Claim E.6.* By our result in Phase I, we have that Claim E.6 holds at $T^{(1)}$. The rest of the proof is similar to the proof of Hypothesis E.1 and we will omit it here. $\qquad\square$

We have shown that for $T = \text{poly}(d)$, the feature has not been learnt well if one uses the train-train method. Thus the testing loss will be large, which is characterized by the following theorem.

**Theorem E.11** (Restatement of Theorem 4.4 (2)). *Under Condition 4.1 and Condition D.1, the test loss is large throughout the whole training process*

$$\min_{t \in [T]} \mathcal{L}_{\text{test}}(\boldsymbol{W}^{(t)}) = \Omega(1).$$

*Proof.* Sample a new example $\mathbf{x}$ from the distribution, since we use the convolutional structure, we can assume that the first patch $\mathbf{x}^{(1)} = y \cdot (\boldsymbol{\nu} + \mathbf{z})$ and the second patch $\mathbf{x}^{(2)} = \boldsymbol{\xi}$. Clearly $\langle \boldsymbol{\xi}_{k,i}, \boldsymbol{\xi} \rangle$ follows the Gaussian distribution with mean zero and standard deviation $\sigma_{\xi} \cdot \|\boldsymbol{\xi}_{k,i}\|_2$. By Lemma D.2, we further know that $\|\boldsymbol{\xi}_{k,i}\|_2 \le 2\sigma_{\xi}\sqrt{d}$. Therefore

$$\mathbb{P}\big(|\langle \boldsymbol{\xi}_{k,i}, \boldsymbol{\xi} \rangle| \ge d^{-1/4}\big) \le 2\exp\left(-\frac{1}{8\sigma_{\xi}^4 d^{3/2}}\right) \le 2\exp(-d^{1/4}). \tag{E.31}$$

Similarly we have that

$$\mathbb{P}\big(|\langle \mathbf{w}_{j,r}^{(0,0)}, \boldsymbol{\xi} \rangle| \ge d^{3/2}\big) \le 2\exp\left(-\frac{1}{8\sigma_0^2 \sigma_{\xi}^2 d^{3/2}}\right) \le 2\exp(-d^{1/4}). \tag{E.32}$$

Denote $\mathcal{E}$ to be the event that $|\langle \boldsymbol{\xi}_{k,i}, \boldsymbol{\xi} \rangle| \le d^{-1/4}$ and $|\langle \mathbf{w}_{j,r}^{(0,0)}, \boldsymbol{\xi} \rangle| \le d^{3/2}$ for all $k \in [K]$, $i \in [n]$, $r \in [m]$ and $j \in \{-1,1\}$. Applying union bound, we have that the $\mathbb{P}(\mathcal{E}) \ge 1 - 2(nK + m)\exp(-d^{1/4}) \ge 1/2$. We can divide $\mathcal{L}_{\text{test}}(\mathbf{W})$ into two parts:

$$\begin{aligned}
\mathcal{L}_{\text{test}}(\mathbf{W}) &= \mathbb{E}\big[\ell\big(yf(\mathbf{W}, \mathbf{x})\big)\big] \\
&= \mathbb{E}[\mathbf{1}(\mathcal{E})\ell\big(yf(\mathbf{W}, \mathbf{x})\big)] + \mathbb{E}[\mathbf{1}(\mathcal{E}^c)\ell\big(yf(\mathbf{W}, \mathbf{x})\big)] \\
&\ge \mathbb{E}[\mathbf{1}(\mathcal{E})\ell\big(yf(\mathbf{W}, \mathbf{x})\big)].
\end{aligned} \tag{E.33}$$

Let $T' \le T$. And suppose that $T' \ge T^{(1)}$. When event $\mathcal{E}$ holds, for any $j \in \{-1,1\}$, we have

$$\begin{aligned}
\langle \mathbf{w}_{j,r}^{(T',0)}, \boldsymbol{\xi} \rangle &= \langle \mathbf{w}_{j,r}^{(0,0)}, \boldsymbol{\xi} \rangle - \frac{\eta}{Kn_2} \sum_{t=0}^{T'-1} \sum_{(k,i) \in \Psi} \ell_{k,i}'^{(t,J)} \sigma'(\langle \mathbf{w}_{j,r}^{(t,J)}, \boldsymbol{\xi}_{k,i} \rangle) \langle \boldsymbol{\xi}, \boldsymbol{\xi}_{k,i} \rangle \\
&\le \langle \mathbf{w}_{j,r}^{(0,0)}, \boldsymbol{\xi} \rangle - \frac{\eta}{Kn_2} \sum_{t=0}^{T'-1} \sum_{(k,i) \in \Psi} \ell_{k,i}'^{(t,J)} \sigma'(\langle \mathbf{w}_{j,r}^{(t,J)}, \boldsymbol{\xi}_{k,i} \rangle) d^{-1/4} \\
&\le \langle \mathbf{w}_{j,r}^{(0,0)}, \boldsymbol{\xi} \rangle - \frac{\eta}{Kn_2} \sum_{t=0}^{T'-1} \sum_{(k,i) \in \Psi} \ell_{k,i}'^{(t,J)} d^{-1/4} \\
&\le \langle \mathbf{w}_{j,r}^{(0,0)}, \boldsymbol{\xi} \rangle - \frac{\eta d^{-1/4}}{Kn_2} \sum_{t=0}^{T^{(2)}-1} \sum_{(k,i) \in \Psi} \ell_{k,i}'^{(t,J)} - \frac{\eta d^{-1/4}}{Kn_2} \sum_{t=T^{(2)}}^{T'-1} \sum_{(k,i) \in \Psi} \ell_{k,i}'^{(t,J)} \\
&\le \langle \mathbf{w}_{j,r}^{(0,0)}, \boldsymbol{\xi} \rangle + \eta d^{-1/4} T^{(2)} + \frac{\eta d^{-1/4}}{Kn_2} \widetilde{\mathcal{O}}(1) \\
&\le \langle \mathbf{w}_{j,r}^{(0,0)}, \boldsymbol{\xi} \rangle + \widetilde{\mathcal{O}}(d^{-1/4}) + \widetilde{\mathcal{O}}(d^{-1/4}) \\
&= \widetilde{\mathcal{O}}(d^{-1/4}),
\end{aligned} \tag{E.34}$$

where the second line is by Eq. (E.31). The third line is by $0 \le \sigma'(\cdot) \le 1$. The third last line is due to $-\ell'(\cdot) \le 1$, $T^{(2)} = \text{polylog}(d)$ and $\sum_{t=T^{(2)}}^{T-1} \sum_{(k,i) \in \Psi} \ell_{k,i}'^{(t,J)} = \widetilde{\mathcal{O}}(1)$. The second last line is by Eq. (E.32). We note that Eq. (E.34) holds for all $j \in \{-1,1\}$, $r \in [m]$, and $y \in \{-1,1\}$. Using the same procedure, one can show that $j\langle \mathbf{w}_{j,r}^{(T',0)}, \mathbf{z} \rangle \le \widetilde{\mathcal{O}}(d^{-1/4})$ for all $r \in [m]$ and $j \in \{-1,1\}$. Now without loss of generality assume $y = 1$. We have

$$\ell\Big(y\Big(F_{+1}(\mathbf{W}_{+1}^{(T')}, \mathbf{x}) - F_{-1}(\mathbf{W}_{-1}^{(T')}, \mathbf{x})\Big)\Big) = \ell\Big(F_{+1}(\mathbf{W}_{+1}^{(T')}, \mathbf{x}) - F_{-1}(\mathbf{W}_{-1}^{(T')}, \mathbf{x})\Big)$$

$$\geq \ell\Big(F_{+1}(\mathbf{W}_{+1}^{(T')}, \mathbf{x})\Big)$$

$$= \ell\bigg(\sum_{r=1}^{m} \sigma(\langle \mathbf{w}_{1,r}^{(T',0)}, \boldsymbol{\nu} + \mathbf{z}\rangle) + \sigma(\langle \mathbf{w}_{1,r}^{(T',0)}, \boldsymbol{\xi}\rangle)\bigg)$$

$$\geq \ell\bigg(\sum_{r=1}^{m} \sigma\big(\widetilde{\mathcal{O}}(d^{-1/4}) + \widetilde{\mathcal{O}}(d^{-1/4})\big) + \sigma\big(\widetilde{\mathcal{O}}(d^{-1/4})\big)\bigg)$$

$$= \ell\Big(m\widetilde{\mathcal{O}}(d^{-1/4})\Big)$$

$$= \Omega(1). \tag{E.35}$$

where second line is because $\ell(\cdot)$ is nonincreasing and the fourth line is due to Claim E.5. Finally plugging Eq. (E.35) into Eq. (E.33) gives that

$$\mathcal{L}_{\text{test}}(\mathbf{W}) \geq \mathbb{E}[\mathbf{1}(\mathcal{E})\ell\big(yf(\mathbf{W}, \mathbf{x})\big)] \geq \mathbb{E}[\mathbf{1}(\mathcal{E})\Omega(1)] = \Omega(\mathbb{P}(\mathcal{E})) = \Omega(1).$$

Since $T' \in [T^{(1)}, T]$ is arbitrary, we get that

$$\min_{t \in [T^{(1)}, T]} \mathcal{L}_{\text{test}}(\mathbf{W}^{(t)}) = \Omega(1).$$

However, the above proof would also work for $T' < T^{(1)}$ using our proofs from Phase I. Hence,

$$\min_{t \in [T]} \mathcal{L}_{\text{test}}(\mathbf{W}^{(t)}) = \Omega(1),$$

which concludes our theorem.

$\square$

## F  TRAIN-VALIDATION METHOD

Contrary to the train-train method, we will show in this section that under Condition D.1, the feature will be learned by our neural network. Abusing notations, let $\boldsymbol{\Xi}_{k,i}^{(t)} = \max\left\{\langle \mathbf{w}_{1,r}^{(t,0)}, \boldsymbol{\xi}_{k,i}\rangle : r \in [m]\right\}$, $\boldsymbol{\Gamma}_{k,i}^{(t)} = \max\left\{\langle \mathbf{w}_{-1,r}^{(t,0)}, \boldsymbol{\xi}_{k,i}\rangle : r \in [m]\right\}$, and $\boldsymbol{\Lambda}_j^{(t)} = \max\left\{j\langle \mathbf{w}_{j,r}^{(t,0)}, \boldsymbol{\nu}\rangle : r \in [m]\right\}$ for $j \in \{-1, 1\}$. Define $C = J\gamma\sigma_\xi^2\sqrt{d}\sqrt{\log(Kn)}$, $\widehat{\boldsymbol{\Xi}}_{k,i}^{(t)} = \max\{C, \boldsymbol{\Xi}_{k,i}^{(t)}\}$, and $\widehat{\boldsymbol{\Gamma}}_{k,i}^{(t)} = \max\{C, \boldsymbol{\Gamma}_{k,i}^{(t)}\}$. Let $\Psi = \{(k, i) : k \in [K], i > n_1\}$, $\Psi_+ = \{(k, i) : k \in [K], i > n_1, y_{k,i} = 1\}$, and $\Psi_- = \{(k, i) : k \in [K], i > n_1, y_{k,i} = -1\}$. So $\Psi$ represents all of the samples from the validation set of all tasks. And $\Psi_j$ represents the samples from the validation set of all tasks with label $j$. We first present a series of useful lemmas that hold for all $t \geq 0$. Theses lemmas basically show that the learning speed of $\mathbf{z}_k$ are very slow compared with the learning speed of the feature $\boldsymbol{\nu}$.

**Lemma F.1.** *Under Condition D.1 and Condition 4.1, for any $t \geq 0$, $k \in [M]$, $j \in \{-1, 1\}$, if one uses the train-validation method, then we have*

$$\max_r \left|j\langle \mathbf{w}_{j,r}^{(t,0)}, \mathbf{z}_k\rangle\right| = o(1)\boldsymbol{\Lambda}_j^{(t)}.$$

*Proof.* Without loss of generality, let $j = 1$, $k \in [M]$. Denote $r(t) = \arg\max_{r \in [m]} \left|\langle \mathbf{w}_{1,r}^{(t,0)}, \mathbf{z}_k\rangle\right|$. Define

$$D(t) := \frac{\left|\langle \mathbf{w}_{1,r(t)}^{(t,0)}, \mathbf{z}_k\rangle\right|}{\boldsymbol{\Lambda}_1^{(t)}}.$$

At $t = 0$, by Lemma D.2 we have

$$D(0) = \frac{\left|\langle \mathbf{w}_{1,r(0)}^{(0,0)}, \mathbf{z}_k\rangle\right|}{\boldsymbol{\Lambda}_1^{(0)}} \leq \frac{\Theta(\sigma_s\sigma_0\sqrt{d}\sqrt{\log(mKn)})}{\Theta(\sigma_0)} = \Theta(\sigma_s\sqrt{d}\sqrt{\log(mKn)}) = o(1).$$

Define $D_0 := \Theta(\sigma_s \sqrt{d} \sqrt{\log(mKn)})$. We show that for any $t' > 0$, we have $D(t') \leq 2D_0 = o(1)$. Observe that similar to Eq. (E.17), we have for any $t \geq 0$,

$$\langle \mathbf{w}_{1,r}^{(t+1,0)}, \mathbf{z}_k \rangle = \langle \mathbf{w}_{1,r}^{(t,0)}, \mathbf{z}_k \rangle - \frac{\eta}{Kn_2} \sum_{(k',i) \in \Psi} \ell_{k',i}'^{(t,J)} \sigma'(\langle \mathbf{w}_{1,r}^{(t,J)}, \boldsymbol{\nu}_{k'} \rangle y_{k',i}) \langle \mathbf{z}_{k'}, \mathbf{z}_k \rangle$$

$$\leq \langle \mathbf{w}_{1,r}^{(t,0)}, \mathbf{z}_k \rangle - \frac{\eta}{Kn_2} \sum_{(k',i) \in \Psi} \ell_{k',i}'^{(t,J)} \sigma'(\langle \mathbf{w}_{1,r}^{(t,J)}, \boldsymbol{\nu}_{k'} \rangle y_{k',i}) \Theta(\sigma_s^2 d), \qquad \text{(F.1)}$$

where we used Condition 4.1 for the inequality, since $\langle \mathbf{z}_{k'}, \mathbf{z}_k \rangle \leq \max(\|\mathbf{z}_k\|_2^2, \|\mathbf{z}_{k'}\|_2^2) = \Theta(\sigma_s^2 d)$. Comparing Eq. (C.11) and Eq. (F.1), we can roughly understand why this lemma is true: for any $r \in [m]$, the growth of $\langle \mathbf{w}_{1,r}^{(t,0)}, \boldsymbol{\nu} \rangle$ is at least a factor of $1/(\sigma_s^2 d) = \omega(1)$ faster than that of $\langle \mathbf{w}_{1,r}^{(t,0)}, \mathbf{z}_k \rangle$. We can compute

$$\left| \langle \mathbf{w}_{1,r(t')}^{(t',0)}, \mathbf{z}_k \rangle \right| \leq \left| \sum_{t=0}^{t'-1} \left( \langle \mathbf{w}_{1,r(t')}^{(t+1,0)}, \mathbf{z}_k \rangle - \langle \mathbf{w}_{1,r(t')}^{(t,0)}, \mathbf{z}_k \rangle \right) \right| + \left| \langle \mathbf{w}_{1,r(t')}^{(0,0)}, \mathbf{z}_k \rangle \right|$$

$$\leq \sum_{t=0}^{t'-1} \left| \langle \mathbf{w}_{1,r(t')}^{(t+1,0)}, \mathbf{z}_k \rangle - \langle \mathbf{w}_{1,r(t')}^{(t,0)}, \mathbf{z}_k \rangle \right| + \left| \langle \mathbf{w}_{1,r(t')}^{(0,0)}, \mathbf{z}_k \rangle \right|$$

$$\leq \sum_{t=0}^{t'-1} \Theta(\sigma_s^2 d) \left| \langle \mathbf{w}_{1,r(t')}^{(t+1,0)}, \boldsymbol{\nu} \rangle - \langle \mathbf{w}_{1,r(t')}^{(t,0)}, \boldsymbol{\nu} \rangle \right| + \left| \langle \mathbf{w}_{1,r(t')}^{(0,0)}, \mathbf{z}_k \rangle \right|$$

$$= \sum_{t=0}^{t'-1} \Theta(\sigma_s^2 d) \left( \langle \mathbf{w}_{1,r(t')}^{(t+1,0)}, \boldsymbol{\nu} \rangle - \langle \mathbf{w}_{1,r(t')}^{(t,0)}, \boldsymbol{\nu} \rangle \right) + \left| \langle \mathbf{w}_{1,r(t')}^{(0,0)}, \mathbf{z}_k \rangle \right|$$

$$= \Theta(\sigma_s^2 d) \left( \langle \mathbf{w}_{1,r(t')}^{(t',0)}, \boldsymbol{\nu} \rangle - \langle \mathbf{w}_{1,r(t')}^{(0,0)}, \boldsymbol{\nu} \rangle \right) + \left| \langle \mathbf{w}_{1,r(t')}^{(0,0)}, \mathbf{z}_k \rangle \right|$$

$$\leq \Theta(\sigma_s^2 d) \left( \boldsymbol{\Lambda}_1^{(t')} - \min_{r \in [m]} \langle \mathbf{w}_{1,r}^{(0,0)}, \boldsymbol{\nu} \rangle \right) + \left| \langle \mathbf{w}_{1,r(0)}^{(0,0)}, \mathbf{z}_k \rangle \right|$$

$$\leq \Theta(\sigma_s^2 d) \left( \boldsymbol{\Lambda}_1^{(t')} + \Theta(\sigma_0 \sqrt{\log(mKn)}) \right) + D_0 \boldsymbol{\Lambda}_1^{(0)},$$

where the first and second lines are by triangle inequality. The third line is by comparing Eq. (C.11) and Eq. (F.1). The fourth line is by the fact that $\langle \mathbf{w}_{1,r}^{(t,0)}, \boldsymbol{\nu} \rangle$ is increasing in $t$ for any $r \in [m]$. The sixth line is by definition of $\boldsymbol{\Lambda}_1^{(t')}$ and definition of $r(0)$. The last line is by Lemma D.2. Then we have

$$\frac{\left| \langle \mathbf{w}_{1,r(t')}^{(t',0)}, \mathbf{z}_k \rangle \right|}{\boldsymbol{\Lambda}_1^{(t')}} \leq \frac{\Theta(\sigma_s^2 d) \left( \boldsymbol{\Lambda}_1^{(t')} + \Theta(\sigma_0 \sqrt{\log(mKn)}) \right) + D_0 \boldsymbol{\Lambda}_1^{(0)}}{\boldsymbol{\Lambda}_1^{(t')}}$$

$$\leq \Theta(\sigma_s^2 d)(1 + \sqrt{\log(mKn)}) + D_0$$

$$< 2D_0,$$

where we used $\boldsymbol{\Lambda}_j^{(t)}$ is increasing in $t$ to get the second inequality. The third inequality is by direct calculation and comparison: $D_0 = \Theta(\sigma_s \sqrt{d} \sqrt{\log(mKn)})$, $\sigma_s^2 d = (\sigma_s \sqrt{d})^2 = \sigma_s \sqrt{d}/\text{polylog}(d) = o(1)D_0$, $\sigma_s^2 d \sqrt{\log(mKn)} = \sigma_s \sqrt{\log(mKn)}/\text{polylog}(d) = o(1)D_0$. This proves the lemma. $\qquad \square$

**Lemma F.2.** *Under Condition D.1 and Condition 4.1, for any $j \in \{-1, 1\}$, $r \in [m]$, if at time $t$ we have $j \langle \mathbf{w}_{j,r}^{(t,0)}, \boldsymbol{\nu} \rangle \geq d^{-1/3}$, then for any $k \in [K]$*

$$j \langle \mathbf{w}_{j,r}^{(t,0)}, \mathbf{z}_k \rangle = \mathcal{O}(\sigma_s^2 d) j \langle \mathbf{w}_{j,r}^{(t,0)}, \boldsymbol{\nu} \rangle = o(1) j \langle \mathbf{w}_{j,r}^{(t,0)}, \boldsymbol{\nu} \rangle .$$

*Proof.* Without loss of generality, assume $j = 1$. And suppose $\langle \mathbf{w}_{1,r}^{(t,0)}, \boldsymbol{\nu} \rangle \geq d^{-1/3}$ for some $r \in [m]$. By Eq. (C.11), we have

$$d^{-1/3} \leq \langle \mathbf{w}_{1,r}^{(t,0)}, \boldsymbol{\nu} \rangle = \langle \mathbf{w}_{1,r}^{(0,0)}, \boldsymbol{\nu} \rangle - \frac{\eta}{Kn_2} \sum_{t'=0}^{t-1} \sum_{(k,i) \in \Psi} \ell_{k,i}'^{(t',J)} \sigma'(\langle \mathbf{w}_{j,r}^{(t',J)}, \boldsymbol{\nu}_k \rangle y_{k,i}) \|\boldsymbol{\nu}\|_2^2 ,$$

$$d^{-1/3} - \langle \mathbf{w}_{1,r}^{(0,0)}, \boldsymbol{\nu} \rangle \leq -\frac{\eta}{Kn_2} \sum_{t'=0}^{t-1} \sum_{(k,i)\in\Psi} \ell_{k,i}'^{(t',J)} \sigma'(\langle \mathbf{w}_{j,r}^{(t',J)}, \boldsymbol{\nu}_k \rangle y_{k,i}) \|\boldsymbol{\nu}\|_2^2 \,,$$

$$d^{-1/3} - \mathcal{O}(\sigma_0 \sqrt{\log(mKn)}) \leq -\frac{\eta}{Kn_2} \sum_{t'=0}^{t-1} \sum_{(k,i)\in\Psi} \ell_{k,i}'^{(t',J)} \sigma'(\langle \mathbf{w}_{j,r}^{(t',J)}, \boldsymbol{\nu}_k \rangle y_{k,i}) \|\boldsymbol{\nu}\|_2^2 \,,$$

$$\Omega(d^{-1/3}) \leq -\frac{\eta}{Kn_2} \sum_{t'=0}^{t-1} \sum_{(k,i)\in\Psi} \ell_{k,i}'^{(t',J)} \sigma'(\langle \mathbf{w}_{j,r}^{(t',J)}, \boldsymbol{\nu}_k \rangle y_{k,i}) \|\boldsymbol{\nu}\|_2^2 \,.$$

Recall that from Condition D.1 and Condition 4.1

$$\left| \langle \mathbf{w}_{1,r}^{(0,0)}, \boldsymbol{\nu} \rangle \right| \leq \widetilde{\mathcal{O}}(d^{-1/2}) \ll \mathcal{O}(d^{-1/3}) \leq \frac{-\eta}{Kn_2} \sum_{t'=0}^{t-1} \sum_{(k,i)\in\Psi} \ell_{k,i}'^{(t',J)} \sigma'(\langle \mathbf{w}_{j,r}^{(t',J)}, \boldsymbol{\nu}_k \rangle y_{k,i}) \|\boldsymbol{\nu}\|_2^2 \,.$$

This implies

$$\left| \langle \mathbf{w}_{1,r}^{(0,0)}, \boldsymbol{\nu} \rangle \right| = -o(1) \frac{\eta}{Kn_2} \sum_{t'=0}^{t-1} \sum_{(k,i)\in\Psi} \ell_{k,i}'^{(t',J)} \sigma'(\langle \mathbf{w}_{j,r}^{(t',J)}, \boldsymbol{\nu}_k \rangle y_{k,i}) \|\boldsymbol{\nu}\|_2^2 \,,$$

Hence,

$$\langle \mathbf{w}_{1,r}^{(t,0)}, \boldsymbol{\nu} \rangle = -\Theta(1) \frac{\eta}{Kn_2} \sum_{t'=0}^{t-1} \sum_{(k,i)\in\Psi} \ell_{k,i}'^{(t',J)} \sigma'(\langle \mathbf{w}_{j,r}^{(t',J)}, \boldsymbol{\nu}_k \rangle y_{k,i}) \|\boldsymbol{\nu}\|_2^2 \,. \tag{F.2}$$

On the other hand, from Eq. (F.1) and Lemma D.2 we have

$$\langle \mathbf{w}_{1,r}^{(t,0)}, \mathbf{z}_k \rangle \leq \langle \mathbf{w}_{1,r}^{(0,0)}, \mathbf{z}_k \rangle - \frac{\eta}{Kn_2} \sum_{t'=0}^{t-1} \sum_{(k,i)\in\Psi} \ell_{k,i}'^{(t',J)} \sigma'(\langle \mathbf{w}_{j,r}^{(t',J)}, \boldsymbol{\nu}_k \rangle y_{k,i}) \Theta(\sigma_s^2 d)$$

$$\leq \mathcal{O}(\sigma_s \sigma_0 \sqrt{d} \sqrt{\log(mKn)}) - \frac{\eta}{Kn_2} \sum_{t'=0}^{t-1} \sum_{(k,i)\in\Psi} \ell_{k,i}'^{(t',J)} \sigma'(\langle \mathbf{w}_{j,r}^{(t',J)}, \boldsymbol{\nu}_k \rangle y_{k,i}) \Theta(\sigma_s^2 d)$$

$$\leq -\mathcal{O}(1) \frac{\eta}{Kn_2} \sum_{t'=0}^{t-1} \sum_{(k,i)\in\Psi} \ell_{k,i}'^{(t',J)} \sigma'(\langle \mathbf{w}_{j,r}^{(t',J)}, \boldsymbol{\nu}_k \rangle y_{k,i}) \Theta(\sigma_s^2 d)$$

$$= -\mathcal{O}(\sigma_s^2 d) \frac{\eta}{Kn_2} \sum_{t'=0}^{t-1} \sum_{(k,i)\in\Psi} \ell_{k,i}'^{(t',J)} \sigma'(\langle \mathbf{w}_{j,r}^{(t',J)}, \boldsymbol{\nu}_k \rangle y_{k,i})$$

$$= \mathcal{O}(\sigma_s^2 d) \langle \mathbf{w}_{1,r}^{(t,0)}, \boldsymbol{\nu} \rangle$$

$$= o(1) \langle \mathbf{w}_{1,r}^{(t,0)}, \boldsymbol{\nu} \rangle \,,$$

where the third line is a direct comparison of the sizes of the first and second term of the second line. The second last equality is due to Eq. (F.2). □

**Lemma F.3.** *Under Condition D.1 and Condition 4.1, for any $j \in \{-1,1\}$, $r \in [m]$, if at time $t$ we have $j\langle \boldsymbol{w}_{j,r}^{(t,0)}, \boldsymbol{\nu} \rangle < d^{-1/3}$, then for any $k \in [K]$*

$$j\langle \boldsymbol{w}_{j,r}^{(t,0)}, \mathbf{z}_k \rangle < d^{-1/3} \,.$$

*Proof.* Without loss of generality, consider $j = 1$. Let $k \in [K]$, $r \in [m]$ and suppose $\langle \mathbf{w}_{1,r}^{(t,0)}, \boldsymbol{\nu} \rangle < d^{-1/3}$ at time $t$. By Eq. (C.11), we have

$$d^{-1/3} > \langle \mathbf{w}_{1,r}^{(t,0)}, \boldsymbol{\nu} \rangle = \langle \mathbf{w}_{1,r}^{(0,0)}, \boldsymbol{\nu} \rangle - \frac{\eta}{Kn_2} \sum_{t'=0}^{t-1} \sum_{(k,i)\in\Psi} \ell_{k,i}'^{(t',J)} \sigma'(\langle \mathbf{w}_{j,r}^{(t',J)}, \boldsymbol{\nu}_k \rangle y_{k,i}) \|\boldsymbol{\nu}\|_2^2 \,,$$

$$d^{-1/3} - \langle \mathbf{w}_{1,r}^{(0,0)}, \boldsymbol{\nu} \rangle > -\frac{\eta}{Kn_2} \sum_{t'=0}^{t-1} \sum_{(k,i) \in \Psi} \ell_{k,i}'^{(t',J)} \sigma'(\langle \mathbf{w}_{j,r}^{(t',J)}, \boldsymbol{\nu}_k \rangle y_{k,i}) \|\boldsymbol{\nu}\|_2^2 \;,$$

$$d^{-1/3} - \min_{r' \in [m]} \langle \mathbf{w}_{1,r'}^{(0,0)}, \boldsymbol{\nu} \rangle > -\frac{\eta}{Kn_2} \sum_{t'=0}^{t-1} \sum_{(k,i) \in \Psi} \ell_{k,i}'^{(t',J)} \sigma'(\langle \mathbf{w}_{j,r}^{(t',J)}, \boldsymbol{\nu}_k \rangle y_{k,i}) \|\boldsymbol{\nu}\|_2^2 \;,$$

$$d^{-1/3} + \mathcal{O}(d^{-1/2}) > -\frac{\eta}{Kn_2} \sum_{t'=0}^{t-1} \sum_{(k,i) \in \Psi} \ell_{k,i}'^{(t',J)} \sigma'(\langle \mathbf{w}_{j,r}^{(t',J)}, \boldsymbol{\nu}_k \rangle y_{k,i}) \|\boldsymbol{\nu}\|_2^2 \;,$$

$$2d^{-1/3} > -\frac{\eta}{Kn_2} \sum_{t'=0}^{t-1} \sum_{(k,i) \in \Psi} \ell_{k,i}'^{(t',J)} \sigma'(\langle \mathbf{w}_{j,r}^{(t',J)}, \boldsymbol{\nu}_k \rangle y_{k,i}) \|\boldsymbol{\nu}\|_2^2 \;, \tag{F.3}$$

where the fourth line is by Lemma D.2. Meanwhile, using Eq. (F.1) and Lemma D.2 we obtain

$$\langle \mathbf{w}_{1,r}^{(t,0)}, \mathbf{z}_k \rangle \le \langle \mathbf{w}_{1,r}^{(0,0)}, \mathbf{z}_k \rangle - \frac{\eta}{Kn_2} \sum_{t'=0}^{t-1} \sum_{(k',i) \in \Psi} \ell_{k',i}'^{(t,J)} \sigma'(\langle \mathbf{w}_{1,r}^{(t,J)}, \boldsymbol{\nu}_{k'} \rangle y_{k',i}) \Theta(\sigma_s^2 d)$$

$$\le \langle \mathbf{w}_{1,r}^{(0,0)}, \mathbf{z}_k \rangle + 2d^{-1/3} \Theta(\sigma_s^2 d)$$

$$\le \mathcal{O}(\sigma_0 \sigma_s \sqrt{d} \sqrt{\log(mKn)}) + 2d^{-1/3} \Theta(\sigma_s^2 d)$$

$$< d^{-1/3} \;,$$

where the second line is by Eq. (F.3). The last inequality is because the first term in the second last line is $\widetilde{\mathcal{O}}(d^{-1/2})$ and $\Theta(\sigma_s^2 d) = o(1)$. This proves our lemma.

$\square$

**Lemma F.4.** *Under Condition D.1 and Condition 4.1, for any $j \in \{-1, 1\}$, $r \in [m]$ and $k \in [K]$ if at time $t$ we have $j\langle \mathbf{w}_{j,r}^{(t,0)}, \mathbf{z}_k \rangle = \mathcal{O}(\sigma_s^2 d) j \langle \mathbf{w}_{j,r}^{(t,0)}, \boldsymbol{\nu} \rangle$, then*

$$j\langle \mathbf{w}_{j,r}^{(t,J)}, \mathbf{z}_k \rangle = \mathcal{O}(\sigma_s^2 d) j \langle \mathbf{w}_{j,r}^{(t,J)}, \boldsymbol{\nu} \rangle \;,$$

*where $\mathbf{w}_{j,r}^{(t,J)}$ on both hand sides refer to the weights $\mathbf{w}_{j,r}^{(t,0)}$ after $J$ steps inner-loop updates using samples from $\mathcal{S}_k^{tr}$.*

*Proof.* Without loss of generality, let $j = 1$. Using samples from $\mathcal{S}_k^{tr}$, similar to Eq. (C.5) we have

$$\langle \mathbf{w}_{1,r}^{(t,\tau+1)}, \boldsymbol{\nu} \rangle = \langle \mathbf{w}_{1,r}^{(t,\tau)}, \boldsymbol{\nu} \rangle - \frac{\gamma}{n_1} \sum_{i=1}^{n_1} \ell_{k,i}'^{(t,\tau)} \sigma'(\langle \mathbf{w}_{j,r}^{(t,\tau)}, \boldsymbol{\nu}_k \rangle y_{k,i}) \|\boldsymbol{\nu}\|_2^2 \;. \tag{F.4}$$

And

$$\langle \mathbf{w}_{1,r}^{(t,\tau+1)}, \mathbf{z}_k \rangle = \langle \mathbf{w}_{1,r}^{(t,\tau)}, \mathbf{z}_k \rangle - \frac{\gamma}{n_1} \sum_{i=1}^{n_1} \ell_{k,i}'^{(t,\tau)} \sigma'(\langle \mathbf{w}_{j,r}^{(t,\tau)}, \boldsymbol{\nu}_k \rangle y_{k,i}) \|\mathbf{z}_k\|_2^2 \;.$$

This implies

$$\langle \mathbf{w}_{1,r}^{(t,J)}, \boldsymbol{\nu} \rangle = \langle \mathbf{w}_{1,r}^{(t,0)}, \boldsymbol{\nu} \rangle - \frac{\gamma}{n_1} \sum_{\tau=0}^{J-1} \sum_{i=1}^{n_1} \ell_{k,i}'^{(t,\tau)} \sigma'(\langle \mathbf{w}_{j,r}^{(t,\tau)}, \boldsymbol{\nu}_k \rangle y_{k,i}) \|\boldsymbol{\nu}\|_2^2 \;,$$

$$\langle \mathbf{w}_{1,r}^{(t,J)}, \mathbf{z}_k \rangle = \langle \mathbf{w}_{1,r}^{(t,0)}, \mathbf{z}_k \rangle - \frac{\gamma}{n_1} \sum_{\tau=0}^{J-1} \sum_{i=1}^{n_1} \ell_{k,i}'^{(t,\tau)} \sigma'(\langle \mathbf{w}_{j,r}^{(t,\tau)}, \boldsymbol{\nu}_k \rangle y_{k,i}) \|\mathbf{z}_k\|_2^2 \;.$$

Recall $\|\boldsymbol{\nu}\|_2^2 = 1$ and $\|\mathbf{z}_k\|_2^2 = \mathcal{O}(\sigma_s^2 d)$ by Condition 4.1. Since we are given $\langle \mathbf{w}_{1,r}^{(t,0)}, \mathbf{z}_k \rangle = \mathcal{O}(\sigma_s^2 d) \langle \mathbf{w}_{1,r}^{(t,0)}, \boldsymbol{\nu} \rangle$, we conclude that

$$\langle \mathbf{w}_{1,r}^{(t,J)}, \mathbf{z}_k \rangle = \mathcal{O}(\sigma_s^2 d) \langle \mathbf{w}_{1,r}^{(t,J)}, \boldsymbol{\nu} \rangle \;.$$

$\square$

The following corollary is an immediate result of Lemma F.2 and Lemma F.4.

**Corollary F.5.** *Fix $j \in \{-1, 1\}$. Then for any $r \in [m]$, if at time $t$ we have $j\langle w_{j,r}^{(t,0)}, \nu \rangle \geq d^{-1/3}$, then for any $k \in [K]$,*

$$j\langle w_{j,r}^{(t,J)}, z_k \rangle = \mathcal{O}(\sigma_s^2 d) j\langle w_{j,r}^{(t,J)}, \nu \rangle = \mathcal{O}(1/\text{polylog}(d)) j\langle w_{j,r}^{(t,J)}, \nu \rangle \,,$$

*where $w_{j,r}^{(t,J)}$ on both hand sides refer to the weights $w_{j,r}^{(t,0)}$ after $J$ steps inner-loop updates using samples from $\mathcal{S}_k^{tr}$.*

**Lemma F.6.** *Under Condition D.1 and Condition 4.1, for any $j \in \{-1, 1\}$, $r \in [m]$, and $k \in [K]$, if we are using the samples from $\mathcal{S}_k^{tr}$, then for any $t \geq 0$ we have*

$$j\langle w_{j,r}^{(t,J)}, \nu \rangle \leq 2 \max \left\{ \Lambda_j^{(t)}, J\gamma \right\} \,.$$

*Proof.* Without loss of generality, consider $j = 1$. Let $r \in [m]$ and $k \in [K]$. Using samples from $\mathcal{S}_k^{tr}$, by Eq. (F.4) we have

$$\langle \mathbf{w}_{1,r}^{(t,\tau+1)}, \nu \rangle \leq \langle \mathbf{w}_{1,r}^{(t,\tau)}, \nu \rangle + \gamma \,,$$

for any $\tau \in [J-1]$, where we have used the fact that $-\ell'(\cdot) \leq 1$ and $\sigma'(\cdot) \leq 1$. Repeating $J$ times we get

$$\langle \mathbf{w}_{1,r}^{(t,J)}, \nu \rangle \leq \langle \mathbf{w}_{1,r}^{(t,0)}, \nu \rangle + J\gamma \leq \Lambda_1^{(t)} + J\gamma \leq 2 \max \left\{ \Lambda_1^{(t)}, J\gamma \right\} \,.$$

$\square$

### F.1 PHASE I

Let $T_j^{(2)}$ be the first iteration such that $\Lambda_j^{(t)} \geq m^{-1/2} (1+\gamma)^{-J} = \widetilde{\Theta}(1)$ and let $T^{(2)} = \max_j T_j^{(2)}$. Without loss of generality, let us assume that $T_{-1}^{(2)} \leq T_1^{(2)}$.

Recall that we assumed $T_1^{(2)} \geq T_{-1}^{(2)}$.

**Lemma F.7** (Restatement of Lemma 6.2). *Under Condition D.1 and Condition 4.1, if one uses the train-validation method, then for any $t \leq T^{(2)}$,*

*1. For any $k \in [K]$ and $r = \arg\max_{r' \in [m]} \langle w_{1,r'}^{(t,0)}, \nu \rangle$*

$$\langle w_{1,r}^{(t,J)}, \nu_k \rangle = \Omega(1) \langle w_{1,r}^{(t,0)}, \nu \rangle (1 + \Theta(1)\gamma)^J \,.$$

*2. For any $r \in [m]$, $(k, i) \in \Psi$, and $j \in \{-1, 1\}$*

$$\langle w_{j,r}^{(t,J)}, \xi_{k,i} \rangle \leq \mathcal{O}\left( \max \left\{ \langle w_{j,r}^{(t,0)}, \xi_{k,i} \rangle, C \right\} \right) \,.$$

*Proof.* The proof of the first part relies on the following hypothesis which we will verify inductively later.

$$- \ell_{k,i}'^{(t,\tau)} = \Theta(1), \text{ for all } \tau \in [J], k \in [K], i \in [n] \text{ such that } y_{k,i} = 1 \,. \tag{F.5}$$

$$\sigma'(\langle \mathbf{w}_{1,r}^{(t,\tau)}, \nu_k \rangle) = \Theta(1) \langle \mathbf{w}_{1,r}^{(t,\tau)}, \nu_k \rangle, \text{ for all } \tau \in [J], k \in [K] \text{ if } \langle \mathbf{w}_{1,r}^{(t,\tau)}, \nu_k \rangle > 0 \,. \tag{F.6}$$

By Eq. (C.5), we know $\langle \mathbf{w}_{1,r}^{(t,\tau)}, \nu_k \rangle$ is increasing in $\tau$ for any $r \in [m]$ and $k \in [K]$. Moreover, we have

$$\langle \mathbf{w}_{1,r}^{(t+1,0)}, \nu_k \rangle = \langle \mathbf{w}_{1,r}^{(t,0)}, \nu_k \rangle - \frac{\eta}{Kn} \sum_{(k',i) \in \Psi} \ell_{k',i}'^{(t,J)} \sigma'(\langle \mathbf{w}_{1,r}^{(t,J)}, \nu_{k'} \rangle y_{k',i}) \langle \nu_k, \nu_{k'} \rangle \,,$$

which shows that $\langle \mathbf{w}_{1,r}^{(t,0)}, \nu_k \rangle$ is increasing in $t$ for all $r \in [m]$, $k \in [K]$, since $\langle \nu_k, \nu_{k'} \rangle = \|\nu\|_2^2 + \langle \mathbf{z}_k, \mathbf{z}_{k'} \rangle > 0$. And Eq. (C.11) shows that $\langle \mathbf{w}_{1,r}^{(t,0)}, \nu \rangle$ is increasing in $t$ for all $r \in [m]$. Let $r(t) = \arg\max_{r \in [m]} \langle \mathbf{w}_{1,r}^{(t,0)}, \nu \rangle$. Recall that under Lemma D.2, we have $\langle \mathbf{w}_{1,r(0)}^{(0,0)}, \nu \rangle = \Omega(\sigma_0) > 0$. Since this is increasing in $t$, we have

$$\langle \mathbf{w}_{1,r(t)}^{(t,0)}, \nu \rangle \geq \langle \mathbf{w}_{1,r(0)}^{(t,0)}, \nu \rangle \geq \langle \mathbf{w}_{1,r(0)}^{(0,0)}, \nu \rangle > 0 \,,$$

where the first inequality is by definition of $r(t)$. By Lemma F.1, we have for any $k \in [K]$,

$$\langle \mathbf{w}_{1,r(t)}^{(t,0)}, \boldsymbol{\nu}_k \rangle = \langle \mathbf{w}_{1,r(t)}^{(t,0)}, \boldsymbol{\nu} \rangle + \langle \mathbf{w}_{1,r(t)}^{(t,0)}, \mathbf{z}_k \rangle \geq (1 - o(1))\langle \mathbf{w}_{1,r(t)}^{(t,0)}, \boldsymbol{\nu} \rangle > 0 \,.$$

And by monotonicity in $\tau$, we have $\langle \mathbf{w}_{1,r(t)}^{(t,\tau)}, \boldsymbol{\nu}_k \rangle > 0$ for all $\tau \in [J]$. From Eq. (C.5) we can compute

$$\begin{aligned}
\langle \mathbf{w}_{1,r(t)}^{(t,\tau+1)}, \boldsymbol{\nu}_k \rangle &= \langle \mathbf{w}_{1,r(t)}^{(t,\tau)}, \boldsymbol{\nu}_k \rangle - \frac{\gamma}{n_1} \sum_{i \leq n_1, y_{k,i}=1} \ell_{k,i}^{\prime(t,\tau)} \sigma'(\langle \mathbf{w}_{1,r(t)}^{(t,\tau)}, \boldsymbol{\nu}_k \rangle) \|\boldsymbol{\nu}_k\|_2^2 \\
&= \langle \mathbf{w}_{1,r(t)}^{(t,\tau)}, \boldsymbol{\nu}_k \rangle + \gamma \Theta(1)\langle \mathbf{w}_{1,r(t)}^{(t,\tau)}, \boldsymbol{\nu}_k \rangle \\
&= \langle \mathbf{w}_{1,r(t)}^{(t,\tau)}, \boldsymbol{\nu}_k \rangle(1 + \Theta(1)\gamma) \,.
\end{aligned}$$

where we have used Hypothesis F.5 F.6 and $\|\boldsymbol{\nu}_k\|_2 = \Theta(1)$ to get the second line. Applying this repeatedly to get

$$\begin{aligned}
\langle \mathbf{w}_{1,r(t)}^{(t,J)}, \boldsymbol{\nu}_k \rangle &= \langle \mathbf{w}_{1,r(t)}^{(t,0)}, \boldsymbol{\nu}_k \rangle(1 + \Theta(1)\gamma)^J \\
&= \Omega(1)\langle \mathbf{w}_{1,r(t)}^{(t,0)}, \boldsymbol{\nu} \rangle(1 + \Theta(1)\gamma)^J \,,
\end{aligned} \tag{F.7}$$

where we have used Lemma F.1 to get the second equality. This proves the first part of the lemma. Consider some $(k,i) \in \Psi$. Without loss of generality we will prove the second part of the lemma for $j = 1$, since the proof for $j = -1$ is the same. By Eq. (C.3), for any $r \in [m]$ we have

$$\langle \mathbf{w}_{1,r}^{(t,\tau+1)}, \boldsymbol{\xi}_{k,i} \rangle \leq \langle \mathbf{w}_{1,r}^{(t,\tau)}, \boldsymbol{\xi}_{k,i} \rangle + \gamma \Theta(1)\Theta(\sigma_\xi^2 \sqrt{d}\sqrt{\log(Kn)}) \,,$$

where we used $-\ell'(\cdot) \leq 1$, $\sigma'(\cdot) \leq 1$ and Lemma D.2. This implies

$$\begin{aligned}
\langle \mathbf{w}_{1,r}^{(t,J)}, \boldsymbol{\xi}_{k,i} \rangle &\leq \langle \mathbf{w}_{1,r}^{(t,0)}, \boldsymbol{\xi}_{k,i} \rangle + J\gamma \Theta(1)\Theta(\sigma_\xi^2 \sqrt{d}\sqrt{\log(Kn)}) \\
&\leq \mathcal{O}\left( \max\left\{ \langle \mathbf{w}_{1,r}^{(t,0)}, \boldsymbol{\xi}_{k,i} \rangle, C \right\} \right) \,.
\end{aligned} \tag{F.8}$$

The second inequality is by definition of $C$. This proves the second part of the lemma. $\square$

*Remark* F.8. Eq. (F.8) is not a surprising result. Since $\boldsymbol{\xi}_{k,i}$ comes from the validation set and the inner-loop updates only uses samples from the training set, we expect that after $J$ steps of inner-loop updates, the inner product should remain as the same order as before the $J$ steps inner-loop updates.

**Theorem F.9** (Restatement of Lemma 6.3). *Under Condition D.1 and Condition 4.1, if one uses the train-validation method, then for any $t \leq T^{(2)}$, $(k,i) \in \Psi$,*

$$\widehat{\Xi}_{k,i}^{(t)} \leq \mathcal{O}(1)\widehat{\Xi}_{k,i}^{(0)} = \widetilde{\mathcal{O}}(d^{-1/2}) \,.$$

*And*

$$\widehat{\Gamma}_{k,i}^{(t)} \leq \mathcal{O}(1)\widehat{\Gamma}_{k,i}^{(0)} = \widetilde{\mathcal{O}}(d^{-1/2}) \,.$$

*Proof.* We will only prove our first statement that involves $\widehat{\Xi}_{k,i}^{(t)}$, because the proof of the second statement that involves $\widehat{\Gamma}_{k,i}^{(t)}$ will be exactly the same.

Similar to the proof of Theorem E.2, we will give lower bound on the growth of $\Lambda_1^{(t)}$ and upper bound on the growth of $\widehat{\Xi}_{k,i}^{(t)}$. We can upper bound the growth of $\Lambda_1^{(t)}$ using Lemma F.7. We first note that since $\langle \mathbf{w}_{1,r(t)}^{(t,J)}, \boldsymbol{\nu}_k \rangle > 0$, we have $-\langle \mathbf{w}_{1,r(t)}^{(t,J)}, \boldsymbol{\nu}_k \rangle < 0$. Thus, if $(k,i) \in \Psi_-$ we have $\sigma'(\langle \mathbf{w}_{1,r(t)}^{(t,J)}, \boldsymbol{\nu}_k \rangle y_{k,i}) = 0$. Then by Eq. (C.11), we get

$$\begin{aligned}
\Lambda_1^{(t+1)} &\geq \langle \mathbf{w}_{1,r(t)}^{(t+1,0)}, \boldsymbol{\nu} \rangle = \langle \mathbf{w}_{1,r(t)}^{(t,0)}, \boldsymbol{\nu} \rangle - \frac{\eta}{Kn_2} \sum_{(k,i) \in \Psi} \ell_{k,i}^{\prime(t,J)} \sigma'(\langle \mathbf{w}_{1,r(t)}^{(t,J)}, \boldsymbol{\nu}_k \rangle y_{k,i}) \\
&= \langle \mathbf{w}_{1,r(t)}^{(t,0)}, \boldsymbol{\nu} \rangle - \frac{\eta}{Kn_2} \sum_{(k,i) \in \Psi_+} \ell_{k,i}^{\prime(t,J)} \Theta(1)\langle \mathbf{w}_{1,r(t)}^{(t,J)}, \boldsymbol{\nu}_k \rangle
\end{aligned}$$

$$
\begin{aligned}
&= \langle \mathbf{w}_{1,r(t)}^{(t,0)}, \boldsymbol{\nu} \rangle + \eta \Omega(1) \langle \mathbf{w}_{1,r(t)}^{(t,0)}, \boldsymbol{\nu} \rangle (1 + \Theta(1)\gamma)^J \\
&= \langle \mathbf{w}_{1,r(t)}^{(t,0)}, \boldsymbol{\nu} \rangle (1 + \Omega(1)\eta\gamma^J) \\
&= \boldsymbol{\Lambda}_1^{(t)} (1 + \Omega(1)\eta\gamma^J),
\end{aligned}
\tag{F.9}
$$

where we used $\sigma'(\langle \mathbf{w}_{1,r(t)}^{(t,J)}, \boldsymbol{\nu}_k \rangle y_{k,i}) = 0$ if $(k,i) \in \Psi_-$, and Hypothesis F.5 F.6 to get the second equality. We used Lemma F.7 to get the third equality. Next, let $(k,i) \in \Psi$ and $r \in [m]$. We will derive an upper bound on the growth of $\widehat{\boldsymbol{\Xi}}_{k,i}^{(t)}$. By Eq. (C.9), we have

$$
\begin{aligned}
\langle \mathbf{w}_{1,r}^{(t+1,0)}, \boldsymbol{\xi}_{k,i} \rangle &\le \langle \mathbf{w}_{1,r}^{(t,0)}, \boldsymbol{\xi}_{k,i} \rangle + \frac{\eta}{Kn_2} \sum_{(k',i') \in \Psi \setminus \{(k,i)\}} \Theta(\sigma_\xi^2 \sqrt{d}\sqrt{\log(Kn)}) \\
&\quad + \frac{\eta}{Kn_2} \sigma'(\langle \mathbf{w}_{1,r}^{(t,J)}, \boldsymbol{\xi}_{k,i} \rangle) \Theta(\sigma_\xi^2 d) \\
&\le \langle \mathbf{w}_{1,r}^{(t,0)}, \boldsymbol{\xi}_{k,i} \rangle + \eta \Theta(\sigma_\xi^2 \sqrt{d}\sqrt{\log(Kn)}) + \frac{\eta}{Kn_2} \Theta(\sigma_\xi^2 d) |\langle \mathbf{w}_{1,r}^{(t,J)}, \boldsymbol{\xi}_{k,i} \rangle| \\
&\le \langle \mathbf{w}_{1,r}^{(t,0)}, \boldsymbol{\xi}_{k,i} \rangle + \eta \Theta(\sigma_\xi^2 \sqrt{d}\sqrt{\log(Kn)}) \\
&\quad + \frac{\eta}{Kn_2} \Theta(\sigma_\xi^2 d) \mathcal{O}\Big( \max\Big\{ \langle \mathbf{w}_{1,r}^{(t,0)}, \boldsymbol{\xi}_{k,i} \rangle, C \Big\} \Big),
\end{aligned}
\tag{F.10}
$$

where we used $-\ell'(\cdot) \le 1$ and $\sigma'(\cdot) \le 1$ and Lemma D.2 in the first inequality. We used $\sigma'(x) \le 2|x|$ for any $x \in \mathbb{R}$ to get the second inequality. And we used Lemma F.7 to get the last inequality. Abusing notation, let $r(t) = \arg\max_{r \in [m]} \langle \mathbf{w}_{1,r}^{(t,0)}, \boldsymbol{\xi}_{k,i} \rangle$. By definition, we have $\boldsymbol{\Xi}_{k,i}^{(t)} = \langle \mathbf{w}_{1,r(t)}^{(t,0)}, \boldsymbol{\xi}_{k,i} \rangle$. Let us compare the size of the second and third term on the right hand side of Eq. (F.10). The third term outweighs the second term at least by a factor of $\Theta(J\gamma\sigma_\xi^2 d/(Kn_2)) = \log(d)^{0.9}$, due to the presence of the constant $C$ in the third term. Therefore, the second term is dominated by the third term for all $r \in [m]$ and for all $t \ge 0$. So Eq. (F.10) becomes

$$
\begin{aligned}
\langle \mathbf{w}_{1,r}^{(t+1,0)}, \boldsymbol{\xi}_{k,i} \rangle &\le \langle \mathbf{w}_{1,r}^{(t,0)}, \boldsymbol{\xi}_{k,i} \rangle + \frac{\eta}{Kn_2} \Theta(\sigma_\xi^2 d) \mathcal{O}\Big( \max\Big\{ \langle \mathbf{w}_{1,r}^{(t,0)}, \boldsymbol{\xi}_{k,i} \rangle, C \Big\} \Big) \\
&\le \max\Big\{ \langle \mathbf{w}_{1,r}^{(t,0)}, \boldsymbol{\xi}_{k,i} \rangle, C \Big\} + \frac{\eta}{Kn_2} \Theta(\sigma_\xi^2 d) \mathcal{O}\Big( \max\Big\{ \langle \mathbf{w}_{1,r}^{(t,0)}, \boldsymbol{\xi}_{k,i} \rangle, C \Big\} \Big) \\
&\le \max\Big\{ \boldsymbol{\Xi}_{k,i}^{(t)}, C \Big\} + \frac{\eta}{Kn_2} \Theta(\sigma_\xi^2 d) \mathcal{O}\Big( \max\Big\{ \boldsymbol{\Xi}_{k,i}^{(t)}, C \Big\} \Big) \\
&= \widehat{\boldsymbol{\Xi}}_{k,i}^{(t)} \Big( 1 + \mathcal{O}(1) \frac{\eta\sigma_\xi^2 d}{Kn_2} \Big).
\end{aligned}
$$

Since the above inequality holds for all $r \in [m]$ on the left hand side, it holds for $r(t+1)$ in particular. This gives

$$
\boldsymbol{\Xi}_{k,i}^{(t+1)} \le \widehat{\boldsymbol{\Xi}}_{k,i}^{(t)} \Big( 1 + \mathcal{O}(1) \frac{\eta\sigma_\xi^2 d}{Kn_2} \Big).
$$

And it is straightforward to check by definition of $\widehat{\boldsymbol{\Xi}}_{k,i}^{(t)}$ that

$$
\widehat{\boldsymbol{\Xi}}_{k,i}^{(t+1)} \le \widehat{\boldsymbol{\Xi}}_{k,i}^{(t)} \Big( 1 + \mathcal{O}(1) \frac{\eta\sigma_\xi^2 d}{Kn_2} \Big).
\tag{F.11}
$$

We are now ready to prove our theorem. Compare Eq. (F.9) and Eq. (F.11), we can use Lemma D.5 to get an upper bound on $\widehat{\boldsymbol{\Xi}}_{k,i}^{(T^{(2)})}$. We take $A = \Omega(1)\eta\gamma^J$, $B = \mathcal{O}(1)\frac{\eta\sigma_\xi^2 d}{Kn_2}$, $G = 2$, $D = m^{-1/2}\gamma^{-J}$ in Lemma D.5. We see that since $\boldsymbol{\Lambda}_1^{(0)} = \Theta(d^{-1/2}) > G^{-A/B} = 2^{-\log(d)^{1.5}}$ under Condition D.1, Lemma D.5 tells us that

$$
\widehat{\boldsymbol{\Xi}}_{k,i}^{(T^{(2)})} \le \mathcal{O}(\widehat{\boldsymbol{\Xi}}_{k,i}^{(0)}).
$$

This proves our main theorem. $\qquad\square$

It remains to verify our previous hypothesis. Note that if we suppose all of our hypothesis hold at time $t < T^{(2)}$, then Theorem F.9 implies that for all $(k,i) \in \Psi$ we have $\widehat{\boldsymbol{\Xi}}_{k,i}^{(t+1)} \le \mathcal{O}(\widehat{\boldsymbol{\Xi}}_{k,i}^{(0)})$. By Lemma D.2, it is clear that Hypothesis F.5 and F.6 hold at initialization.

*Proof of Hypothesis F.5.* The proof will be similar to the proof of Hypothesis E.2. Let $k \in [K]$ and $i \in [n]$ be such that $y_{k,i} = 1$. Recall Eq. (C.7) and that

$$\sum_{j=\pm 1} \sum_{r=1}^{m} j \Big( \sigma(\langle \mathbf{w}_{j,r}^{(t+1,\tau)}, \boldsymbol{\xi}_{k,i} \rangle) + \sigma(\langle \mathbf{w}_{j,r}^{(t+1,\tau)}, \boldsymbol{\nu}_k \rangle) \Big)$$

$$\leq \sum_{r=1}^{m} \Big( \sigma(\langle \mathbf{w}_{1,r}^{(t+1,\tau)}, \boldsymbol{\xi}_{k,i} \rangle) + \sigma(\langle \mathbf{w}_{1,r}^{(t+1,\tau)}, \boldsymbol{\nu}_k \rangle) \Big). \qquad \text{(F.12)}$$

Assume our Hypothesis hold at time $t$. We have

$$\sum_{r=1}^{m} \sigma(\langle \mathbf{w}_{1,r}^{(t+1,0)}, \boldsymbol{\xi}_{k,i} \rangle) \leq \sum_{r=1}^{m} \sigma(\widehat{\boldsymbol{\Xi}}_{k,i}^{(t+1)})$$

$$\leq m\sigma(\mathcal{O}(\widehat{\boldsymbol{\Xi}}_{k,i}^{(0)}))$$

$$= o(1), \qquad \text{(F.13)}$$

where the first inequality is by definition of $\widehat{\boldsymbol{\Xi}}_{k,i}^{(t+1)}$ and the second inequality is because $\widehat{\boldsymbol{\Xi}}_{k,i}^{(t+1)} \leq \mathcal{O}(\widehat{\boldsymbol{\Xi}}_{k,i}^{(0)})$. The third equality is because $m = \text{polylog}(d)$ and $\sigma(\mathcal{O}(\widehat{\boldsymbol{\Xi}}_{k,i}^{(0)})) = \widetilde{\mathcal{O}}(d^{-1})$. For any $r \in [m]$, we have

$$\langle \mathbf{w}_{1,r}^{(t+1,0)}, \boldsymbol{\nu}_k \rangle = \langle \mathbf{w}_{1,r}^{(t+1,0)}, \boldsymbol{\nu} \rangle + \langle \mathbf{w}_{1,r}^{(t+1,0)}, \mathbf{z}_k \rangle$$

$$\leq \boldsymbol{\Lambda}_1^{(t+1)} + \max_{r' \in [m]} \langle \mathbf{w}_{1,r'}^{(t+1,0)}, \mathbf{z}_k \rangle$$

$$\leq \mathcal{O}(1)\boldsymbol{\Lambda}_1^{(t+1)}$$

$$\leq \mathcal{O}(1)\boldsymbol{\Lambda}_1^{(T^{(2)})}$$

$$= \mathcal{O}(m^{-1/2}(1+\gamma)^{-J}),$$

where the second line is by definition of $\boldsymbol{\Lambda}_1^{(t+1)}$. The third line is by Lemma F.1. The fourth line is because $t + 1 \leq T^{(2)}$ and $\boldsymbol{\Lambda}_1^{(t)}$ is increasing in $t$. The last line is by definition of $T^{(2)}$. Thus

$$\sum_{r=1}^{m} \sigma(\langle \mathbf{w}_{1,r}^{(t+1,\tau)}, \boldsymbol{\nu}_k \rangle) \leq m \cdot \mathcal{O}(m^{-1/2}(1+\gamma)^{-J}) = o(1). \qquad \text{(F.14)}$$

Combining Eq. (C.7), Eq. (F.12), Eq. (F.13), and Eq. (F.14), we have

$$-\ell_{k,i}'^{(t+1,0)} \geq -\ell'(o(1) + o(1)) = \Omega(1).$$

Now, suppose that $-\ell_{k,i}'^{(t+1,\tau)} = \Omega(1)$ for some $0 \leq \tau \leq J-1$. We wish to show that $-\ell_{k,i}'^{(t+1,\tau+1)} = \Omega(1)$. By Eq. (F.8), we know that for $i > n_1$, we have

$$\langle \mathbf{w}_{1,r}^{(t+1,\tau+1)}, \boldsymbol{\xi}_{k,i} \rangle \leq \mathcal{O}\Big( \max \Big\{ \langle \mathbf{w}_{1,r}^{(t+1,0)}, \boldsymbol{\xi}_{k,i} \rangle, C \Big\} \Big)$$

$$\leq \mathcal{O}\Big( \max \Big\{ \boldsymbol{\Xi}_{k,i}^{(t+1)}, C \Big\} \Big)$$

$$= \mathcal{O}\Big( \widehat{\boldsymbol{\Xi}}_{k,i}^{(t+1)} \Big),$$

where the second and third line are by definition of $\boldsymbol{\Xi}_{k,i}^{(t+1)}$ and $\widehat{\boldsymbol{\Xi}}_{k,i}^{(t+1)}$ respectively. This gives

$$\sum_{r=1}^{m} \sigma(\langle \mathbf{w}_{1,r}^{(t+1,\tau+1)}, \boldsymbol{\xi}_{k,i} \rangle) \leq \sum_{r=1}^{m} \sigma(\mathcal{O}(\widehat{\boldsymbol{\Xi}}_{k,i}^{(t+1)}))$$

$$\leq m\sigma(\mathcal{O}(\widehat{\boldsymbol{\Xi}}_{k,i}^{(0)}))$$

$$= o(1), \qquad \text{(F.15)}$$

where the second inequality is because $\widehat{\Xi}_{k,i}^{(t+1)} \leq \mathcal{O}(\widehat{\Xi}_{k,i}^{(0)})$. If $i \in [n_1]$, then by Eq. (C.4) we can bound

$$
\begin{aligned}
\langle \mathbf{w}_{1,r}^{(t+1,\tau+1)}, \boldsymbol{\xi}_{k,i} \rangle &\leq \langle \mathbf{w}_{1,r}^{(t+1,\tau)}, \boldsymbol{\xi}_{k,i} \rangle + \gamma \Theta(\sigma_\xi^2 \sqrt{d}\sqrt{\log(Kn)}) + \frac{\gamma}{n_1}\Theta(\sigma_\xi^2 d)\sigma'(\langle \mathbf{w}_{1,r}^{(t+1,\tau)}, \boldsymbol{\xi}_{k,i} \rangle) \\
&\leq \big|\langle \mathbf{w}_{1,r}^{(t+1,\tau)}, \boldsymbol{\xi}_{k,i} \rangle\big|\big(1 + \gamma\Theta(\sigma_\xi^2 d)/n_1\big) + \gamma\Theta(\sigma_\xi^2 \sqrt{d}\sqrt{\log(Kn)}) \\
&\leq \max\Big\{\big|\langle \mathbf{w}_{1,r}^{(t+1,\tau)}, \boldsymbol{\xi}_{k,i} \rangle\big|, C\Big\}\big(1 + \gamma\Theta(\sigma_\xi^2 d)/n_1\big),
\end{aligned}
$$

where we used $-\ell'(\cdot) \leq 1$ and $\sigma'(\cdot) \leq 1$ and Lemma D.2 in the first inequality. And we used $\sigma'(x) \leq 2|x|$ for any $x \in \mathbb{R}$ to get the second inequality. This implies

$$
\begin{aligned}
\max\Big\{\big|\langle \mathbf{w}_{1,r}^{(t+1,\tau+1)}, \boldsymbol{\xi}_{k,i} \rangle\big|, C\Big\} &\leq \max\Big\{\big|\langle \mathbf{w}_{1,r}^{(t+1,\tau)}, \boldsymbol{\xi}_{k,i} \rangle\big|, C\Big\}\big(1 + \gamma\Theta(\sigma_\xi^2 d)/n_1\big) \\
&\leq \max\Big\{\big|\langle \mathbf{w}_{1,r}^{(t+1,0)}, \boldsymbol{\xi}_{k,i} \rangle\big|, C\Big\}\big(1 + \gamma\Theta(\sigma_\xi^2 d)/n_1\big)^{\tau+1} \\
&= \widehat{\Xi}_{k,i}^{(t+1)}\big(1 + \gamma\Theta(\sigma_\xi^2 d)/n_1\big)^{\tau+1} \\
&\leq \mathcal{O}\big(\widehat{\Xi}_{k,i}^{(0)}\big)\big(1 + \gamma\Theta(\sigma_\xi^2 d)/n_1\big)^{\tau+1} \\
&= \widetilde{\mathcal{O}}(d^{-1/2}),
\end{aligned}
$$

where the third line is by definition of $\widehat{\Xi}_{k,i}^{(t+1)}$. The fourth line is because $\widehat{\Xi}_{k,i}^{(t+1)} \leq \mathcal{O}(\widehat{\Xi}_{k,i}^{(0)})$. And the last equality is by direct calculation. Thus

$$
\langle \mathbf{w}_{1,r}^{(t+1,\tau+1)}, \boldsymbol{\xi}_{k,i} \rangle \leq \max\Big\{\big|\langle \mathbf{w}_{1,r}^{(t+1,\tau+1)}, \boldsymbol{\xi}_{k,i} \rangle\big|, C\Big\} \leq \widetilde{\mathcal{O}}(d^{-1/2}).
$$

This gives

$$
\sum_{r=1}^m \sigma(\langle \mathbf{w}_{1,r}^{(t+1,\tau+1)}, \boldsymbol{\xi}_{k,i} \rangle) \leq \sum_{r=1}^m \sigma\big(\widetilde{\mathcal{O}}(d^{-1/2})\big) = m \cdot \sigma\big(\widetilde{\mathcal{O}}(d^{-1/2})\big) = o(1).
$$

Using Eq. (C.5), we have

$$
\begin{aligned}
\langle \mathbf{w}_{1,r}^{(t+1,\tau+1)}, \boldsymbol{\nu}_k \rangle &\leq \langle \mathbf{w}_{1,r}^{(t+1,\tau)}, \boldsymbol{\nu}_k \rangle + \gamma\Theta(1)\big|\langle \mathbf{w}_{1,r}^{(t+1,\tau)}, \boldsymbol{\nu}_k \rangle\big| \\
&\leq \big|\langle \mathbf{w}_{1,r}^{(t+1,\tau)}, \boldsymbol{\nu}_k \rangle\big|(1 + \Theta(1)\gamma) \\
&\leq \big|\langle \mathbf{w}_{1,r}^{(t+1,0)}, \boldsymbol{\nu}_k \rangle\big|(1 + \Theta(1)\gamma)^{\tau+1} \\
&\leq \big|\langle \mathbf{w}_{1,r}^{(t+1,0)}, \boldsymbol{\nu}_k \rangle\big|(1 + \Theta(1)\gamma)^J \\
&\leq \mathcal{O}(m^{-1/2}(1+\gamma)^{-J})(1 + \Theta(1)\gamma)^J \\
&= \mathcal{O}(m^{-1/2}), \quad\quad\quad\quad\quad\quad\quad\quad\quad\quad\quad\quad\quad\quad\quad\quad (\text{F.16})
\end{aligned}
$$

where we used $\ell'(\cdot) \leq 1$, $\sigma'(x) \leq 2|x|$ for any $x \in \mathbb{R}$ and $\|\boldsymbol{\nu}_k\|_2 = \Theta(1)$ in the first inequality. The fifth inequality is by $t + 1 \leq T^{(2)}$ and the definition of $T^{(2)}$. Then we have

$$
\sum_{r=1}^m \sigma(\langle \mathbf{w}_{1,r}^{(t+1,\tau+1)}, \boldsymbol{\nu}_k \rangle) \leq m \cdot \mathcal{O}(m^{-1}) = \mathcal{O}(1). \quad\quad\quad\quad (\text{F.17})
$$

The inequality is because $\sigma(\mathcal{O}(m^{-1/2})) = \mathcal{O}(m^{-1})$. Combining Eq. (C.7), Eq. (F.12), Eq. (F.15), and Eq. (F.17) we obtain

$$
-\ell_{k,i}^{\prime(t+1,\tau+1)} \geq -\ell'(\mathcal{O}(1) + o(1)) = \Omega(1).
$$

This proves the hypothesis. $\qquad\qquad\square$

*Proof of Hypothesis F.6.* By Eq. (F.16), we have that $\langle \mathbf{w}_{1,r}^{(t+1,\tau)}, \boldsymbol{\nu}_k \rangle < \mathcal{O}(m^{-1/2})$. Thus if $\langle \mathbf{w}_{1,r}^{(t+1,\tau)}, \boldsymbol{\nu}_k \rangle > 0$, then

$$
\sigma'(\langle \mathbf{w}_{1,r}^{(t+1,\tau)}, \boldsymbol{\nu}_k \rangle) = 2\langle \mathbf{w}_{1,r}^{(t+1,\tau)}, \boldsymbol{\nu}_k \rangle,
$$

by definition of $\sigma'(\cdot)$. $\qquad\qquad\square$

## F.2 Phase II

We have shown that when the feature inner product has grown to $\widetilde{\Theta}(1)$, the noise inner products remain at $\widetilde{\Theta}(d^{-1/2})$, which is exactly the opposite of Theorem E.2. Next we will show that this difference is maintained at least when $\mathbf{\Lambda}_j^{(t)}$ has grown to $\mathcal{O}(\log(d)^{0.1})$. For $(k,i) \in \Psi$, let

$$\widetilde{\mathbf{\Xi}}_{k,i}^{(t)} = \max\left\{ \max_{r \in [m]}\left\{\langle \mathbf{w}_{y_{k,i},r}^{(t,0)}, \boldsymbol{\xi}_{k,i}\rangle\right\}, d^{-1/3}\right\}.$$

And

$$\widetilde{\mathbf{\Gamma}}_{k,i}^{(t)} = \max\left\{ \max_{r \in [m]}\left\{\langle \mathbf{w}_{-y_{k,i},r}^{(t,0)}, \boldsymbol{\xi}_{k,i}\rangle\right\}, C\right\}.$$

Let $T_j^{(3)}$ be the first iteration such that $\mathbf{\Lambda}_j^{(t)} \geq \log(d)^{0.1}$. Let $T^{(3)} = \max\left\{T_{-1}^{(3)}, T_1^{(3)}\right\}$. And recall $T = \mathrm{poly}(d)$ is the total number of iterations. We first present two claims under Condition D.1 and Condition 4.1.

**Claim F.10.** *For any $t \geq T^{(2)}$, any $(k,i) \in \Psi$, we have $\widetilde{\mathbf{\Gamma}}_{k,i}^{(t)} \leq \widetilde{\Theta}(d^{-1/2})$.*

**Claim F.11.** *For any $t \leq T_j^{(3)}$, $(k,i) \in \Psi_j$, we have $\widetilde{\mathbf{\Xi}}_{k,i}^{(t)} \leq \mathcal{O}(d^{-1/4})$.*

*Remark F.12.* Under Claim F.10, it holds that $\widetilde{\mathbf{\Gamma}}_{k,i}^{(t)} = o(1)\widetilde{\mathbf{\Xi}}_{k',i'}^{(t)}$ for all $(k,i), (k',i') \in \Psi$.

*Remark F.13.* By Theorem F.9, Claim F.10 holds at time $T^{(2)}$. Also by Theorem F.9, Claim F.11 holds for all $t \leq T^{(2)}$.

Note that from Eq. (C.11), the growth of $j\langle \mathbf{w}_{j,r}^{(t,0)}, \boldsymbol{\nu}\rangle$ depends on the size of $j\langle \mathbf{w}_{j,r}^{(t,J)}, \boldsymbol{\nu}_k\rangle$. We first need a lemma that can gives a lower bound on $j\langle \mathbf{w}_{j,r}^{(t,J)}, \boldsymbol{\nu}_k\rangle$ for $t \geq T^{(2)}$. The following lemma will be of a similar form to Lemma E.7.

**Lemma F.14.** *For any $T_j^{(3)} \geq t \geq T^{(2)}$, let $r(t) = \arg\max_{r \in [m]} j\langle \mathbf{w}_{j,r}^{(t,0)}, \boldsymbol{\nu}\rangle$. Assume Claim F.11 holds, then $j\langle \mathbf{w}_{j,r(t)}^{(t,J)}, \boldsymbol{\nu}_k\rangle \geq \Omega(m^{-1/2})$ for any $k \in [K]$.*

*Proof.* For simplicity, take $j = 1$. We shall see that our proof does not depend on the choice of $j$ and thus taking $j = 1$ is not a loss of generality. By Eq. (C.5), we know that $\langle \mathbf{w}_{1,r(t)}^{(t,\tau)}, \boldsymbol{\nu}_k\rangle$ is an increasing function of $\tau$. Thus if $\langle \mathbf{w}_{1,r(t)}^{(t,0)}, \boldsymbol{\nu}_k\rangle = \Omega(m^{-1/2})$, then we must have $\langle \mathbf{w}_{1,r(t)}^{(t,J)}, \boldsymbol{\nu}_k\rangle \geq \Omega(m^{-1/2})$. By Eq. (C.11) and the definition of $r(t)$, we also have that $\langle \mathbf{w}_{1,r(t)}^{(t,0)}, \boldsymbol{\nu}\rangle$ is an increasing function in $t$. By definition of $T^{(2)}$, for any $t \geq T^{(2)}$, it holds that $\langle \mathbf{w}_{1,r(t)}^{(t,0)}, \boldsymbol{\nu}\rangle \geq m^{-1/2}(1+\gamma)^{-J}$. Then by Lemma F.2, we have $\langle \mathbf{w}_{1,r(t)}^{(t,0)}, \boldsymbol{\nu}_k\rangle \geq \mathcal{O}(1)m^{-1/2}(1+\gamma)^{-J}$. It suffices to prove that if

$$m^{-1/2}(1+\gamma)^{-\tau} \leq \langle \mathbf{w}_{1,r(t)}^{(t,0)}, \boldsymbol{\nu}_k\rangle \leq m^{-1/2}(1+\gamma)^{-\tau+1}, \tag{F.18}$$

for $\tau \in [J]$, then $\langle \mathbf{w}_{1,r(t)}^{(t,J)}, \boldsymbol{\nu}_k\rangle \geq \Omega(m^{-1/2})$. Let us suppose that F.18 is true for some $\tau$. Under our assumption that $\widetilde{\mathbf{\Xi}}_{k,i}^{(t)} \leq \mathcal{O}(d^{-1/4})$, we can apply the same proof as Hypothesis F.5 to get that $-\ell_{k,i}^{\prime(t,\tau')} = \Omega(1)$ for any $\tau' \in [\tau]$. Then using the same derivation of Eq. (F.7), we obtain

$$\langle \mathbf{w}_{1,r(t)}^{(t,\tau)}, \boldsymbol{\nu}_k\rangle = \Omega(1)\langle \mathbf{w}_{1,r(t)}^{(t,0)}, \boldsymbol{\nu}\rangle(1+\Theta(1)\gamma)^\tau \geq m^{-1/2}(1+\gamma)^{-\tau}\Omega(1)(1+\Theta(1)\gamma)^\tau = \Omega(1)m^{-1/2}.$$

This concludes the lemma. □

The next lemma shows that as long as the noise vectors have not been learnt well by our network, the loss from different samples are essentially due to the feature and hence are of similar sizes.

**Lemma F.15.** *Let $j \in \{-1, 1\}$. Under Condition D.1 and Condition 4.1, while $\widetilde{\mathbf{\Xi}}_{k,i}^{(t)} \leq \mathcal{O}(d^{-1/4})$ and $\widetilde{\mathbf{\Gamma}}_{k,i}^{(t)} \leq \mathcal{O}(d^{-1/2})$ for all $(k,i) \in \Psi_j$, and $m\sigma_s^2 d\mathbf{\Lambda}_j^{(t)} \leq \mathcal{O}(1)$, we have $\ell_{k_1,i_1}^{\prime(t,J)} = \Theta(1)\ell_{k_2,i_2}^{\prime(t,J)}$ if $(k_1,i_1), (k_2,i_2) \in \Psi_j$.*

*Proof.* Without loss of generality, assume $y_{k_1,i_1} = y_{k_2,i_2} = j = 1$. We can essentially use the proof of Hypothesis F.5 to get that

$$\sum_{r=1}^{m} \sigma(\langle \mathbf{w}_{1,r}^{(t,J)}, \boldsymbol{\xi}_{k_q,i_q} \rangle) = o(1), \tag{F.19a}$$

$$\sum_{r=1}^{m} \sigma(\langle \mathbf{w}_{-1,r}^{(t,J)}, \boldsymbol{\xi}_{k_q,i_q} \rangle) = o(1), \tag{F.19b}$$

for $q \in \{1,2\}$. Moreover, since $j\langle \mathbf{w}_{j,r}^{(t,\tau)}, \boldsymbol{\nu}_k \rangle$ is increasing in $t$ and $\tau$ for any $j \in \{-1,1\}, r \in [m], k \in [K]$, we get that $\langle \mathbf{w}_{-1,r}^{(t,J)}, \boldsymbol{\nu}_k \rangle \leq \langle \mathbf{w}_{-1,r}^{(0,0)}, \boldsymbol{\nu}_k \rangle \leq \mathcal{O}(d^{-1/2})$. This implies

$$\sum_{r=1}^{m} \sigma(\langle \mathbf{w}_{-1,r}^{(t,J)}, \boldsymbol{\nu}_{k_q} \rangle) = o(1),$$

for $q \in \{1,2\}$. By Eq. (C.7), it suffices to show that

$$\left| \sum_{r=1}^{m} \sigma(\langle \mathbf{w}_{1,r}^{(t,J)}, \boldsymbol{\nu}_{k_1} \rangle) - \sigma(\langle \mathbf{w}_{1,r}^{(t,J)}, \boldsymbol{\nu}_{k_2} \rangle) \right| \leq \mathcal{O}(1).$$

We divide the weights into two sets. Let $\mathcal{G} = \{ r \in [m] : \langle \mathbf{w}_{1,r}^{(t,0)}, \boldsymbol{\nu} \rangle \geq d^{-1/3} \}$, and $\mathcal{H} = \{ r \in [m] : r \notin \mathcal{G} \}$. Let us first consider $r \in \mathcal{G}$. By Corollary F.5, we have that

$$\langle \mathbf{w}_{1,r}^{(t,J)}, \mathbf{z}_{k_q} \rangle = \mathcal{O}(\sigma_s^2 d)\langle \mathbf{w}_{1,r}^{(t,J)}, \boldsymbol{\nu} \rangle,$$

for $q \in \{1,2\}$, which implies

$$\langle \mathbf{w}_{1,r}^{(t,J)}, \boldsymbol{\nu}_{k_q} \rangle = (1 + \mathcal{O}(\sigma_s^2 d))\langle \mathbf{w}_{1,r}^{(t,J)}, \boldsymbol{\nu} \rangle.$$

Moreover, by Lemma F.6 we get

$$\langle \mathbf{w}_{1,r}^{(t,J)}, \boldsymbol{\nu}_{k_q} \rangle \leq \langle \mathbf{w}_{1,r}^{(t,J)}, \boldsymbol{\nu} \rangle + \mathcal{O}(\sigma_s^2 d)2\max\left\{ \boldsymbol{\Lambda}_1^{(t)}, J\gamma \right\}.$$

By a direct calculation, we have $m\sigma_s^2 dJ\gamma = o(1)$ and by assumption $m\sigma_s^2 d\boldsymbol{\Lambda}_1^{(t)} \leq \mathcal{O}(1)$. We can then calculate

$$\begin{aligned}
\sum_{r \in \mathcal{G}} \sigma(\langle \mathbf{w}_{1,r}^{(t,J)}, \boldsymbol{\nu}_{k_q} \rangle) &\leq \sum_{r \in \mathcal{G}} \sigma\big(\langle \mathbf{w}_{1,r}^{(t,J)}, \boldsymbol{\nu} \rangle + \mathcal{O}(\sigma_s^2 d)2\max\left\{ \boldsymbol{\Lambda}_1^{(t)}, J\gamma \right\}\big) \\
&\leq \sum_{r \in \mathcal{G}} \sigma(\langle \mathbf{w}_{1,r}^{(t,J)}, \boldsymbol{\nu} \rangle) + \sum_{r \in \mathcal{G}} \mathcal{O}(1)\mathcal{O}(\sigma_s^2 d)2\max\left\{ \boldsymbol{\Lambda}_1^{(t)}, J\gamma \right\} \\
&\leq \sum_{r \in \mathcal{G}} \sigma(\langle \mathbf{w}_{1,r}^{(t,J)}, \boldsymbol{\nu} \rangle) + m\mathcal{O}(\sigma_s^2 d)2\max\left\{ \boldsymbol{\Lambda}_1^{(t)}, J\gamma \right\} \\
&\leq \sum_{r \in \mathcal{G}} \sigma(\langle \mathbf{w}_{1,r}^{(t,J)}, \boldsymbol{\nu} \rangle) + \mathcal{O}(1), \tag{F.20}
\end{aligned}$$

where in the second line we used $\sigma(x + \varepsilon) \leq \sigma(x) + \mathcal{O}(1)\varepsilon$ for $\varepsilon \geq 0$. The third line is due to $|\mathcal{G}| \leq m$. If $r \in \mathcal{H}$, then by Lemma F.3 we have $\langle \mathbf{w}_{1,r}^{(t,0)}, \mathbf{z}_{k_q} \rangle < d^{-1/3}$. Then by Eq. (C.5) we have

$$\left| \langle \mathbf{w}_{1,r}^{(t,\tau+1)}, \boldsymbol{\nu}_{k_q} \rangle \right| \leq \left| \langle \mathbf{w}_{1,r}^{(t,\tau)}, \boldsymbol{\nu}_{k_q} \rangle \right| (1 + \mathcal{O}(1)\gamma),$$

for $\tau \in [J-1]$. Hence

$$\begin{aligned}
\left| \langle \mathbf{w}_{1,r}^{(t,J)}, \boldsymbol{\nu}_{k_q} \rangle \right| &\leq \left| \langle \mathbf{w}_{1,r}^{(t,0)}, \boldsymbol{\nu}_{k_q} \rangle \right| (1 + \mathcal{O}(1)\gamma)^J \\
&\leq \widetilde{\mathcal{O}}(d^{-1/3}). \tag{F.21}
\end{aligned}$$

Combining the above results, we finally have

$$\sum_{r=1}^{m} \sigma(\langle \mathbf{w}_{1,r}^{(t,J)}, \boldsymbol{\nu}_{k_q} \rangle) = \sum_{r \in \mathcal{G}} \sigma(\langle \mathbf{w}_{1,r}^{(t,J)}, \boldsymbol{\nu}_{k_q} \rangle) + \sum_{r \in \mathcal{H}} \sigma(\langle \mathbf{w}_{1,r}^{(t,J)}, \boldsymbol{\nu}_{k_q} \rangle)$$

$$\begin{aligned}
&\leq \sum_{r \in \mathcal{G}} \sigma(\langle \mathbf{w}_{1,r}^{(t,J)}, \boldsymbol{\nu} \rangle) + \mathcal{O}(1) + \sum_{r \in \mathcal{H}} \sigma(\widetilde{\mathcal{O}}(d^{-1/3})) \\
&\leq \sum_{r \in \mathcal{G}} \sigma(\langle \mathbf{w}_{1,r}^{(t,J)}, \boldsymbol{\nu} \rangle) + \mathcal{O}(1) + m\sigma(\widetilde{\mathcal{O}}(d^{-1/3})) \\
&= \sum_{r \in \mathcal{G}} \sigma(\langle \mathbf{w}_{1,r}^{(t,J)}, \boldsymbol{\nu} \rangle) + \mathcal{O}(1) + o(1) \\
&= \sum_{r \in \mathcal{G}} \sigma(\langle \mathbf{w}_{1,r}^{(t,J)}, \boldsymbol{\nu} \rangle) + \mathcal{O}(1) \,, \tag{F.22}
\end{aligned}$$

where the second line is due to Eq. (F.20) and Eq. (F.21). Since the above holds for $q \in \{1, 2\}$, we get

$$\left| \sum_{r=1}^{m} \sigma(\langle \mathbf{w}_{1,r}^{(t,J)}, \boldsymbol{\nu}_{k_1} \rangle) - \sigma(\langle \mathbf{w}_{1,r}^{(t,J)}, \boldsymbol{\nu}_{k_2} \rangle) \right| \leq \mathcal{O}(1) + \mathcal{O}(1) = \mathcal{O}(1) \,.$$

And using Eq. (C.7), we conclude that $\ell_{k_1,i_1}^{\prime(t,J)} = \Theta(1) \ell_{k_2,i_2}^{\prime(t,J)}$. $\qquad \square$

**Lemma F.16.** *Under Claim F.11, for any $t \leq T_j^{(3)}$ we have $-\ell_{k,i}^{\prime(t,J)} = \omega(d^{-1/8})$ for all $(k,i) \in \Psi_j$.*

*Proof.* Without loss of generality, let us consider $j = 1$. Let $(k,i) \in \Psi_+$. By Eq. (C.7), Eq. (F.19) and Eq. (F.22) we have that

$$\begin{aligned}
-\ell_{k,i}^{\prime(t,J)} &\geq -\ell'\left( \sum_{r \in \mathcal{G}} \sigma(\langle \mathbf{w}_{1,r}^{(t,J)}, \boldsymbol{\nu} \rangle) + \mathcal{O}(1) \right) \\
&\geq -\ell'\left( \sum_{r \in [m]} \sigma(\langle \mathbf{w}_{1,r}^{(t,J)}, \boldsymbol{\nu} \rangle) + \mathcal{O}(1) \right) \,,
\end{aligned}$$

where in the first line $\mathcal{G} = \{r \in [m] : \langle \mathbf{w}_{1,r}^{(t,0)}, \boldsymbol{\nu} \rangle \geq d^{-1/3}\}$. Then recall that $t \leq T_1^{(3)}$ implies $\boldsymbol{\Lambda}_1^{(t)} < \log(d)^{0.1}$. Using Lemma F.6, we have that for all $r \in [m]$

$$\langle \mathbf{w}_{1,r}^{(t,J)}, \boldsymbol{\nu} \rangle \leq 2\max\{\log(d)^{0.1}, J\gamma\} = 2J\gamma \,.$$

Then by a direct calculation using Condition D.1, we get

$$\begin{aligned}
-\ell_{k,i}^{\prime(t,J)} &\geq -\ell'\left( m\mathcal{O}(1)J\gamma + \mathcal{O}(1) \right) \\
&= \omega(d^{-1/8}) \,.
\end{aligned}$$

$\qquad \square$

*Remark* F.17. The $-1/8$ factor on $d$ can be replaced by any $c < 0$. We choose $-1/8$ for convenience in the later proof.

*Proof of Claim F.11.* Consider $T^{(2)} \leq t \leq T_1^{(3)}$. Let $\widetilde{\boldsymbol{\Xi}}_1^{(t)} = \max\{\widetilde{\boldsymbol{\Xi}}_{k,i}^{(t)} : (k,i) \in \Psi_+\}$. By Eq. (F.8) and the definition of $\widetilde{\boldsymbol{\Xi}}_{k,i}^{(t)}$, we have that for any $(k,i) \in \Psi_+$,

$$\sigma'(\langle \mathbf{w}_{1,r}^{(t,J)}, \boldsymbol{\xi}_{k,i} \rangle) \leq \mathcal{O}(1) \widetilde{\boldsymbol{\Xi}}_{k,i}^{(t)} \leq \mathcal{O}(1) \widetilde{\boldsymbol{\Xi}}_1^{(t)} \,. \tag{F.23}$$

And if $(k,i) \in \Psi_-$,

$$\sigma'(\langle \mathbf{w}_{1,r}^{(t,J)}, \boldsymbol{\xi}_{k,i} \rangle) \leq \mathcal{O}(1) \widetilde{\boldsymbol{\Gamma}}_{k,i}^{(t)} = o(1) \widetilde{\boldsymbol{\Xi}}_1^{(t)} \,, \tag{F.24}$$

where we note the second inequality in Remark F.12. It is convenient to define

$$\ell_1^{\prime(t,J)} = \sum_{(k,i) \in \Psi_+} \ell_{k,i}^{\prime(t,J)} \Big/ |\Psi_+| \,,$$

because by Lemma F.15, $\ell'^{(t,J)}_{k,i} = \Theta(1)\ell'^{(t,J)}_1$ for all $(k,i) \in \Psi_+$. Then by our assumption that $t \leq T^{(3)}_1$ and Lemma F.16 we have that $-\ell'^{(t,J)}_1 = \omega(d^{-1/8})$. By Eq. (C.9) we have that for any $r \in [m]$, and $(k,i) \in \Psi_+$

$$\langle \mathbf{w}^{(t+1,0)}_{1,r}, \boldsymbol{\xi}_{k,i} \rangle \leq \langle \mathbf{w}^{(t,0)}_{1,r}, \boldsymbol{\xi}_{k,i} \rangle - \frac{\eta}{Kn_2} \sum_{(k',i')\in\Psi} \ell'^{(t,J)}_{k',i'} \sigma'(\langle \mathbf{w}^{(t,J)}_{1,r}, \boldsymbol{\xi}_{k',i'} \rangle) \Theta(\sigma^2_\xi \sqrt{d}\sqrt{\log(Kn)})$$

$$- \frac{\eta}{Kn_2} \Theta(1)\ell'^{(t,J)}_1 \sigma'(\langle \mathbf{w}^{(t,J)}_{1,r}, \boldsymbol{\xi}_{k,i} \rangle) \Theta(\sigma^2_\xi d)$$

$$\leq \widetilde{\boldsymbol{\Xi}}^{(t)}_1 - \Theta\Big(\frac{\eta\sigma^2_\xi \sqrt{d}}{Kn_2}\Big) \sum_{(k',i')\in\Psi} \ell'^{(t,J)}_{k',i'} \sigma'(\langle \mathbf{w}^{(t,J)}_{1,r}, \boldsymbol{\xi}_{k',i'} \rangle) - \Theta\Big(\frac{\eta\sigma^2_\xi d}{Kn_2}\Big) \ell'^{(t,J)}_1 \widetilde{\boldsymbol{\Xi}}^{(t)}_1$$

$$\leq \widetilde{\boldsymbol{\Xi}}^{(t)}_1 - \Theta\Big(\frac{\eta\sigma^2_\xi \sqrt{d}}{Kn_2}\Big) \sum_{(k',i')\in\Psi_+} \ell'^{(t,J)}_{k',i'} \sigma'(\langle \mathbf{w}^{(t,J)}_{1,r}, \boldsymbol{\xi}_{k',i'} \rangle)$$

$$- \Theta\Big(\frac{\eta\sigma^2_\xi \sqrt{d}}{Kn_2}\Big) \sum_{(k',i')\in\Psi_-} \ell'^{(t,J)}_{k',i'} \sigma'(\langle \mathbf{w}^{(t,J)}_{1,r}, \boldsymbol{\xi}_{k',i'} \rangle) - \Theta\Big(\frac{\eta\sigma^2_\xi d}{Kn_2}\Big) \ell'^{(t,J)}_1 \widetilde{\boldsymbol{\Xi}}^{(t)}_1$$

$$\leq \widetilde{\boldsymbol{\Xi}}^{(t)}_1 - \Theta\Big(\frac{\eta\sigma^2_\xi \sqrt{d}}{Kn_2}\Big) \sum_{(k',i')\in\Psi_+} \ell'^{(t,J)}_1 \widetilde{\boldsymbol{\Xi}}^{(t)}_1$$

$$+ \Theta\Big(\frac{\eta\sigma^2_\xi \sqrt{d}}{Kn_2}\Big) \sum_{(k',i')\in\Psi_-} o(1)\widetilde{\boldsymbol{\Xi}}^{(t)}_1 - \Theta\Big(\frac{\eta\sigma^2_\xi d}{Kn_2}\Big) \ell'^{(t,J)}_1 \widetilde{\boldsymbol{\Xi}}^{(t)}_1$$

$$\leq \widetilde{\boldsymbol{\Xi}}^{(t)}_1 - \Big(\widetilde{\Theta}(d^{-1/2}) + \Theta\Big(\frac{\eta\sigma^2_\xi d}{Kn_2}\Big)\Big) \ell'^{(t,J)}_1 \widetilde{\boldsymbol{\Xi}}^{(t)}_1 + o(d^{-1/2})\widetilde{\boldsymbol{\Xi}}^{(t)}_1$$

$$= \widetilde{\boldsymbol{\Xi}}^{(t)}_1 \Big(1 - \Theta\Big(\frac{\eta\sigma^2_\xi d}{Kn_2}\Big)\ell'^{(t,J)}_1 + o(d^{-1/2})\Big)$$

$$= \widetilde{\boldsymbol{\Xi}}^{(t)}_1 \Big(1 - \Theta\Big(\frac{\eta\sigma^2_\xi d}{Kn_2}\Big)\ell'^{(t,J)}_1\Big),$$

where we used Eq. (F.23) to get the second inequality. We used Eq. (F.23) and Eq. (F.24) to get the fourth inequality. The last equality is because $-\Theta\Big(\frac{\eta\sigma^2_\xi d}{Kn_2}\Big)\ell'^{(t,J)}_1 = \widetilde{\Theta}(1)\omega(d^{-1/8}) \gg o(d^{-1/2})$. And maximizing the left hand side over $r \in [m]$ and $(k,i) \in \Psi_+$ we get

$$\widetilde{\boldsymbol{\Xi}}^{(t+1)}_1 \leq \widetilde{\boldsymbol{\Xi}}^{(t)}_1 \Big(1 - \Theta\Big(\frac{\eta\sigma^2_\xi d}{Kn_2}\Big)\ell'^{(t,J)}_1\Big). \tag{F.25}$$

By Theorem F.9, we have that $\widetilde{\boldsymbol{\Xi}}^{(T^{(2)})}_1 = d^{-1/3}$. Let $T'$ be the first iteration such that $\widetilde{\boldsymbol{\Xi}}^{(T')}_1 \geq d^{-1/4}$. Then, using Eq. (F.25) we get

$$\widetilde{\boldsymbol{\Xi}}^{(T')}_1 \leq \widetilde{\boldsymbol{\Xi}}^{(T'-1)}_1 \Big(1 - \Theta\Big(\frac{\eta\sigma^2_\xi d}{Kn_2}\Big)\ell'^{(T'-1,J)}_1\Big)$$

$$\leq d^{-1/4}\Big(1 - \Theta\Big(\frac{\eta\sigma^2_\xi d}{Kn_2}\Big)\ell'^{(T'-1,J)}_1\Big)$$

$$\leq d^{-1/4}\Big(1 + \Theta\Big(\frac{\eta\sigma^2_\xi d}{Kn_2}\Big)\Big)$$

$$= \widetilde{\Theta}(d^{-1/4}),$$

where the second inequality is by definition of $T'$. The third inequality is due to $-\ell'(\cdot) \leq 1$. We have

$$d^{-1/4} \leq \widetilde{\boldsymbol{\Xi}}^{(T')}_1 \leq \widetilde{\boldsymbol{\Xi}}^{(T^{(2)})}_1 \prod_{t=T^{(2)}}^{T'-1} \Big(1 - \Theta\Big(\frac{\eta\sigma^2_\xi d}{Kn_2}\Big)\ell'^{(t,J)}_1\Big),$$

$$d^{-1/4}/d^{-1/3} \leq \prod_{t=T^{(2)}}^{T'-1} \left(1 - \Theta\left(\frac{\eta\sigma_\xi^2 d}{Kn_2}\right)\ell_1'^{(t,J)}\right),$$

$$\Omega(\log(d)) \leq \sum_{t=T^{(2)}}^{T'-1} \log\left(1 - \Theta\left(\frac{\eta\sigma_\xi^2 d}{Kn_2}\right)\ell_1'^{(t,J)}\right)$$

$$\leq \sum_{t=T^{(2)}}^{T'-1} -\mathcal{O}(1)\left(\frac{\eta\sigma_\xi^2 d}{Kn_2}\right)\ell_1'^{(t,J)},$$

$$\Omega\left(\frac{\log(d)Kn_2}{\eta\sigma_\xi^2 d}\right) \leq \sum_{t=T^{(2)}}^{T'-1} -\ell_1'^{(t,J)}, \tag{F.26}$$

where we get the third inequality by taking the logarithm of the second inequality. In the fourth inequality, we used $\log(1+x) \leq \mathcal{O}(1)x$ for $x \in [0,1)$. By the second line of Eq. (F.9) we have

$$\mathbf{\Lambda}_1^{(t+1)} \geq \mathbf{\Lambda}_1^{(t)} - \frac{\eta}{Kn_2} \sum_{(k,i)\in\Psi_+} \ell_{k,i}'^{(t,J)}\Omega(m^{-1/2})$$

$$\geq \mathbf{\Lambda}_1^{(t)} - \eta\ell_1'^{(t,J)}\Omega(m^{-1/2}).$$

To get the second inequality, we use Lemma F.15 and absorb all the constants into $\Omega(m^{-1/2})$. Summing over $t$, we obtain

$$\log(d)^{0.1} \geq \mathbf{\Lambda}^{(T_1^{(3)}-1)} \geq \mathbf{\Lambda}^{(T^{(2)})} - \sum_{t=T^{(2)}}^{T_1^{(3)}-2} \eta\ell_1'^{(t,J)}\Omega(m^{-1/2}),$$

$$\log(d)^{0.1} \geq - \sum_{t=T^{(2)}}^{T_1^{(3)}-2} \eta\ell_1'^{(t,J)}\Omega(m^{-1/2}),$$

$$\mathcal{O}\left(\frac{\log(d)^{0.1}\sqrt{m}}{\eta}\right) \geq \sum_{t=T^{(2)}}^{T_1^{(3)}-2} -\ell_1'^{(t,J)},$$

$$\mathcal{O}\left(\frac{\log(d)^{0.1}\sqrt{m}+\eta}{\eta}\right) \geq \sum_{t=T^{(2)}}^{T_1^{(3)}-1} -\ell_1'^{(t,J)}. \tag{F.27}$$

We used $-\ell'(\cdot) \leq 1$ to get the last inequality. By a direct computation, we have

$$\frac{\log(d)^{0.1}\sqrt{m}+\eta}{\eta} = \frac{\log(d)^{0.2}+\eta}{\eta} = o(1)\frac{\log(d)Kn_2}{\eta\sigma_\xi^2 d} = o(1)\frac{\log(d)^{0.5}}{\eta}.$$

Combining this fact with Eq. (F.26) and Eq. (F.27) we conclude that $T_1^{(3)} - 1 < T' - 1$ and hence $T_1^{(3)} < T'$. Now let $\widetilde{\mathbf{\Xi}}_{-1}^{(t)} = \max\left\{\widetilde{\mathbf{\Xi}}_{k,i}^{(t)} : (k,i) \in \Psi_-\right\}$. If we redefine $T'$ to be the first iteration such that $\widetilde{\mathbf{\Xi}}_{-1}^{(T')} \geq d^{-1/4}$, then using the exactly same argument, we obtain $T_{-1}^{(3)} < T'$. This proves our Claim. $\qquad\square$

**Theorem F.18** (Restatement of Lemma 6.5). *Under Claim F.10, we have $T^{(3)} < T$.*

*Proof.* We will only prove $T_1^{(3)} < T$, since the proof of $T_{-1}^{(3)} < T$ will be the same. For the sake of contradiction, let us assume that $T_1^{(3)} \geq T$. Then, for $t \leq T$, we have that $\mathbf{\Lambda}_1^{(t)} \leq \log(d)^{0.1}$. From the proof of Lemma F.16, we have

$$-\ell_{k,i}'^{(t,J)} \geq -\ell'\left(m\mathcal{O}(1)\gamma J + \mathcal{O}(1)\right) \geq \Omega(1)\exp(-\mathcal{O}(\log(d)^{0.6})),$$

for all $(k,i) \in \Psi_+$. By the second line of Eq. (F.9)

$$\mathbf{\Lambda}_1^{(t+1)} \geq \mathbf{\Lambda}_1^{(t)} - \frac{\eta}{Kn_2} \sum_{(k,i)\in\Psi_+} \ell_{k,i}'^{(t,J)}\Theta(1)\Omega(m^{-1/2})$$

$$\geq \mathbf{\Lambda}_1^{(t)} + \frac{\eta}{Kn_2} \sum_{(k,i)\in\Psi_+} \Omega(1)\exp(-\mathcal{O}(\log(d)^{0.6}))\Theta(1)\Omega(m^{-1/2})$$

$$= \mathbf{\Lambda}_1^{(t)} + \Omega(1)\eta\exp(-\mathcal{O}(\log(d)^{0.6}))m^{-1/2} .$$

This implies

$$\mathbf{\Lambda}_1^{(T)} \geq \mathbf{\Lambda}_1^{(T^{(2)})} + \sum_{t=T^{(2)}}^{T-1} \Omega(1)\eta\exp(-\mathcal{O}(\log(d)^{0.6}))m^{-1/2}$$

$$\geq \widetilde{\Theta}(1) + (T - 1 - T^{(2)})\Omega(1)\eta\exp(-\mathcal{O}(\log(d)^{0.6}))m^{-1/2}$$

$$\geq \mathrm{poly}(d)\Omega(1)\exp(-\mathcal{O}(\log(d)^{0.6}))/\mathrm{polylog}(d)$$

$$= \Omega(\mathrm{poly}(d)) ,$$

where the second inequality is by definition of $T^{(2)}$. The third inequality is because $T = \mathrm{poly}(d)$, $T^{(2)} = \mathrm{polylog}(d)$, $\eta = (\mathrm{polylog}(d))^{-1}$, and $m = \mathrm{polylog}(d)$. And $\mathrm{poly}(d)\exp(-\mathcal{O}(\log(d)^{0.6})) = \Omega(\mathrm{poly}(d))$. But this is a contradiction since we assumed $T \leq T_1^{(3)}$ which means $\mathbf{\Lambda}_1^{(T)} \leq \log(d)^{0.1}$. Thus, we must have $T > T_1^{(3)}$. And the proof of $T > T_{-1}^{(3)}$ is the same. This proves our theorem. This will be useful for our convergence result later. $\square$

Theorem F.18 implies that before training ends, feature will be memorized by the neurons with an inner product of size at least $\log(d)^{0.1}$. As we shall see later, this already guarantees a small test loss. Our next theorem makes sure that the feature inner product will not grow too big by the end of the training.

**Theorem F.19.** *Under Claim F.10, we have* $\mathbf{\Lambda}_j^{(T)} \leq \log(d)^{1.2}$ *for* $j \in \{-1, 1\}$.

*Proof.* Without loss of generality, let $j = 1$. Let $t'$ be the first iteration such that $\mathbf{\Lambda}_1^{(t)} \geq \log(d)^{1.1}$. Let $r(t) = \arg\max_{r\in[m]} \left\{ \langle \mathbf{w}_{1,r}^{(t,0)}, \boldsymbol{\nu} \rangle \right\}$. Using Eq. (C.11) we have

$$\langle \mathbf{w}_{1,r(t+1)}^{(t+1,0)}, \boldsymbol{\nu} \rangle = \langle \mathbf{w}_{1,r(t+1)}^{(t,0)}, \boldsymbol{\nu} \rangle - \frac{\eta}{Kn_2} \sum_{(k,i)\in\Psi} \ell_{k,i}^{\prime(t,J)} \sigma'(\langle \mathbf{w}_{1,r(t+1)}^{(t,J)}, \boldsymbol{\nu}_k \rangle y_{k,i})$$

$$\leq \langle \mathbf{w}_{1,r(t)}^{(t,0)}, \boldsymbol{\nu} \rangle - \frac{\eta}{Kn_2} \sum_{(k,i)\in\Psi} \ell_{k,i}^{\prime(t,J)} \sigma'(\langle \mathbf{w}_{1,r(t)}^{(t,J)}, \boldsymbol{\nu}_k \rangle y_{k,i})$$

$$= \langle \mathbf{w}_{1,r(t)}^{(t,0)}, \boldsymbol{\nu} \rangle - \frac{\eta}{Kn_2} \sum_{(k,i)\in\Psi_+} \ell_{k,i}^{\prime(t,J)} \sigma'(\langle \mathbf{w}_{1,r(t)}^{(t,J)}, \boldsymbol{\nu}_k \rangle)$$

$$\leq \langle \mathbf{w}_{1,r(t)}^{(t,0)}, \boldsymbol{\nu} \rangle - \frac{\eta}{Kn_2} \sum_{(k,i)\in\Psi_+} \ell_{k,i}^{\prime(t,J)} ,$$

where the second line is by definition of $r(t)$. We get the third line using the second line of Eq. (F.9). We have used $\sigma'(\cdot) \leq 1$ to get the last line. By definition of $r(t)$, this means

$$\mathbf{\Lambda}_1^{(t+1)} \leq \mathbf{\Lambda}_1^{(t)} - \frac{\eta}{Kn_2} \sum_{(k,i)\in\Psi_+} \ell_{k,i}^{\prime(t,J)} .$$

And by definition of $t'$, we have

$$\mathbf{\Lambda}_1^{(t')} \leq \mathbf{\Lambda}_1^{(t'-1)} - \frac{\eta}{Kn_2} \sum_{(k,i)\in\Psi_+} \ell_{k,i}^{\prime(t'-1,J)}$$

$$\leq \log(d)^{1.1} + \eta$$

$$\leq \mathcal{O}(1)\log(d)^{1.1} ,$$

where we have used $-\ell'(\cdot) \leq 1$ to get the second inequality. Hence if $T \leq t'$, then we are done. So let us suppose that $T > t'$. Since $\mathbf{\Lambda}_1^{(t)}$ is increasing in $t$, we know that for all $t \geq t'$, $\mathbf{\Lambda}_1^{(t)} \geq$

$\mathbf{\Lambda}_1^{(t')} \geq \log(d)^{1.1}$. By Corollary F.5, we have that $\langle \mathbf{w}_{1,r(t)}^{(t,0)}, \boldsymbol{\nu}_k \rangle = \Theta(1) \langle \mathbf{w}_{1,r(t)}^{(t,0)}, \boldsymbol{\nu} \rangle$ for any $k \in [K]$. Moreover, for $t \geq t'$

$$\langle \mathbf{w}_{1,r(t)}^{(t,J)}, \boldsymbol{\nu}_k \rangle \geq \langle \mathbf{w}_{1,r(t)}^{(t,0)}, \boldsymbol{\nu}_k \rangle = \Theta(1) \langle \mathbf{w}_{1,r(t)}^{(t,0)}, \boldsymbol{\nu} \rangle = \Theta(1) \mathbf{\Lambda}_1^{(t)} \geq \Theta(1) \log(d)^{1.1}.$$

By Eq. (C.7), for $(k,i) \in \Psi_+$ we have

$$
\begin{aligned}
-\ell_{k,i}^{\prime(t,J)} &\leq -\ell' \Big( \sigma(\langle \mathbf{w}_{1,r(t)}^{(t,J)}, \boldsymbol{\nu}_k \rangle) - \sum_{r=1}^{m} \big( \sigma(\langle \mathbf{w}_{-1,r}^{(t,J)}, \boldsymbol{\xi}_{k,i} \rangle) + \sigma(\langle \mathbf{w}_{-1,r}^{(t,J)}, \boldsymbol{\nu}_k \rangle) \big) \Big) \\
&\leq -\ell' \Big( \Theta(1) \log(d)^{1.1} - o(1) \Big) \\
&= -\ell' \big( \Theta(1) \log(d)^{1.1} \big) \\
&\leq \mathcal{O}(1) \exp \big( -\Theta(1) \log(d)^{1.1} \big),
\end{aligned}
$$

where the second inequality is due to Eq. (F.19). The last inequality is by property of the function $\ell'(\cdot)$. This implies

$$
\begin{aligned}
\mathbf{\Lambda}_1^{(T)} &\leq \mathbf{\Lambda}_1^{(t')} - \frac{\eta}{Kn_2} \sum_{t=t'}^{T-1} \sum_{(k,i) \in \Psi_+} \ell_{k,i}^{\prime(t,J)} \\
&\leq \mathcal{O}(1) \log(d)^{1.1} + \eta T \mathcal{O}(1) \exp \big( -\Theta(1) \log(d)^{1.1} \big) \\
&\leq \mathcal{O}(1) \log(d)^{1.1} + o(1) \\
&= \mathcal{O}(1) \log(d)^{1.1},
\end{aligned}
$$

where the second last inequality is because $\eta T = \mathcal{O}(\text{poly}(d)/\text{polylog}(d)) = \mathcal{O}(\text{poly}(d))$ and $\mathcal{O}(\text{poly}(d)) \exp \big( -\Theta(1) \log(d)^{1.1} \big) = o(1)$. This proves our theorem. $\qquad \square$

*Proof of Claim F.10.* Suppose our claim holds for all $t \in [T^{(2)}, T'-1]$ where $T' \leq T$. Then by Theorem F.19, we have that $\mathbf{\Lambda}_j^{(T')} \leq \log(d)^{1.2}$ for $j \in \{-1, 1\}$. Let $(k,i) \in \Psi_-$. Denote $r(t) = \arg\max_{r \in [m]} \langle \mathbf{w}_{1,r}^{(t,0)}, \boldsymbol{\xi}_{k,i} \rangle$. From Eq. (C.9), we get that

$$
\begin{aligned}
\langle \mathbf{w}_{1,r}^{(t+1,0)}, \boldsymbol{\xi}_{k,i} \rangle &\leq \langle \mathbf{w}_{1,r}^{(t,0)}, \boldsymbol{\xi}_{k,i} \rangle - \frac{\eta}{Kn_2} \sum_{(k',i') \in \Psi \setminus \{(k,i)\}} \ell_{k',i'}^{\prime(t,J)} \Theta(\sigma_\xi^2 \sqrt{d} \sqrt{\log(Kn)}) \\
&\leq \langle \mathbf{w}_{1,r}^{(t,0)}, \boldsymbol{\xi}_{k,i} \rangle - \frac{\eta}{Kn_2} \sum_{(k',i') \in \Psi} \ell_{k',i'}^{\prime(t,J)} \Theta(\sigma_\xi^2 \sqrt{d} \sqrt{\log(Kn)}) \\
&= \langle \mathbf{w}_{1,r}^{(t,0)}, \boldsymbol{\xi}_{k,i} \rangle + \widetilde{\Theta}(d^{-1/2}) \eta \sum_{(k',i') \in \Psi} \ell_{k',i'}^{\prime(t,J)},
\end{aligned}
\tag{F.28}
$$

where we also used Lemma D.2 in the first line. The third line is a direct calculation using Condition D.1. We have for $t \geq T^{(2)}$

$$
\begin{aligned}
\mathbf{\Lambda}_1^{(t+1)} &\geq \mathbf{\Lambda}_1^{(t)} - \frac{\eta}{Kn_2} \sum_{(k,i) \in \Psi_+} \ell_{k,i}^{\prime(t,J)} \Theta(1) \widetilde{\Theta}(1) \\
&= \mathbf{\Lambda}_1^{(t)} - \widetilde{\Theta}(1) \eta \sum_{(k,i) \in \Psi_+} \ell_{k,i}^{\prime(t,J)},
\end{aligned}
$$

where the first inequality is by definition of $T^{(2)}$ and the monotonicity of $\mathbf{\Lambda}_1^{(t)}$. And similarly

$$\mathbf{\Lambda}_{-1}^{(t+1)} \geq \mathbf{\Lambda}_{-1}^{(t)} - \widetilde{\Theta}(1) \eta \sum_{(k,i) \in \Psi_-} \ell_{k,i}^{\prime(t,J)}.$$

Summing them up we obtain

$$\mathbf{\Lambda}_1^{(t+1)} + \mathbf{\Lambda}_{-1}^{(t+1)} \geq \mathbf{\Lambda}_1^{(t)} + \mathbf{\Lambda}_{-1}^{(t)} - \widetilde{\Theta}(1) \eta \sum_{(k,i) \in \Psi} \ell_{k,i}^{\prime(t,J)}.$$

Summing over $t$ we get

$$2\log(d)^{1.2} \geq \mathbf{\Lambda}_1^{(T')} + \mathbf{\Lambda}_{-1}^{(T')} \geq \mathbf{\Lambda}_1^{(T^{(2)})} + \mathbf{\Lambda}_{-1}^{(T^{(2)})} - \widetilde{\Theta}(1)\eta \sum_{t=T^{(2)}}^{T'-1} \sum_{(k,i)\in\Psi} \ell_{k,i}'^{(t,J)},$$

$$2\log(d)^{1.2} \geq -\widetilde{\Theta}(1)\eta \sum_{t=T^{(2)}}^{T'-1} \sum_{(k,i)\in\Psi} \ell_{k,i}'^{(t,J)},$$

$$\widetilde{\Theta}(1) \geq -\eta \sum_{t=T^{(2)}}^{T'-1} \sum_{(k,i)\in\Psi} \ell_{k,i}'^{(t,J)}. \tag{F.29}$$

Plugging Eq. (F.29) into Eq. (F.28) we have

$$\langle \mathbf{w}_{1,r}^{(T',0)}, \boldsymbol{\xi}_{k,i} \rangle \leq \langle \mathbf{w}_{1,r}^{(T^{(2)},0)}, \boldsymbol{\xi}_{k,i} \rangle - \widetilde{\Theta}(d^{-1/2})\eta \sum_{t=T^{(2)}}^{T'-1} \sum_{(k',i')\in\Psi} \ell_{k',i'}'^{(t,J)}$$

$$\leq \langle \mathbf{w}_{1,r}^{(T^{(2)},0)}, \boldsymbol{\xi}_{k,i} \rangle + \widetilde{\Theta}(d^{-1/2})\widetilde{\Theta}(1)$$

$$\leq \widetilde{\Theta}(d^{-1/2}) + \widetilde{\Theta}(d^{-1/2})$$

$$= \widetilde{\Theta}(d^{-1/2}),$$

where the third inequality is by Theorem F.9. Since this relation holds for all $r \in [m]$, we get that $\max_{r\in[m]} \langle \mathbf{w}_{1,r}^{(T',0)}, \boldsymbol{\xi}_{k,i} \rangle \leq \widetilde{\Theta}(d^{-1/2})$. And hence $\widetilde{\mathbf{\Gamma}}_{k,i}^{(T')} \leq \widetilde{\Theta}(d^{-1/2})$. This proves our claim. $\qquad\square$

**Lemma F.20** (Restatement of Lemma 6.4). *Let $T = \text{poly}(d)$ be the total number of iterations. Under Condition D.1 and Condition 4.1, we have*

$$\min_{t\in[T^{(2)},T-1]} \sum_{(k,i)\in\Psi} -\ell_{k,i}'^{(t,J)} \leq \widetilde{\mathcal{O}}\left(\frac{1}{T\eta}\right).$$

*Proof.* By Theorem F.19, we have $\mathbf{\Lambda}_j^{(T)} \leq \log(d)^{1.2}$ for $j \in \{-1,1\}$. Using the same derivation of Eq. (F.29), we have

$$-\sum_{t=T^{(2)}}^{T-1} \sum_{(k,i)\in\Psi} \eta \ell_{k,i}'^{(t,J)} \leq \widetilde{\Theta}(1).$$

And we can bound the minimum over $t$ by

$$\min_{t\in[T^{(2)},T-1]} \sum_{(k,i)\in\Psi} -\ell_{k,i}'^{(t,J)} \leq \widetilde{\mathcal{O}}\left(\frac{1}{(T-T^{(2)})\eta}\right) = \widetilde{\mathcal{O}}\left(\frac{1}{T\eta}\right),$$

where the second equality is because $T^{(2)} = \text{polylog}(d) = o(1)T$, and hence $T - T^{(2)} = \Omega(1)T$. $\qquad\square$

Lastly, we have the following theorem for the convergence of training similar to Theorem E.10.

**Theorem F.21** (Restatement of Theorem 4.5 (1)). *Let $T = \text{poly}(d)$ be the total number of iterations. Under Condition D.1 and Condition 4.1, we have*

$$\min_{t\in[T]} \widehat{\mathcal{L}}^{\text{tr-val}}(\boldsymbol{W}^{(t)}) \leq \widetilde{\Theta}(T^{-1}\eta^{-1}).$$

*Proof.* Denote $-\ell_{k,i}'^{(t,J)} = -\ell'(\alpha(t,J,k,i))$, i.e. the argument of $-\ell_{k,i}'^{(t,J)}$ is some function $\alpha(t,J,k,i)$. Then the training loss at time $t$ is related to $-\ell_{k,i}'^{(t,J)}$ by

$$\widehat{\mathcal{L}}^{\text{tr-val}}(\boldsymbol{W}^{(t)}) = \sum_{(k,i)\in\Psi} \ell\big(\alpha(t,J,k,i)\big).$$

Let $t' = \arg\min_{t \in [T^{(2)}]}^{T-1} \sum_{(k,i) \in \Psi} -\ell_{k,i}'^{(t,J)}$. By definition of $\ell'(\cdot)$ and $\ell(\cdot)$ we can bound $-\ell'(x) \geq \exp(-x)/2 \geq \ell(x)/2$ for $x > 0$. This can be used to bound our loss by

$$
\min_{t \in [T]} \widehat{\mathcal{L}}^{\text{tr-val}}(\mathbf{W}^{(t)}) \leq \widehat{\mathcal{L}}^{\text{tr-val}}(\mathbf{W}^{(t')})
$$

$$
= \sum_{(k,i) \in \Psi} \ell\big(\alpha(t', J, k, i)\big)
$$

$$
\leq -2 \sum_{(k,i) \in \Psi} \ell'\big(\alpha(t', J, k, i)\big)
$$

$$
\leq \widetilde{\mathcal{O}}\left(\frac{1}{T\eta}\right),
$$

where the last inequality is by Lemma F.20. This proves our theorem. $\qquad\square$

**Theorem F.22** (Restatement of Theorem 4.5 (2))**.** *Let $\mathbf{W} = \mathbf{W}^{(T)}$ be our trained weights using the train-validation method. Then the test loss is small*

$$
\mathcal{L}_{\text{test}}(\mathbf{W}) = o\big(1/\text{polylog}(d)\big).
$$

*Proof.* Sample a new example $\mathbf{x}$ from the distribution, since we use the convolutional structure, we can assume that the first patch $\mathbf{x}^{(1)} = y \cdot (\boldsymbol{\nu} + \mathbf{z})$ and the second patch $\mathbf{x}^{(2)} = \boldsymbol{\xi}$. Clearly $\langle \boldsymbol{\xi}_{k,i}, \boldsymbol{\xi} \rangle$ follows the Gaussian distribution with mean zero and standard deviation $\sigma_\xi \cdot \|\boldsymbol{\xi}_{k,i}\|_2$. By Lemma D.2, we further know that $\|\boldsymbol{\xi}_{k,i}\|_2 \leq 2\sigma_\xi \sqrt{d}$. Therefore

$$
\mathbb{P}\big(|\langle \boldsymbol{\xi}_{k,i}, \boldsymbol{\xi} \rangle| \geq d^{-1/4}\big) \leq 2\exp\left(-\frac{1}{8\sigma_\xi^4 d^{3/2}}\right) \leq 2\exp(-d^{1/4}). \tag{F.30}
$$

Similarly we have that

$$
\mathbb{P}\big(|\langle \mathbf{w}_{j,r}^{(0,0)}, \boldsymbol{\xi} \rangle| \geq d^{3/2}\big) \leq 2\exp\left(-\frac{1}{8\sigma_0^2 \sigma_\xi^2 d^{3/2}}\right) \leq 2\exp(-d^{1/4}). \tag{F.31}
$$

Denote $\mathcal{E}$ to be the event that $|\langle \boldsymbol{\xi}_{k,i}, \boldsymbol{\xi} \rangle| \leq d^{-1/4}$ and $|\langle \mathbf{w}_{j,r}^{(0,0)}, \boldsymbol{\xi} \rangle| \leq d^{3/2}$ for all $k \in [K], i \in [n], r \in [m]$ and $j \in \{-1, 1\}$. Applying union bound, we have that the $\mathbb{P}(\mathcal{E}) \geq 1 - 2(nK + m)\exp(-d^{1/4})$. We can divide $\mathcal{L}_{\text{test}}(\mathbf{W})$ into two parts:

$$
\mathcal{L}_{\text{test}}(\mathbf{W}) = \mathbb{E}\big[\ell\big(yf(\mathbf{W}, \mathbf{x})\big)\big] = \underbrace{\mathbb{E}\big[\mathbf{1}(\mathcal{E})\ell\big(yf(\mathbf{W}, \mathbf{x})\big)\big]}_{I_1} + \underbrace{\mathbb{E}\big[\mathbf{1}(\mathcal{E}^c)\ell\big(yf(\mathbf{W}, \mathbf{x})\big)\big]}_{I_2}. \tag{F.32}
$$

In the following, we bound $I_1$ and $I_2$ respectively.

**Bounding $I_1$:** When event $\mathcal{E}$ holds, for any $j \in \{-1, 1\}$, we have

$$
\langle \mathbf{w}_{j,r}^{(T,0)}, \boldsymbol{\xi} \rangle = \langle \mathbf{w}_{j,r}^{(0,0)}, \boldsymbol{\xi} \rangle - \frac{\eta}{Kn_2} \sum_{t=0}^{T-1} \sum_{(k,i) \in \Psi} \ell_{k,i}'^{(t,J)} \sigma'(\langle \mathbf{w}_{j,r}^{(t,J)}, \boldsymbol{\xi}_{k,i} \rangle)\langle \boldsymbol{\xi}, \boldsymbol{\xi}_{k,i} \rangle
$$

$$
\leq \langle \mathbf{w}_{j,r}^{(0,0)}, \boldsymbol{\xi} \rangle - \frac{\eta}{Kn_2} \sum_{t=0}^{T-1} \sum_{(k,i) \in \Psi} \ell_{k,i}'^{(t,J)} \sigma'(\langle \mathbf{w}_{j,r}^{(t,J)}, \boldsymbol{\xi}_{k,i} \rangle)d^{-1/4}
$$

$$
\leq \langle \mathbf{w}_{j,r}^{(0,0)}, \boldsymbol{\xi} \rangle - \frac{\eta}{Kn_2} \sum_{t=0}^{T-1} \sum_{(k,i) \in \Psi} \ell_{k,i}'^{(t,J)} d^{-1/4}
$$

$$
\leq \langle \mathbf{w}_{j,r}^{(0,0)}, \boldsymbol{\xi} \rangle - \frac{\eta d^{-1/4}}{Kn_2} \sum_{t=0}^{T^{(2)}-1} \sum_{(k,i) \in \Psi} \ell_{k,i}'^{(t,J)} - \frac{\eta d^{-1/4}}{Kn_2} \sum_{t=T^{(2)}}^{T-1} \sum_{(k,i) \in \Psi} \ell_{k,i}'^{(t,J)}
$$

$$
\leq \langle \mathbf{w}_{j,r}^{(0,0)}, \boldsymbol{\xi} \rangle + \eta d^{-1/4} T^{(2)} + \frac{\eta d^{-1/4}}{Kn_2} \widetilde{\mathcal{O}}(1)
$$

$$\leq \langle \mathbf{w}_{j,r}^{(0,0)}, \boldsymbol{\xi} \rangle + \widetilde{\mathcal{O}}(d^{-1/4}) + \widetilde{\mathcal{O}}(d^{-1/4})$$
$$= \widetilde{\mathcal{O}}(d^{-1/4}), \tag{F.33}$$

where the second line is by Eq. (F.30). The third line is by $0 \leq \sigma'(\cdot) \leq 1$. The third last line is due to $-\ell'(\cdot) \leq 1$, $T^{(2)} = \text{polylog}(d)$ and $\sum_{t=T^{(2)}}^{T-1} \sum_{(k,i) \in \Psi} \ell_{k,i}'^{(t,J)} = \widetilde{\mathcal{O}}(1)$ (recall Eq. (F.29)). The second last line is by Eq. (F.31). We note that Eq. (F.33) holds for all $j \in \{-1,1\}$, $r \in [m]$, and $y \in \{-1,1\}$. Now without loss of generality we assume $y = 1$. Then,

$$F_{-1}(\mathbf{W}_{-1}, \mathbf{x}) = \sum_{r=1}^{m} \sigma(\langle \mathbf{w}_{-1,r}^{(T,0)}, \boldsymbol{\nu} + \mathbf{z} \rangle) + \sigma(\langle \mathbf{w}_{-1,r}^{(T,0)}, \boldsymbol{\xi} \rangle)$$
$$\leq \sum_{r=1}^{m} \sigma(\langle \mathbf{w}_{-1,r}^{(T,0)}, \boldsymbol{\nu} + \mathbf{z} \rangle) + m\sigma(\widetilde{\mathcal{O}}(d^{-1/4}))$$
$$\leq \sum_{r=1}^{m} \sigma(\langle \mathbf{w}_{-1,r}^{(0,0)}, \boldsymbol{\nu} + \mathbf{z} \rangle) + m\sigma(\widetilde{\mathcal{O}}(d^{-1/4}))$$
$$\leq \sum_{r=1}^{m} \sigma(\widetilde{\mathcal{O}}(d^{-1/4})) + m\sigma(\widetilde{\mathcal{O}}(d^{-1/4}))$$
$$\leq \log 2,$$

where the second line is by Lemma Eq. (F.33), the third line is by monotonicity of $j\langle \mathbf{w}_{j,r}^{(t,0)}, \boldsymbol{\nu} \rangle$ and $j\langle \mathbf{w}_{j,r}^{(t,0)}, \mathbf{z} \rangle$ in $t$, the fourth line is by Lemma D.2. Thus

$$\ell\big(yf(\mathbf{W}, \mathbf{x})\big) = \ell\bigg(F_{+1}(\mathbf{W}_{+1}, \mathbf{x}) - F_{-1}(\mathbf{W}_{-1}, \mathbf{x})\bigg)$$
$$\leq \ell\bigg(\sum_{r=1}^{m} \sigma(\langle \mathbf{w}_{1,r}^{(T,0)}, \boldsymbol{\nu} + \mathbf{z} \rangle) - F_{-1}(\mathbf{W}_{-1}, \mathbf{x})\bigg)$$
$$\leq \ell\bigg(\sigma\big(\Lambda_1^{(T)}\big) - F_{-1}(\mathbf{W}_{-1}, \mathbf{x})\bigg)$$
$$\leq \ell\bigg(\sigma\big(\Lambda_1^{(T^{(3)})}\big) - F_{-1}(\mathbf{W}_{-1}, \mathbf{x})\bigg)$$
$$\leq \ell\bigg(\log(d)^{0.1} - \log 2\bigg)$$
$$\leq 2\exp\big(-\log(d)^{0.1}\big),$$

where the second and the third line are due to the fact that $\sigma(\cdot) \geq 0$, the fourth line is by Theorem F.18, the fifth line is by definition of $T^{(3)}$, the last line is by $\log(1 + x) \leq x, \forall x \geq 0$. Therefore we have that

$$I_1 \leq 2\exp\big(-\log(d)^{0.1}\big). \tag{F.34}$$

**Bounding $I_2$:** Next we bound the second term $I_2$. When event $\mathcal{E}$ holds, for any $j \in \{-1,1\}$, we have

$$\langle \mathbf{w}_{j,r}^{(T,0)}, \boldsymbol{\xi} \rangle = \langle \mathbf{w}_{j,r}^{(0,0)}, \boldsymbol{\xi} \rangle - \frac{\eta}{Kn_2} \sum_{t=0}^{T-1} \sum_{(k,i) \in \Psi} \ell_{k,i}'^{(t,J)} \sigma'(\langle \mathbf{w}_{j,r}^{(t,J)}, \boldsymbol{\xi}_{k,i} \rangle)\langle \boldsymbol{\xi}, \boldsymbol{\xi}_{k,i} \rangle$$
$$\leq \langle \mathbf{w}_{j,r}^{(0,0)}, \boldsymbol{\xi} \rangle - \frac{\eta}{Kn_2} \sum_{t=0}^{T-1} \sum_{(k,i) \in \Psi} \ell_{k,i}'^{(t,J)} \sigma'(\langle \mathbf{w}_{j,r}^{(t,J)}, \boldsymbol{\xi}_{k,i} \rangle)\widetilde{\mathcal{O}}(\|\boldsymbol{\xi}\|_2)$$
$$\leq \langle \mathbf{w}_{j,r}^{(0,0)}, \boldsymbol{\xi} \rangle - \frac{\eta}{Kn_2} \sum_{t=0}^{T-1} \sum_{(k,i) \in \Psi} \ell_{k,i}'^{(t,J)} \widetilde{\mathcal{O}}(\|\boldsymbol{\xi}\|_2)$$

$$\leq \langle \mathbf{w}_{j,r}^{(0,0)}, \boldsymbol{\xi} \rangle - \frac{\eta \widetilde{\mathcal{O}}(\|\boldsymbol{\xi}\|_2)}{Kn_2} \sum_{t=0}^{T^{(2)}-1} \sum_{(k,i)\in\Psi} \ell_{k,i}'^{(t,J)} - \frac{\eta \widetilde{\mathcal{O}}(\|\boldsymbol{\xi}\|_2)}{Kn_2} \sum_{t=T^{(2)}}^{T-1} \sum_{(k,i)\in\Psi} \ell_{k,i}'^{(t,J)}$$

$$\leq \langle \mathbf{w}_{j,r}^{(0,0)}, \boldsymbol{\xi} \rangle + \eta \widetilde{\mathcal{O}}(\|\boldsymbol{\xi}\|_2)T^{(2)} + \frac{\eta \widetilde{\mathcal{O}}(\|\boldsymbol{\xi}\|_2)}{Kn_2}$$

$$\leq \langle \mathbf{w}_{j,r}^{(0,0)}, \boldsymbol{\xi} \rangle + \widetilde{\mathcal{O}}(\|\boldsymbol{\xi}\|_2) + \widetilde{\mathcal{O}}(\|\boldsymbol{\xi}\|_2)$$

$$= \widetilde{\mathcal{O}}(\|\boldsymbol{\xi}\|_2), \tag{F.35}$$

where the second line is by $|\langle \boldsymbol{\xi}, \boldsymbol{\xi}_{k,i} \rangle| \leq \|\boldsymbol{\xi}_{k,i}\|_2\|\boldsymbol{\xi}\|_2 = \widetilde{\mathcal{O}}(\|\boldsymbol{\xi}\|_2)$. The third line is by $0 \leq \sigma'(\cdot) \leq 1$, $T^{(2)} = \text{polylog}(d)$ and $\sum_{t=T^{(2)}}^{T-1} \sum_{(k,i)\in\Psi} \ell_{k,i}'^{(t,J)} = \widetilde{\mathcal{O}}(1)$ (recall Eq. (F.29)). The second last line is by $|\langle \mathbf{w}_{j,r}^{(0,0)}, \boldsymbol{\xi} \rangle| \leq \|\mathbf{w}_{j,r}^{(0,0)}\|_2\|\boldsymbol{\xi}\|_2 = \widetilde{\mathcal{O}}(\|\boldsymbol{\xi}\|_2)$. We note that Eq. (F.33) holds for all $j \in \{-1,1\}$, $r \in [m]$, and $y \in \{-1,1\}$. Now again without loss of generality we assume $y = 1$ and have that

$$F_{-1}(\mathbf{W}_{-1}, \mathbf{x}) = \sum_{r=1}^{m} \sigma(\langle \mathbf{w}_{-1,r}^{(T,0)}, \boldsymbol{\nu} + \mathbf{z} \rangle) + \sigma(\langle \mathbf{w}_{-1,r}^{(T,0)}, \boldsymbol{\xi} \rangle)$$

$$\leq \sum_{r=1}^{m} \sigma(\langle \mathbf{w}_{-1,r}^{(T,0)}, \boldsymbol{\nu} + \mathbf{z} \rangle) + m\sigma(\widetilde{\mathcal{O}}(\|\boldsymbol{\xi}\|_2))$$

$$\leq \sum_{r=1}^{m} \sigma(\langle \mathbf{w}_{-1,r}^{(0,0)}, \boldsymbol{\nu} + \mathbf{z} \rangle) + m\sigma(\widetilde{\mathcal{O}}(\|\boldsymbol{\xi}\|_2))$$

$$\leq \sum_{r=1}^{m} \sigma(\widetilde{\mathcal{O}}(d^{-1/2})) + m\sigma(\widetilde{\mathcal{O}}(\|\boldsymbol{\xi}\|_2))$$

$$\leq 1 + \widetilde{\mathcal{O}}(\|\boldsymbol{\xi}\|_2),$$

where the second line is by Eq. (F.35). The third line is by monotonicity of $j\langle \mathbf{w}_{j,r}^{(t,0)}, \boldsymbol{\nu} \rangle$ and $j\langle \mathbf{w}_{j,r}^{(t,0)}, \mathbf{z} \rangle$ in $t$. The fourth line is by Lemma D.2. Thus

$$\ell\big(yf(\mathbf{W}, \mathbf{x})\big) = \ell\bigg(F_{+1}(\mathbf{W}_{+1}, \mathbf{x}) - F_{-1}(\mathbf{W}_{-1}, \mathbf{x})\bigg)$$

$$\leq \ell\bigg(-F_{-1}(\mathbf{W}_{-1}, \mathbf{x})\bigg)$$

$$= \log(1 + \exp(F_{-1}(\mathbf{W}_{-1}, \mathbf{x})))$$

$$\leq 1 + F_{-1}(\mathbf{W}_{-1}, \mathbf{x})$$

$$\leq 2 + \widetilde{\mathcal{O}}(\|\boldsymbol{\xi}\|_2), \tag{F.36}$$

where the second line is due to the fact that $\sigma(\cdot) \geq 0$, the fourth line is by by the property of cross-entropy loss, i.e., $\log(1 + \exp(x)) \leq 1 + x$ for all $x \geq 0$. Then we further have that

$$I_2 \leq \sqrt{\mathbb{E}[\mathbb{1}(\mathcal{E}^c)]} \cdot \sqrt{\mathbb{E}\Big[\ell\big(yf(\mathbf{W}, \mathbf{x})\big)^2\Big]}$$

$$\leq \sqrt{\mathbb{P}(\mathcal{E}^c)} \cdot \sqrt{8 + \widetilde{\mathcal{O}}(1)\mathbb{E}[\|\boldsymbol{\xi}\|_2^2]}$$

$$\leq \sqrt{\mathcal{O}(\text{poly}(d))\exp(-d^{1/4})}$$

$$= \mathcal{O}(\text{poly}(d))\exp(-0.5d^{1/4}),$$

where the first inequality is by Cauchy-Schwartz inequality. The second inequality is by Eq. (F.36). The third inequality is by the fact that $\sqrt{8 + \widetilde{\mathcal{O}}(1)\mathbb{E}[\|\boldsymbol{\xi}\|_2^2]} \leq \mathcal{O}(\text{poly}(d))$. Plugging the bounds of $I_1, I_2$ into Eq. (F.32) gives that

$$\mathcal{L}_{\text{test}}(\mathbf{W}) \leq 2\exp\big(-\log(d)^{0.1}\big) + \mathcal{O}(\text{poly}(d))\exp(-0.5d^{1/4}) = O\big(\exp\big(-\log(d)^{0.1}\big)\big),$$

which completes the proof.

$$\square$$

# G   AUXILIARY LEMMA

**Lemma G.1** (Restatement of Lemma B.2 from Cao et al. (2022)). *Suppose that $\delta > 0$ and $d = \Omega\big(\log(4Kn/\delta)\big)$. Then with probability at least $1 - \delta$*

$$\frac{1}{2}\sigma_\xi^2 d \leq \|\boldsymbol{\xi}_{k,i}\|_2^2 \leq \frac{3}{2}\sigma_\xi^2 d,$$

$$|\langle \boldsymbol{\xi}_{k,i}, \boldsymbol{\xi}_{k',i'} \rangle| \leq 2\sigma_\xi^2 \sqrt{d}\sqrt{\log\big(4(Kn)^2/\delta\big)},$$

*for all $k, k' \in [K]$ and $i, i' \in [n]$ with $(k, i) \neq (k', i')$.*

**Lemma G.2** (Restatement of Lemma B.3 from Cao et al. (2022)). *Suppose that $\delta > 0$, $d = \Omega(\log(mKn/\delta))$ and $m = \Omega(\log(1/\delta))$. Then with probability at least $1 - \delta$*

$$|\langle \mathbf{w}_{j,r}^{(0,0)}, \boldsymbol{\nu} \rangle| \leq \sqrt{2\log(8m/\delta)}\sigma_0,$$

$$|\langle \mathbf{w}_{j,r}^{(0,0)}, \boldsymbol{\xi}_{k,i} \rangle| \leq \sqrt{2\log(8mKn/\delta)}\sigma_0\sigma_\xi\sqrt{d},$$

*for all $r \in [m]$, $j \in \{-1, 1\}$, $k \in [K]$ and $i \in [n]$. Moreover,*

$$\frac{\sigma_0}{2} \leq \max_{r \in [m]} j\langle \mathbf{w}_{j,r}^{(0,0)}, \boldsymbol{\nu} \rangle \leq \sqrt{2\log(8m/\delta)}\sigma_0,$$

$$\frac{\sigma_0\sigma_\xi\sqrt{d}}{4} \leq \max_{r \in [m]} \langle \mathbf{w}_{j,r}^{(0,0)}, \boldsymbol{\xi}_{k,i} \rangle \leq \sqrt{2\log(8mKn/\delta)}\sigma_0\sigma_\xi\sqrt{d},$$

*for all $j \in \{-1, 1\}$, $k \in [K]$ and $i \in [n]$.*

