# OpenReview forum: "Understanding Train-Validation Split in Meta-Learning with Neural Networks"
_ICLR.cc/2023/Conference — ICLR 2023 poster_

### Official Review · Reviewer_SqBj · 2022-10-23

**Confidence:** 2
**Correctness:** 4
**Technical Novelty And Significance:** 3
**Empirical Novelty And Significance:** 2
**Recommendation:** 6

**Clarity, Quality, Novelty And Reproducibility:**

The paper is clearly written.

The novelty is moderate.

Results seem to be reproducible.

**Strength And Weaknesses:**

**Strength**:

* This paper presents theoretical results of FOMAML and advances previous results to convolutional neural networks.

* The paper is well-written

**Weakness**:

* Does the theoretical results or analysis also applicable for second-order MAML, implicit MAML[1], closed-form solver[2], etc?


* If the number of samples is large, how do the theoretical results change with different numbers of samples? In this case, does train-validation still better train-train meta learning?


* How does the results compare to Reptile[3], which does not need a train-validation split.

References;

[1] Meta-Learning with Implicit Gradients. NeurIPS 2019

[2] Meta-learning with differentiable closed-form solvers. ICLR 2019

[3] On First-Order Meta-Learning Algorithms



**Summary Of The Paper:**

This paper proposes a theoretical results of FOMAML that shows train-train and train-validation methods can achieve a small training loss. It is necessary to perform a train-validation split in the task data to get good generalization results. Compared to previous work that only studies linear models . This paper advances the results to two-layer convolutional networks.


**Summary Of The Review:**

This paper proposes a theoretical results of FOMAML that shows train-train and train-validation methods can achieve a small training loss. The results advance previous results and provide a deeper understanding of meta-learning.

---

> ### Author Response · Authors · 2022-11-15
> **Response to Reviewer SqBj**
>
> Thank you for your supportive comments. We address your questions and suggestions as follows.
>
> **Q1**: “applicable for second-order MAML, implicit MAML[1], closed-form solver[2], etc?”
>
> **A1**: We believe our analysis can be applied to the above algorithms to some extent. But it will require some extra effort to carefully study different algorithms. Thus we believe they are more suitable for future work. We have modified our conclusion section to include the above algorithms as future directions.
>
> ----
> **Q2**: “If the number of samples is large, …  still better train-train meta learning?”
>
> **A2**: As the number of samples in each task goes to infinity, the advantage of train-validation over train-train will diminish. This can be seen from Lemma E.1 in the Appendix. Comparing the two growth equations in Lemma E.1, we can see that the growth of the noise inner product can be much faster than the growth of the feature inner product because of the term $\sigma_{\xi}^2 d / n$. However, if we let $n$ be large, then $\sigma_{\xi}^2 d / n$ will become smaller, which implies that the difference between the feature growth and noise growth will decrease or even be reversed if $\sigma_{\xi}^2 d / n = o(1)$. We are motivated to study the case for $n = O(1)$ because this is a common setting for meta-learning problems: the scarcity of training examples (e.g., few-shot learning) makes it difficult for vanilla gradient descent to learn the features and that is why we need meta-learning to alleviate this problem. We have also added some experiments in Appendix B.4 to discuss the performance with respect to the number of examples per class. The result is summarized in the table below (see also Table 6 in Appendix B.4). We can observe that as the number of examples per class increases (1->3->5), both the train-train and train-validation methods achieve a better performance. But since 5 is still $O(1)$, we should still expect the train-validation method outperforms the train-train method by a large margin.
> | Number of samples per class | 1                  | 3                  | 5                  |
> |----------------------------|--------------------|--------------------|--------------------|
> | Train-train                | 25.93 $\pm$ 1.10\% | 36.23 $\pm$ 0.82\% | 38.76 $\pm$ 0.66\% |
> | Train-validation           | 46.15 $\pm$ 1.36\% | 59.97 $\pm$ 1.02\% | 63.18 $\pm$ 0.90\% |
>
> ----
> **Q3**: “How does the results compare to Reptile[3], which does not need a train-validation split.”
>
> **A3**: We have added experiments to compare the performance of Reptile and FOMAML (with train-validation). The result (in terms of test accuracy) is summarized in the following table. We can see that on both RainbowMNIST and miniImagenet datasets,, FOMAML with train-validation outperforms Reptile by a small margin. We have also included this result in our newly added Appendix B.1. Note that Reptile does not perform gradient descent in the outer-loop, which is a major difference from FOMAML. We have also added some discussion on Reptile in Appendix A.2.
> | Algorithm | RainbowMNIST       | miniImagenet       |
> |-----------|--------------------|--------------------|
> | FOMAML    | 87.52 $\pm$ 0.20\% | 46.15 $\pm$ 1.36\% |
> | Reptile   | 84.97 $\pm$ 0.26\% | 45.19 $\pm$ 1.31\% |

---

> ### Author Response · Authors · 2022-11-17
> **Follow up with Reviewer SqBj**
>
> Dear reviewer, since the revision deadline is fast approaching, we would like to follow up with you to see if our response has addressed your questions/concerns on our work. We are looking forward to your feedback. Thank you very much!

---

### Official Review · Reviewer_7Hwe · 2022-10-25

**Confidence:** 3
**Correctness:** 2
**Technical Novelty And Significance:** 2
**Empirical Novelty And Significance:** 1
**Recommendation:** 3

**Clarity, Quality, Novelty And Reproducibility:**

**Clarity**

The paper is written in a succinct way - it unambiguously states the problem it wants to analyze, clearly defines the notations and describes the lemmas. Empirical evaluation setup is also described well.

**Quality and Novelty**

The proposed work is one of the very few which explores the usefulness of train-train vs train-val strategy used in episodic meta-learning setup. The theoretical framework borrows ideas from Bai et al., however, the results they found are different so there is novelty in that aspect. The quality of the paper is still in doubt - especially with respect to the contradictory empirical vs theoretical results when compared to Bai et al. I would like authors to comment on it before I finalize my thoughts on the quality/novelty aspect.

**Reproducibility**
This is mostly a theoretical paper and implementation ideas analyzed in this paper are well-known.

**Details Of Ethics Concerns:**

N/A.

**Strength And Weaknesses:**

**Strengths**

* The idea of training using a train split and back-propagating using the loss on the validation split has been a longstanding practice in the meta-learning literature - however, there has not been a lot of interest in analyzing that, either empirically of theoretically. Only existing work on this is from [Bai et al.](https://arxiv.org/pdf/2010.05843.pdf) which the authors discuss here. Theoretically analyzing whether train-val split is needed for meta-learning is an important topic.
* Most of the assumptions made in terms of proving the bounds make sense to me. Authors use a Huberized-ReLU instead of a normal ReLU to simplify the proof and show that empirically it does not make much difference. I have not looked at the derivation in details provided in the appendix - the lemmas stated in the main manuscript makes sense to me at a high-level.
* Authors perform experiments using synthetic datasets and also using a two-layer network on standard datasets like Mini-ImageNet which corroborates their theoretical findings.

**Weakness**

For me, the biggest weakness or doubt about this paper that I have in mind is how well these results will extrapolate to other type of networks or networks with more layers. There is a discrepancy in terms of findings between this work and Bai et al. where Bai et al. claimed train-train split is enough with a linearized network and this work claims that their hypothesis is not valid when using a two-layer convolutional network. However, the empirical results of Bai et al. speaks a different story - if we see their Table 1 (page 11), they experiment with the same convolutional backbone as this paper on Mini-ImageNet and Tiered-ImageNet and show that train-train works equal or better compared to train-val, which contradicts the claims made in this paper.

Along the same line, my other question is - is there any guarantee that claims made in this paper will hold true for a different kind of architecture e.g. vision transformers or if we simply increase the number of layers in the convolutional network/add a normalization layer/add a residual connection? If these networks are too complicated to be analyzed theoretically, I think some empirical evaluation using train/train vs train/val strategy will help to justify the claims made in this paper.


**Summary Of The Paper:**

In this work, authors provide a theoretical analysis regarding the usage of training data only during episodic training in a meta-learning setup against train each episode using train and compute the loss on a validation split to propagate back through the network. The second strategy is commonly used by practitioners, however, some existing relevant work argue that it might not be required. In this work, authors show that using a train-val split strategy can achieve lower test loss in a meta-learning setup (when used with a two-layer convolutional network and first-order MAML as the meta-learning algorithm).

**Summary Of The Review:**

The paper is well-written, analyzes an important implementation details from the meta-learning literature from a theoretical viewpoint and provides bounds which show that train-val is a better strategy for the model to generalize as opposed to train-train which is the common practice in the literature. However, the empirical and theoretical results are in conflict with the previous work from Bai et al. which casts doubts about the contributions of this paper.

I'll be willing to modify my score if authors can clarify my confusion about the results from Bai et al.

---

> ### Author Response · Authors · 2022-11-15
> **Response to Reviewer 7Hwe**
>
> Thank you for your detailed comments. We address your questions and concerns as follows.
>
> **Q1**: “how well these results will extrapolate to other type of networks or networks with more layers.”
>
> **A1**: We have added several experiments to show that our results extend to other neural network structures, including CNN with more layers and ResNet18. Note that in our current experiments, we used CNN with 4 layers and normalization layer. We summarize our results in the following table. We can see that FOMAML with train-validation outperforms FOMAML with train-train by a large margin under all 3 neural network structures. We have also included these results in our newly added Appendix B.
> | Setting          | CNN 4 layers + normalization | CNN 6 layers + normalization | ResNet 18           |
> |------------------|------------------------------|------------------------------|---------------------|
> | Train-train      | 25.93 $\pm$ 1.10\%           | 26.89 $\pm$ 1.18\%           | 35.63 $\pm$ 1.26\%  |
> | Train-validation | 46.15 $\pm$ 1.36\%           |  50.89 $\pm$ 1.39\%          |  55.32 $\pm$ 1.43\% |

---

> > ### Author Response · Authors · 2022-11-15
> > **Response to Reviewer 7Hwe**
> >
> > **Q2**: “There is a discrepancy in terms of … contradicts the claims made in this paper.”
> >
> > **A2**: We have added a new section in Appendix A to compare our work with Bai et al. 2021. We believe that the key reason why our paper’s conclusion is different from Bai et al. 2021’s conclusion is because (a) Bai et al. 2021 has a regularization term in the inner loop controlled by $\lambda$ for both train-train and train-validation, while we do not have this regularization term (which makes our loss function different from Bai et al. 2021). This regularization term is popularized by [1] and [2] (for example, see Eq.(3) of [1] and Eq.(1) of [2]). In practice, many applications of meta-learning do not incorporate regularizer in the inner loop [4-7]. (b) The experiment in Bai et al. 2021 does not use FOMAML but other algorithms for training. More specifically, Bai et al. 2021 carried out experiments using iMAML [1] for train-validation and Meta-MiniBatchProx [2] for train-train. Although both iMAML and Meta-MiniBatchProx incorporate regularization in the inner loop, the idea behind iMAML is more closely related to MAML, whereas the idea behind Meta-MiniBatchProx is more closely related to Reptile [3], which does not perform gradient descent in the outer-loop. We believe that the differences in the loss function and the training algorithms contribute to the discrepancy between our experimental results and that in Bai et al. 2021. For a more detailed discussion, please see our newly added section in Appendix A. We have also performed more experiments to compare iMAML (with train-validation), Meta-MiniBatchProx (with train-train), and FOMAML (with train-validation and with train-train). We run our experiments on both miniImagenet and RainbowMNIST datasets for the 5-way 1-shot problem. The results are shown in the following table. Based on our experiments, adding a regularizer in the inner-loop makes the difference between the train-train (Meta-MiniBatchProx) and train-validation (iMAML) methods smaller, as compared with the difference when using FOMAML without a regularizer in the inner-loop. But the train-validation (iMAML) method still outperformed the train-train (Meta-MiniBatchprox) method slightly. On miniImagenet, it is also worthwhile mentioning that the performance of Meta-MiniBatchProx outperforms iMAML in Bai et al. 2021 since they allow all meta-testing tasks to use the same Adam optimizer in the inner-loop. In this case, the second order momentum is shared across different tasks, which causes information leakage to some extent. Since different meta-testing tasks may have overlapping classes, such information leakage can potentially lead to better performance of Meta-MiniBatchProx.
> > | Data set     | iMAML                 | Meta-MiniBatchProx | FOMAML(train-val)     | FOMAML(train-train)    |
> > |--------------|-----------------------|--------------------|--------------------|------------------|
> > | miniImagenet | 47.53 $\pm$ 1.33\%    | 46.91 $\pm$ 1.35\% | 46.15 $\pm$ 1.36\% | 25.93 $\pm$1.10% |
> > | RainbowMNIST | 88.07 $\pm$ 0.21\%    | 86.59 $\pm$ 0.23\% | 87.52 $\pm$ 0.20\% | 65.32 $\pm$0.54% |
> >
> > [1] Rajeswaran, Aravind, Chelsea Finn, Sham M. Kakade, and Sergey Levine. "Meta-learning with implicit gradients." Advances in neural information processing systems 32 (2019).
> >
> > [2] Zhou, Pan, Xiaotong Yuan, Huan Xu, Shuicheng Yan, and Jiashi Feng. "Efficient meta learning via minibatch proximal update." Advances in Neural Information Processing Systems 32 (2019).
> >
> > [3] Nichol, Alex, Joshua Achiam, and John Schulman. "On first-order meta-learning algorithms." arXiv preprint arXiv:1803.02999 (2018).
> >
> > [4] Zhang, Xi Sheryl, Fengyi Tang, Hiroko H. Dodge, Jiayu Zhou, and Fei Wang. "Metapred: Meta-learning for clinical risk prediction with limited patient electronic health records." In Proceedings of the 25th ACM SIGKDD International Conference on Knowledge Discovery & Data Mining, pp. 2487-2495. 2019.
> >
> > [5] Lu, Yuanfu, Yuan Fang, and Chuan Shi. "Meta-learning on heterogeneous information networks for cold-start recommendation." In Proceedings of the 26th ACM SIGKDD International Conference on Knowledge Discovery & Data Mining, pp. 1563-1573. 2020.
> >
> > [6] Yu, Tianhe, Chelsea Finn, Annie Xie, Sudeep Dasari, Tianhao Zhang, Pieter Abbeel, and Sergey Levine. "One-shot imitation from observing humans via domain-adaptive meta-learning." arXiv preprint arXiv:1802.01557 (2018).
> >
> > [7] Rußwurm, Marc, Sherrie Wang, Marco Korner, and David Lobell. "Meta-learning for few-shot land cover classification." In Proceedings of the ieee/cvf conference on computer vision and pattern recognition workshops, pp. 200-201. 2020.

---

> > > ### Comment · Reviewer_7Hwe · 2022-11-16
> > > **Better summarizing the findings**
> > >
> > > Hi, thanks for getting back to my comments. If I understand the new results, are we saying that whether train-train is better or train-validation is better is dependent on the underlying meta-learning algorithm and also whether a regularization is used or not? This may require you to change the narrative of the paper to a certain extent. On that note, I also have not used the regularization in the inner loop for training meta-learning models in the past so I agree with your statement there.
> > >
> > > One more thing - "More specifically, Bai et al. 2021 carried out experiments using iMAML [1] for train-validation and Meta-MiniBatchProx [2] for train-train" - are you saying that Bai et al. used a different algorithm for train-train vs train-val and reported results? That does not sound plausible to me.

---

> > > > ### Author Response · Authors · 2022-11-16
> > > > **Re: Better summarizing the findings**
> > > >
> > > > Thank you for your quick feedback!
> > > >
> > > > **Q1**: “If I understand the new results, … whether a regularization is used or not?”
> > > >
> > > > **A1**: Based on our theoretical analysis of FOMAML, and our experiments including the newly added experiments, we have shown that train-validation outperforms train-train for FOMAML. For other meta learning algorithms, according to our experiments, we also found that train-validation outperforms train-train. Adding a regularizer in the inner-loop can make the gap between the train-train and train-validation smaller, but according to our experiments (See above), iMAML (train-validation) is still better than Meta-MiniBatchProx (train-train).
> > > >
> > > > ----
> > > >
> > > > **Q2**: “This may require you to change the narrative of the paper to a certain extent.”
> > > >
> > > > **A2**: Thank you for your suggestion. We have already modified our narrative in the paper (e.g., abstract, introduction, etc.)  to make this point clear.
> > > >
> > > > ----
> > > >
> > > > **Q3**: “One more thing, … and reported results?”
> > > >
> > > > **A3**: When discussing their loss function for train-train and train-validation, Bai et al. 2021 wrote “We note that both loss functions above have been considered in prior work ($L^{tr-val}$ in iMAML (Rajeswaran et al., 2019), and $L^{tr-tr}$ in Meta-MinibatchProx (Zhou et al., 2019)), though we use slightly different implementation details from these prior work to make sure that the two methods here are exactly the same except for whether the split is used.” However, they did not provide the details about the modification or implementation. We have communicated with the authors of Bai et al. 2021 and received their code. Their implementation of the train-train method is the same as Meta-MiniBatchProx (Zhou et al., 2019). But we did not find the implementation of iMAML (the train-validation method) in their code. Their results for the train-train method is exactly the same as the results reported in Meta-MinibatchProx (Zhou et al., 2019) (see the last row in Tables 1 and 2 of Meta-MinibatchProx (Zhou et al., 2019) and compare it with the train-train result from Table 1 of Bai et al. 2021). As we have mentioned in the previous response, the original code of Meta-MinibatchProx (Zhou et al., 2019) reuses the second-order momentum information during the test phase, which may lead to an unfair comparison (i.e., one should not use the information of one test task to improve the other test task). So we implemented iMAML and a corrected version of Meta-MiniBatchProx by ourselves. According to our experiments (See above), iMAML outperforms Meta-MiniBatchProx, which is opposite to the results reported in Bai et al. 2021.

---

> ### Author Response · Authors · 2022-11-17
> **Follow up with reviewer 7Hwe**
>
> Dear reviewer, since the revision deadline is fast approaching, we would like to follow up with you to see if our response has addressed your questions/concerns on our work. We are very happy to make further revision if you have any other suggestions or questions. If you are satisfied with our response, we sincerely hope you could reconsider your score. Thank you very much!

---

> ### Author Response · Authors · 2022-11-29
> **We would appreciate your feedback**
>
> Dear reviewer, we understand you may have a busy schedule, but we believe that we have addressed all your questions/concerns. If you have any further comments or questions, we are very happy to answer them. If you are satisfied with our responses so far, we sincerely hope you could reconsider your score. Thank you very much!

---

> > ### Comment · Reviewer_7Hwe · 2022-12-01
> > **Appreciate your efforts in responding to my comments**
> >
> > I appreciate all the efforts you have put in to respond to my comments and also reach out to Bai et al. to get their code and analyze it. So Bai et al. used two different algorithms for train-train and train-val but tried to ensure that the comparison is fair - that makes sense to me, thanks for getting it clarified. From your investigation, it seems like you have found another outcome that contradicts the results from Bai et al. but we can ignore that for now as it does not change the main findings.
> >
> > Overall, I went through the latest manuscript and I am not convinced that what we discussed is captured well in the updated narrative. The paper still says train-val works better than train-train for CNNs whereas Bai et al. showed opposite results for CNNs and what seems to be causing this contradiction in reality is the specific implementation details of the meta-learner w.r.t. to the regularizer. That should be at the forefront of the narrative to alleviate the confusion as to what is truly better and what all factors it depend on.
> >
> > It's possible that I may have missed something as I didn't spend a lot of time going through the updated narrative - please feel free to correct me.

---

> > > ### Author Response · Authors · 2022-12-01
> > > **Clarification**
> > >
> > > Dear reviewer,
> > >
> > > Thank you for your feedback. We are sorry that you feel the current revision does not capture what has been discussed so far. We, however, do believe that we have made the point clear. In all of our conclusions, we have stated that train-validation outperforms train-train for **FOMAML** in learning CNN. After all, all of our analysis is based on the FOMAML algorithm, which does not have any regularizers. Thus, our conclusion is accurate. In addition, we would like to emphasize that based on the code we get from Bai et al., their comparison of train-validation and train-train is not fair due to their problematic implementation of Meta-MiniBatchProx (Zhou et al., 2019), which we have discussed in the previous reply. We ourselves implemented the corrected version of Meta-MiniBatchProx, which makes the comparison fair. Under the fair comparison, train-validation still outperformed train-train even with regularizers. Overall, we have made it clear that our analysis and conclusions are based on the **FOMAML** algorithm which does not have regularizers; and even for algorithms with regularizers, our fair comparison shows that train-validation is still better than train-train.

---

### Official Review · Reviewer_YDGh · 2022-10-26

**Confidence:** 5
**Correctness:** 3
**Technical Novelty And Significance:** 2
**Empirical Novelty And Significance:** 3
**Recommendation:** 5

**Clarity, Quality, Novelty And Reproducibility:**

The clarity and quality of the work are its main problem, with the writing clarity affecting both the overall clarity of the work as well as its precision.

The work seems reproducible from what is in the paper.

**Strength And Weaknesses:**

Strengths:

- Clear empirical evaluation, with thorough experiments.
- Inclusion of both theoretical justification and empirical results.

Weaknesses:

- Writing quality is clumsy, and at worst significantly affecting clarity and precision of what is being communicated.

**Summary Of The Paper:**

The authors conduct an investigation into the importance of having train-validation sets for episodic few-shot learning, learned by meta-learning. They do as such by both theoretical proofs and empirical evaluation.

While the intuitions behind having a train - validation set for meta-learning are quite clear and reasonable, this is a very good complementary study that showcases that the intuitions do in fact measure up to reality.

**Summary Of The Review:**

I recommended a weak reject due to the writing quality being problematic enough to cause precision and clarity issues in the work. If that is improved, I'd be happy to accept this paper.

---

> ### Author Response · Authors · 2022-11-15
> **Response to Reviewer YDGh**
>
> Thank you for your constructive comments. We have revised the paper and improved the clarity. If you have any additional concerns about the clarity, please kindly point out the unclear part and we are happy to further revise the paper.

---

> ### Author Response · Authors · 2022-11-17
> **Follow up with Reviewer YDGh**
>
> Dear reviewer, since the revision deadline is fast approaching, we would like to follow up with you to see if our response has addressed your questions/concerns on our work. We are looking forward to your feedback. Thank you very much!

---

### Official Review · Reviewer_kJcp · 2022-10-26

**Confidence:** 2
**Correctness:** 3
**Technical Novelty And Significance:** 3
**Empirical Novelty And Significance:** 2
**Recommendation:** 6

**Clarity, Quality, Novelty And Reproducibility:**

Overall, the quality and clarity are good, and the originality is great since there is few work studying train-validation split of meta-learning with CNN.

**Strength And Weaknesses:**

Strength:
- There is few research work studying train-validation split of meta-learning with neural networks, so it is great to have such work targeting on the theoretical foundations.
- The theoretical analyses seem to be solid but I have not checked the proof details.

Weakness:
- What does it mean by "FOMAML" (first appears at Page 2)? It seems no where the term is introduced.
- There is a little gap between the theories and the experiments, namely it would be great to have some synthetic data and experiments to meet the theoretical assumptions and to see how it matches to the theoretical claims. For example, we can generate synthetic data by Definition 3.2, then we can use a two-layer CNN described in the theoretical analysis as the backbones. To verify Thm 4.4 and 4.5, we can set different values of $d$ in synthetic data and to confirm the training loss is at the order of O(1/poly(d)) as claimed in the theorems. Similarly for the test loss, we want to see a figure plotting the test loss evolution, where train-train test loss converges to some constant while train-validation test loss continues to decrease. I think the theories and experiments can be more closely connected in this way.

**Summary Of The Paper:**

The paper analyzes the train-validation split in meta-learning for classification problems with a two-layer CNN model. It proves that the train-validation split is necessary to learn a good prior model compared with a train-train model. Experiments justify the theoretical claims.

**Summary Of The Review:**

Based on the novelty and strengths discussed above, I am close to an acceptation of the paper.

---

> ### Author Response · Authors · 2022-11-15
> **Response to Reviewer kJcp**
>
> Thank you for your supportive comments. We address your comments as follows. \
> **Q1**: “FOMAML”.
>
> **A1**: Thank you for pointing it out. We have added its full name and a reference the first time we introduce it.
>
> ----
> **Q2**: “To verify Thm 4.4 and 4.5, …  of O(1/poly(d)) as claimed in the theorems”
>
> **A2**: It is difficult to perform numerical experiments to verify the $O(1/poly(d))$ rate. Our training loss is at the order of $\tilde{O}(1/(T\eta))$, where $T$ is the total number of iterations and $\eta$ is the step size of the outer loop (see Theorem E.10 and Theorem F.20 in the Appendix of the revised version). The reason that the training loss is at the order of $O(1/poly(d))$ is because we choose $T=poly(d)$ for early stopping. But this choice is not essential as long as T is sufficiently large.
>
> ----
> **Q3**: “Similarly for the test loss, … train-validation test loss continues to decrease.”
>
> **A3**: Thanks for the suggestion. We have added a plot showing the evolution of the test loss over time for the train-train and train-validation method in Appendix B.2 (Figure 1). As shown in the plot, the test loss for the train-validation method decreases almost monotonically, whereas the test loss for the train-train method first decreases and then increases due to overfitting.

---

> ### Author Response · Authors · 2022-11-17
> **Follow up with Reviewer kJcp**
>
> Dear reviewer, since the revision deadline is fast approaching, we would like to follow up with you to see if our response has addressed your questions/concerns on our work. We are looking forward to your feedback. Thank you very much!

---

### Decision · Program_Chairs · 2023-01-20

**Decision:**

Accept: poster

**Justification For Why Not Higher Score:**

It is a good paper that is worth accepting. The paper claims the importance of train-valid splits for a very specific meta-learning algorithm - FOMAML. Hence the recommendation for poster and not spotlight.

**Justification For Why Not Lower Score:**

This is a very useful discussion to have in the meta-learning community and there are no major issues with the paper that requires rejection.

**Metareview: Summary, Strengths And Weaknesses:**

This paper focuses on the benefits of the train-validation split for FOMAML and shows both theoretically and empirically that it is better to use a train-valid split than using train-train setups.

The main criticism against this paper is that the results in this paper contradict the results of Bai et al. 2021. However, the authors have made a very convincing rebuttal explaining the problem with the setup of Bai et al. 2021 and why there is a contradiction. Further, the authors have added more results which strengthen their claims.

Given that I am convinced by their rebuttal, I recommend an acceptance. However, I strongly recommend the authors add all the new results in the paper and also improve the clarity of the paper. Also, make sure to not overclaim the results (which I see that you have already done in the revision).



**Note From Pc:**

if the above contains the word "oral" or "spotlight" please see: "oral" presentation means -> notable-top-5% and "spotlight" means -> notable-top-25%. As stated in our emails, we are disassociating presentation type from AC recommendations